# Extracellular vesicles engineering by silicates-activated endothelial progenitor cells for myocardial infarction treatment in male mice

Bin Yu[1,2,8], Hekai Li[3,8], Zhaowenbin Zhang [4,5,6,8], Peier Chen[3], Ling Wang [7], Xianglin Fan[3], Xiaodong Ning[1], Yuxuan Pan[1], Feiran Zhou[3], Xinyi Hu[3], Jiang Chang [1,4,5,6] ✉ & Caiwen Ou [1,2,3] ✉

Extracellular vesicles have shown good potential in disease treatments including ischemic injury such as myocardial infarction. However, the efficient production of highly active extracellular vesicles is one of the critical limitations for their clinical applications. Here, we demonstrate a biomaterial-based approach to prepare high amounts of extracellular vesicles with high bioactivity from endothelial progenitor cells (EPCs) by stimulation with silicate ions derived from bioactive silicate ceramics. We further show that hydrogel microspheres containing engineered extracellular vesicles are highly effective in the treatment of myocardial infarction in male mice by significantly enhancing angiogenesis. This therapeutic effect is attributed to significantly enhanced revascularization by the high content of miR-126a-3p and angiogenic factors such as VEGF and SDF-1, CXCR4 and eNOS in engineered extracellular vesicles, which not only activate endothelial cells but also recruit EPCs from the circulatory system.

Myocardial infarction is a disease that occurs due to sudden occlusion of the coronary arteries leading to ischemic necrosis of the heart muscle[1]. One of the most important strategies for cardiac repair after myocardial infarction is the promotion of angiogenesis in the border zone of the myocardial infarction, which can effectively prolong the survival of damaged myocardium and improve ventricular remodeling and cardiac function[1]. Traditionally, blood vessels in ischemic tissue are rebuilt primarily by endothelial cells adjacent to the site of injury[2,3].

However, mature endothelial cells are terminally differentiated cells with insufficient proliferative potential and long-distance migration, so their ability to remodel the vascular network at the site of myocardial infarction is limited[4]. Interestingly, endothelial progenitor cells (EPCs), as the precursor cells of endothelial cells, provide a new perspective for research on postischemic revascularization[5,6]. EPCs have a high proliferative and differentiation capacity and can induce neovascularization in ischemic areas through two independent mechanisms

[1]The 10th Affiliated Hospital of Southern Medical University (Dongguan People's Hospital), Southern Medical University, Guangdong Provincial Key Laboratory of Cardiac Function and Microcirculation, 510280 Guangzhou, China. [2]Department of Rehabilitation Medicine, Zhujiang Hospital, Southern Medical University, 510280 Guangzhou, China. [3]Department of Cardiology and Laboratory of Heart Center, Zhujiang Hospital, Guangdong Provincial Biomedical Engineering Technology Research Center for Cardiovascular Disease, Southern Medical University, 510515 Guangzhou, China. [4]Wenzhou Institute, Zhejiang Engineering Research Center for Tissue Repair Materials, University of Chinese Academy of Sciences, 325000 Wenzhou, China. [5]State Key Laboratory of High-Performance Ceramics and Super fine Microstructure, Shanghai Institute of Ceramics, Chinese Academy of Sciences, 200050 Shanghai, People's Republic of China. [6]Joint Centre of Translational Medicine, The First Affiliated Hospital of Wenzhou Medical University, 325000 Wenzhou, China. [7]School of Biomedical Engineering, Biomaterials Research Center, Southern Medical University, 510515 Guangzhou, People's Republic of China. [8]These authors contributed equally: Bin Yu, Hekai Li, Zhaowenbin Zhang. ✉e-mail: jchang@mail.sic.ac.cn; oucaiwen@smu.edu.cn

because of two subgroups named early outgrowth cells (EOCs) and endothelial colony-forming cells (ECFCs)[7,8]. On the one hand, ECFCs can differentiate into mature endothelial cells and directly promote neovascularization by incorporating themselves into newly formed vessels[8]. On the other hand, EOCs can stimulate the proliferation and migration of endothelial cells and recruit more EPCs from the blood circulation system by releasing proangiogenic factors and chemokines such as stromal cell-derived factor 1 (SDF-1) and vascular endothelial growth factor (VEGF) in a paracrine manner[9–11], although EOCs themselves do not form new blood vessels, which indirectly stimulates angiogenesis[12,13]. However, since the number and migratory capacity of circulating EPCs are inversely associated with the risk of coronary artery disease[14], only a few EPCs are able to actually reach the myocardial infarction area[15–17]. This issue severely limits the repair effect of EPCs on myocardial infarction. Previous studies have shown that targeted delivery of EPCs, such as transplantation of EPCs to the marginal area of myocardial infarction, can improve the therapeutic effect of EPCs[15,17,18]. However, hardly any EPCs can survive in harsh conditions such as ischemia, inflammation, oxidative stress and mechanical stress, and these exogenous cells also carry the risk of immune rejection, tumorigenicity and infusion toxicity, thus limiting their clinical application[18,19].

Extracellular vesicles (EVs), tiny membrane vesicles with lipid bilayer membranes, can carry proteins, mRNAs, miRNAs and other bioactive substances from source cells and protect them from disruption by nucleases and proteases. Therefore, EVs are an important medium for information transmission and material exchange among cells[20]. A study has shown that EPC-derived EVs have similar biological properties as EPCs and can effectively promote cardiac repair after myocardial infarction[20]. Furthermore, compared with stem cells, EVs have the advantages of small size, low toxicity, low immunogenicity and good permeability, so they have become a powerful alternative for cell therapy[21,22]. However, EV therapy still has problems such as low yield, unstable biological activity, and low tissue retention[23]. Therefore, for the clinical application of EV therapy, it is important to develop new extracellular vesicle engineering approaches to produce high-yield EVs with higher bioactivity.

Silicate bioceramic-derived ionic solutions have been found to be highly bioactive in stimulating the proangiogenic capacity of endothelial cells and significantly enhancing cell–cell paracrine signaling[24,25]. Other studies showed that silicate ions could regulate the main components of EVs (such as the content of miRNA and cytokines) secreted by bone marrow mesenchymal cells to improve their biological functions, thereby promoting bone regeneration[26–28]. Therefore, we hypothesize that silicate bioceramic-derived ionic solutions may be able to induce EPCs to produce high-yield EVs with higher bioactivity for the effective treatment of myocardial infarction. To test our hypothesis and explore the regulatory effect and mechanism of silicate bioceramic ionic solutions on EV production by EPCs, we developed a protocol to treat EPCs with silicate ion solutions derived from calcium silicate (CS, a typical silicate bioceramic) and isolated silicate ion solution-activated EVs (CS-EPC-EVs), analyzed their effect on the proangiogenic and antiapoptotic effects and recruitment of progenitor cells of human umbilical vein endothelial cells (HUVECs), and explored the regulatory mechanism of silicate ions on miRNA expression sorting and secretion in EVs using high-throughput sequencing technology. Furthermore, most studies have used EV injection to treat myocardial infarction within 72 h[29–31], and some studies have found that the treatment effect of EVs alone on myocardial infarction within 28 days was not as good as that of EVs encapsulated by hydrogels[32,33]. Based on these studies, we speculated that the low retention rate of EVs in vivo may be one of the main reasons for the limited therapeutic effect. To overcome the problem of low tissue retention of locally

injected EVs, we used microfluidic technology to prepare PEG/GelMA injectable hydrogel microspheres encapsulated with silicate ion-activated EVs (microsphere+CS-EPC-EV) for the treatment of myocardial infarction in a mouse model (Fig. 1).

Here, we show that silicate ion solutions stimulate the high-yield production of highly bioactive EVs in EPCs and that highly bioactive EVs delivered in the form of hydrogel microspheres are effective in the treatment of myocardial infarction, suggesting that this biomaterial-based approach is effective for the preparation of high-yield bioactive extracellular vesicles from EPCs for the treatment of ischemic tissue and organ injury.

## Results

### Activation of EPCs to secrete highly bioactive EVs by silicate ions

First, EPCs isolated from C57Bl/6 mice were characterized by CD133 (Biolegend, 141203), CD34 (BD, 560238), VEGFR2 (Thermo, 17-5821-81) immunofluorescence staining and flow cytometry[34,35]. The results showed that $85.3 \pm 0.2\%$ of the cells expressed CD133, $80.9 \pm 0.4\%$ expressed CD34, and $91.9 \pm 0.4\%$ expressed VEGFR2. Since CD133, CD34 and VEGFR2 are the main cell markers of EPCs[34,35], this result suggests that EPCs were successfully isolated and could be used for subsequent experiments (Supplementary Fig. 1). Furthermore, EOCs within the isolated EPCs were characterized by the coexpression of CD133, CD34 and VEGFR2 (CD133+/CD34+/VEGFR2+), and ECFCs were characterized by the coexpression of CD34 and VEGFR2 (CD133-/CD34+/VEGFR2+) or CD133 and VEGFR2 (CD133+/CD34−/VEGFR2+)[7,8,36–39]. Therefore, in the isolated EPCs, approximately $71.61 \pm 2.84\%$ of cells were EOCs, and $26.08 \pm 2.68\%$ of cells were ECFCs (Supplementary Fig. 2).

To determine the optimal bioactive concentration of silicate ion solutions for the regulation of EPCs, we used the 3-(4,5-dimethylthiazol-2-yl)-5-(3-carboxymethoxyphenyl)-2-(4-sulfophenyl)-2H-tetrazolium assay (MTS) to compare the proliferation of EPCs cultured in different concentrations of silicate ion solutions. As shown in Fig. 2a, culture media containing silicate ions in different dilutions from 1/4 to 1/256 could significantly enhance the proliferation of EPCs in comparison to normal media (Endothelial Cell Growth Medium-2, EGM-2), especially in the dilution range between 1/64 and 1/256. Next, qRT–PCR was used to examine the expression of proangiogenesis-related genes in EPCs on Days 2 and 4 after culturing in silicate ion-containing media. Obviously, compared with normal medium, 1/64, 1/128, and 1/256 dilutions of the silicate ion solution significantly promoted the expression of VEGFA, eNOS, SDF-1, IGF-1, and HGF in EPCs, in which the 1/128 dilution showed the highest stimulatory effect (Fig. 2b–f). Therefore, we further used a 1/128 dilution of the silicate ion solution (1/128CS) to stimulate EPCs for EV production in follow-up studies.

Furthermore, ICP–AES was used to detect calcium (Ca), silicon (Si) and phosphorus (P) ion concentrations in different culture media. The results showed that Ca and P ion concentrations in the 1/64, 1/128, and 1/256 CS dilutions were similar to those in the EGM-2 medium, but the Si ion concentration ($0.67–1.84 \mu g/mL$) was significantly higher than that in EGM-2 medium ($0.33 \mu g/mL$) (Supplementary Table 1). This result indicates that silicate ions were the main factor regulating EPC activity.

After the optimal activity concentration of silicate ions was determined, a 1/128 silicate ion dilution was used to stimulate EPCs to secrete EVs (CS-EPC-EVs). Transmission electron microscopy (TEM) clearly showed that CS-EPC-EVs exhibited a typical goblet vesicle structure similar to that of EPC-derived EVs under conventional culture conditions (EPC-EVs) (Fig. 2g). Nanoparticle tracking analysis (NTA) showed that both CS-EPC-EVs and EPC-EVs were distributed in a single bell-shape pattern, and the peak value of EPC-EVs appeared at 111 nm, while that of CS-EPC-EVs appeared at 109 nm (Fig. 2h). Since the particle size was 40–150 nm, the obtained EVs were small EVs[40]. In addition, no particles were found in the CS ion solution, suggesting that the

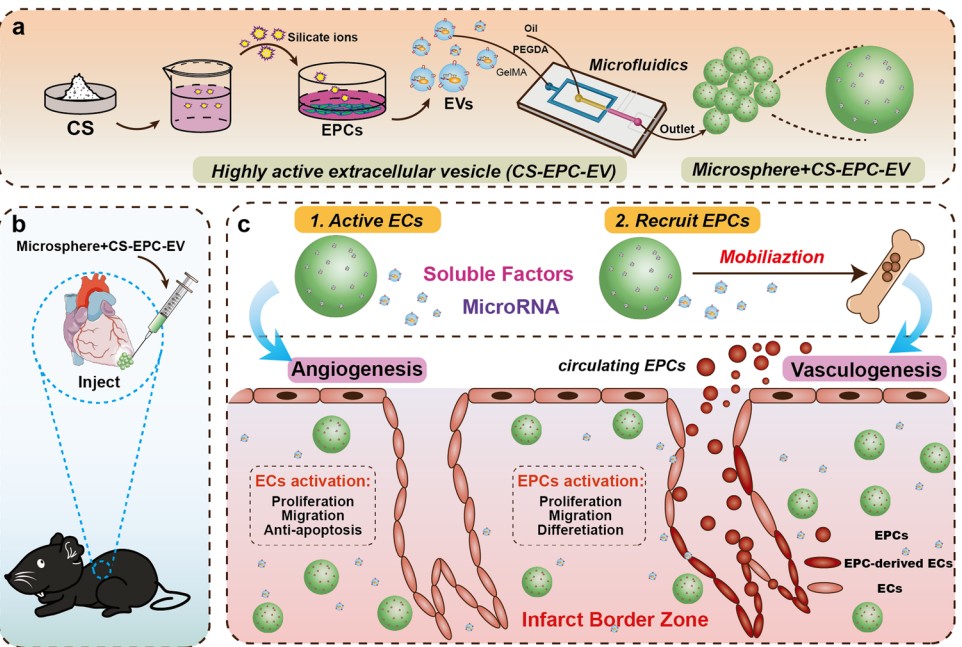

**Fig. 1 | High-efficiency engineering of highly active extracellular vesicles by the treatment of endothelial progenitor cells (EPCs) with silicate ions derived from bioactive ceramics for myocardial infarction therapy through enhancement of revascularization of infarcted tissues. a** EPCs were induced by calcium silicate (CS) ions to secrete highly active extracellular vesicles (CS-EPC-EV), and microspheres loaded with highly active extracellular vesicles (microsphere+CS-EPC-EVs) were prepared by microfluidic technology. **b** In situ injection of microsphere+CS-EPC-EVs at the site of myocardial infarction in mice to repair myocardial injury. **c** The mechanism of microspheres for the treatment of myocardial infarction. On the one hand, the extracellular vesicles in the microspheres inhibit apoptosis and promote proliferation, migration and angiogenesis of surviving endothelial cells (ECs) in the infarct border zone of myocardial infarction. On the other hand, extracellular vesicles recruit EPCs generated in the bone marrow from the circulatory system to the infarct border zone, which integrate themselves into damaged blood vessels through proliferation, migration, and differentiation, thereby promoting angiogenesis. This Figure was created with BioRender.com.

CS ion solution does not affect the determination of EV yield (Supplementary Fig. 3). Interestingly, quantitative analysis of NTA revealed that the particle concentration of CS-EPC-EVs was significantly higher than that of EPC-EVs, indicating that silicate ions improved the yield of EVs in EPC culture (Fig. 2i). The purities of CS-EPC-EVs and EPC-EVs were $(2.32 \pm 0.14) * 10^9$ and $(2.43 \pm 0.19) * 10^9$ particles/µg, respectively (Supplementary Table 7). Furthermore, Western blot analysis showed that both CS-EPC-EVs and EPC-EVs expressed the characteristic proteins of EVs, such as Alix, CD81, and CD63, which further demonstrated that we successfully isolated the desired EVs from the supernatant of EPC cultures (Fig. 2j). Next, to verify whether silicate ions regulate the content of proangiogenic factors in EPC-EVs, we assessed the expression levels of key angiogenic factors (VEGFA, eNOS, HGF, IGF-1 and SDF-1) in CS-EPC-EVs and EPC-EVs by ELISAs (Fig. 2k) and calculated the content of angiogenic factors in a single EV particle (Supplementary Table 5). The angiogenic factor content was as follows: EPC-EVs: VEGFA: $(1.05 \pm 0.17) * 10^{-15}$ µg; SDF-1: $(3.78 \pm 0.44) * 10^{-14}$ µg; IGF-1: $(1.63 \pm 0.05) * 10^{-16}$ µg; eNOS: $(1.09 \pm 0.05) * 10^{-13}$ µg; HGF: $(0.73 \pm 0.11) * 10^{-11}$ µg. CS-EPC-EVs: VEGFA: $(2.52 \pm 0.53) * 10^{-15}$ µg; SDF-1: $(0.70 \pm 0.06) * 10^{-13}$ µg; IGF-1: $(2.29 \pm 0.18) * 10^{-16}$ µg; eNOS: $(1.70 \pm 0.22) * 10^{-13}$ µg; HGF: $(1.24 \pm 0.01) * 10^{-11}$ µg. The maximum measurement variation was in the VEGFA groups, and the VEGFA variation value was 21% in CS-EPC-EVs and 16% in EPC-EVs. This variation was comparable to the variation in angiogenic factor content in EVs in other studies[41,42]. Interestingly, the content of angiogenic factors (VEGFA, eNOS, HGF, IGF-1) in CS-EPC-EVs was indeed significantly higher than that in EPC-EVs. More interestingly, compared with that of EPC-EVs, the content of SDF-1 (chemotactic recruitment factor of hematopoietic stem cells), which plays a critical role in the recruitment of EPCs, was also significantly increased in CS-EPC-EVs. This result indicates that CS-EPC-EVs may not only have higher angiogenic activity but also promote EPC homing.

## Stimulation of HUVEC angiogenesis by CS-EPC-EVs under oxygen and glucose deprivation

To find the optimal intervention concentration of CS-EPC-EVs on HUVECs, we first evaluated the effect of various concentrations of CS-EPC-EVs (10, 20, 40, 50, 60, 80, and 100 µg/mL) on the proliferation of HUVECs under normoxic conditions. The results showed that both EPC-EVs and CS-EPC-EVs significantly enhanced the proliferation of HUVECs, and CS-EPC-EVs exhibited a more notable effect on the proliferation of HUVECs at 50 µg/mL (Fig. 3a). Moreover, EdU staining confirmed that 50 µg/mL CS-EPC-EVs significantly enhanced the proliferation of HUVECs under oxygen-glucose deprivation conditions (OGD) (Fig. 3b, c). In addition, after treating phalloidin-labeled HUVECs (green fluorescence) with PKH-26-labeled EVs (red fluorescence) at a concentration of 50 µg/mL for 12 h, we clearly observed red fluorescence-labeled EVs in the cytoskeleton of HUVECs by confocal microscopy, indicating the endocytosis of EVs (Fig. 3d). Based on these results, the concentration of 50 µg/mL CS-EPC-EVs was chosen as the optimal concentration for further experiments.

Wound healing and Transwell experiments were used to evaluate the effect of different EVs on the migration of HUVECs in vitro under glucose-oxygen deprivation conditions. The results showed that both EPC-EVs and CS-EPC-EVs enhanced the migration of HUVECs under glucose-oxygen deprivation (Fig. 3e, f). Quantitative statistical results further revealed that CS-EPC-EVs had better migration-promoting activity than EPC-EVs (Fig. 3g). In addition, HUVECs were seeded on Matrigel-coated Petri dishes to evaluate the effect of different EVs on vascular tube formation in vitro under glucose and oxygen deprivation. As shown in Fig. 3h, compared with EPC-EVs, HUVECs treated with CS-EPC-EVs were able to form more tube networks (Fig. 3i), indicating that CS-EPC-EVs could also significantly enhance the vascular tube formation of HUVECs in vitro.

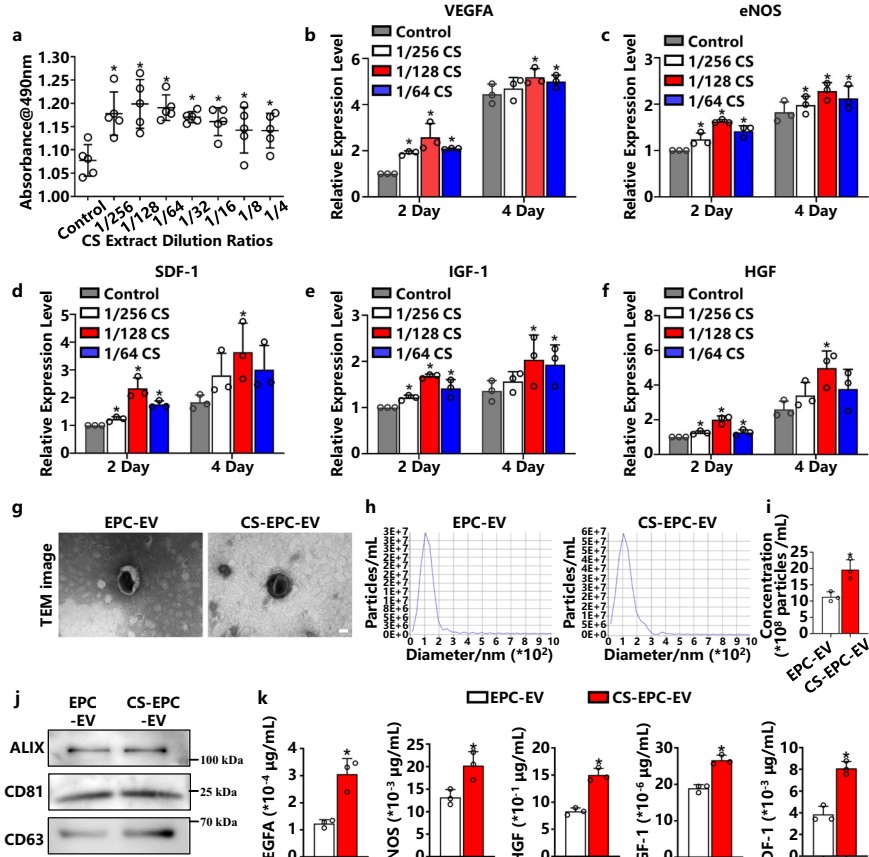

**Fig. 2 | Activation of EPCs to secrete highly bioactive EVs by silicate ions (CS-EPC-EV) or normal culture medium (EPC-EV). a** The effect of different concentrations of CS extracts on the proliferation of EPCs was tested by 3-(4,5-dimethylthiazol-2-yl)-5-(3-carboxymethoxyphenyl)-2-(4-sulfophenyl)−2H-tetrazolium (MTS) ($n = 5$, biological replicates per group) *$p < 0.05$ vs. the control. **b**–**f** The expression of proangiogenic genes was measured by quantitative reverse transcriptase Polymerase Chain Reaction (qRT–PCR) on Day 2 and Day 4 in EPCs cultured in 1/256, 1/128, and 1/64 dilutions of (**b** Vascular endothelial growth factor A, VEGFA; **c** Endothelial nitric oxide synthase, eNOS; **d** Stromal cell-derived factor 1, SDF-1; **e** Insulin-like growth factor 1, IGF-1; **f** Hepatocyte growth factor, HGF) ($n = 3$, biological replicates per group). *$p < 0.05$ vs. the control. **g** Transmission electron microscopy (TEM) image of EPC-EVs, scale bar = 50 nm, ($n = 3$ independent experiment replicates per group). **h** Nanoparticle tracking analysis (NTA) analysis of CS-EPCs-EVs and EPC-EVs. **i** Particle concentrations of different types of EVs. *$p < 0.05$ vs. EPC-EVs ($n = 3$, independent experiment replicates per group). **j** Western blot analysis of the extracellular vesicle markers ALG-2-interacting protein X (Alix), Cluster of Differentiation 81 (CD81) and Cluster of Differentiation 63 (CD63). **k** ELISAs detected the content of VEGFA, eNOS, HGF, IGF-1, and SDF-1, in different EVs ($n = 3$, biological replicates per group). *$p < 0.05$ vs. EPC-EVs. Data are presented as the mean ± standard. Two-tailed Student's $t$-test was used to compare the differences between two groups. One-way ANOVA and post hoc Bonferroni tests were used to compare differences among more than two groups. Source data are provided as a Source Data file. Each experiment was repeated 3 times or more independently with similar results.

To determine the effect of CS-EPC-EVs on the apoptosis of HUVECs under glucose and oxygen deprivation, we performed TUNEL staining on HUVECs after EV intervention. The fluorescence staining results showed that compared with the blank group (OGD), both the EPC-EV and CS-EPC-EV groups revealed a significantly lower number of TUNEL-positive cells (Fig. 3j). Quantitative analysis further showed fewer TUNEL-positive cells in the CS-EPC-EV group than in the EPC-EV group (Fig. 3k). These results fully prove that CS-EPC-EVs have higher bioactivity than EPC-EVs in preventing apoptosis of HUVECs under oxygen and glucose deprivation.

Finally, to prove the effect of CS-EPC-EVs on the angiogenic activity of HUVECs in vitro under glucose and oxygen deprivation, we analyzed the expression levels of proangiogenesis-related genes in HUVECs after EV intervention by qRT–PCR and Western blots. Compared with those of the HUVECs without any treatment, after EPC-EV and CS-EPC-EV intervention, the expression levels of five angiogenic genes (VEGFA, bFGF, IGF-1, eNOS, and SDF-1) in HUVECs were significantly increased, and the effect of CS-EPC-EVs was more significant than that of EPC-EVs (Fig. 3l). Western blot analysis further confirmed that CS-EPC-EVs significantly increased the protein expression levels of eNOS, VEGFA and SDF-1 compared with EPC-EVs (Fig. 3m, n).

## Preparation and characterization of injectable GelMA-PEG microspheres

Based on in vitro experiments, we speculate that CS-EPC-EVs may have better therapeutic efficacy for myocardial infarction than EPC-EVs. To verify the therapeutic effect of CS-EPC-EVs for the treatment of heart infarction, we established a mouse model of acute myocardial infarction (AMI) and prepared EV-loaded GelMA-PEG microspheres by microfluidic technology for sustained delivery of EVs in the infarct area (Fig. 4a). We clearly observed that the microspheres were uniformly distributed in the solution under the optical microscope (Fig. 4b) with an average particle size of approximately 30 μm (Fig. 4c) and had a smooth surface, as revealed by scanning electron microscopy (Fig. 4d). Further observation by confocal microscopy revealed that the EVs were uniformly dispersed inside the microspheres (fluorescein isothiocyanate (FITC)-labeled microspheres and PKH26-labeled EVs). (Fig. 4e and Supplementary Movie 1). For verification of whether GelMA-PEG microspheres can release EVs in vitro, the fluorescence intensity of PKH26 red-labeled EVs (PKH26-EV) was quantitatively and qualitatively analyzed. As shown in Fig. 4f, the measurement of the fluorescence intensity of PKH26 in the supernatant showed that the release rate of microsphere-EVs was faster in the first 16 days with a

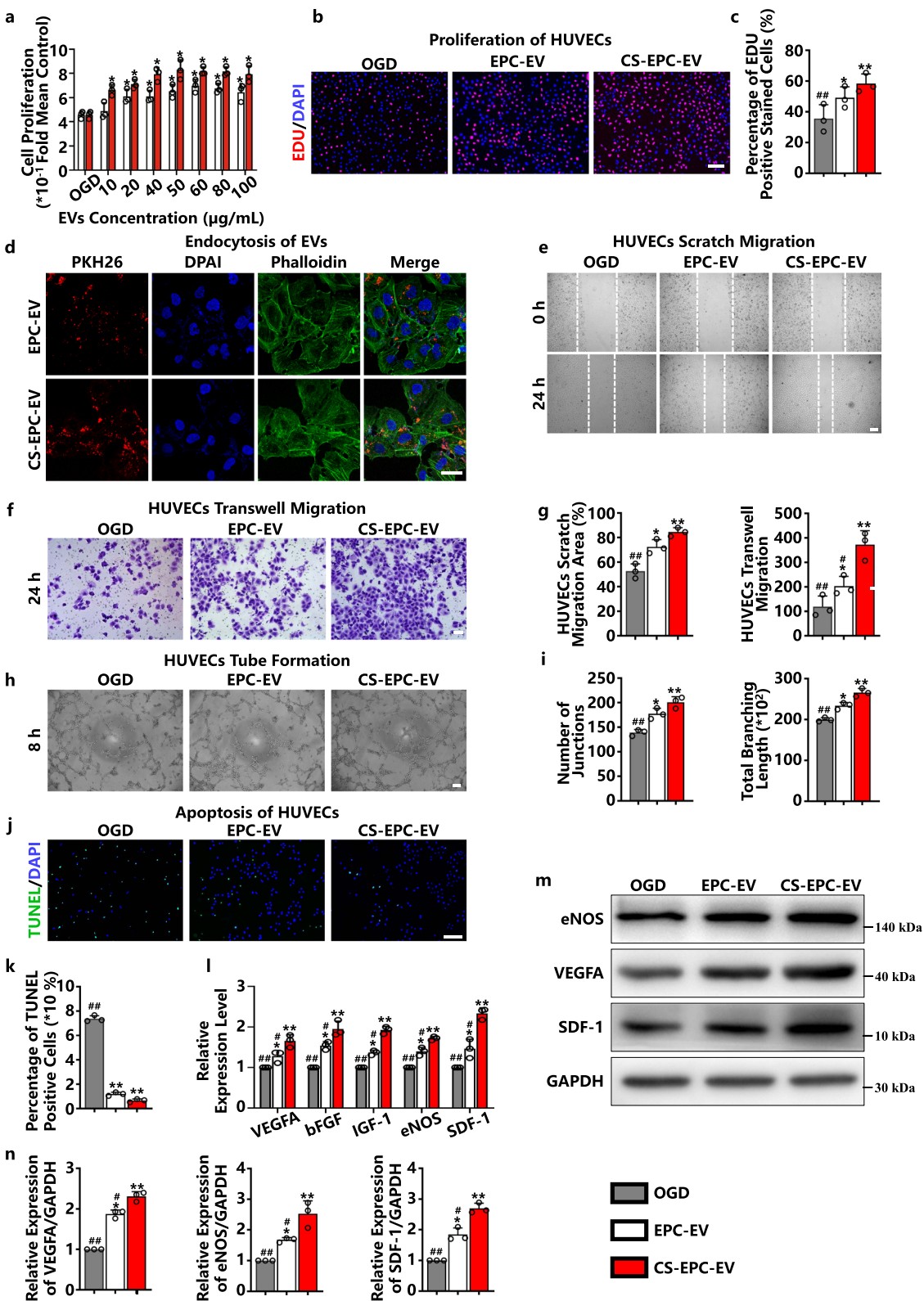

linear trend, especially on Day 2, when the release rate of extracellular vesicles was 18.6%. Then, the release rate started to slow down after 16 days (Day 16: 66.7%; Day 18: 70.7%). Finally, on Day 20, the release rate of microsphere-EVs reached 72.7%. Moreover, from the fluorescence intensity of the supernatant, it is clear that the EV release amount was the highest on the 2nd day (Supplementary Fig. 4), which was mainly due to the rapid release of EVs in the early stage, and the phenomenon that the fluorescence intensity gradually increased from the 4th day to the 16th day and began to decline after the 16th day (Supplementary Fig. 4) might be due to the degradation of Gelma microspheres[43]. Moreover, the fluorescence photos show that the fluorescence intensity of microsphere-EVs decreased with time, but weak red fluorescence was still visible on Day 21 (Fig. 4g). This result indicates that the prepared microspheres can continuously release EVs for at least 21 days, which is sufficient for the treatment of AMI.

**Fig. 3 | CS-EPC-EVs can enhance the angiogenic ability of human umbilical vein endothelial cells (HUVECs) under glucose and oxygen deprivation. a** The proliferation of HUVECs treated with different EVs was detected by MTS under normal culture conditions (*n* = 3, biological replicates per group). **b** 5-ethynyl-2'-deoxyuridine (EdU) proliferation experiments of HUVECs treated with different EVs under glucose and oxygen deprivation (OGD). **c** Quantitative analysis of the EdU proliferation assay (*n* = 3, biological replicates per group). **d** PKH-26-labeled EVs (red fluorescence) were cocultured with HUVECs for 12 h, and phalloidin (green fluorescence) and 4',6-diamidino-2-phenylindole (DAPI, blue fluorescence) were used to label the cytoskeleton and nucleus, respectively, (*n* = 3, biological replicates per group), scale bar = 25 μm. **e, f** Wound healing assays (**e**) (scale bar = 150 μm) and Transwell experiments (**f**) (scale bar = 100 μm) of HUVECs treated with different EVs under glucose and oxygen deprivation. **g** Quantitative analysis of the wound healing assay and Transwell assay (*n* = 3, biological replicates per group). **h** Tube formation experiments of HUVECs treated with different EVs under glucose and oxygen deprivation, scale bar = 150 μm. **i** Quantitative analysis of tube formation experiments (*n* = 3, biological replicates per group). **j** Terminal deoxynucleotidyl transferase dUTP nick end labeling (TUNEL) experiments of HUVECs treated with different EVs under glucose and oxygen deprivation, scale bar = 200 μm. **k** Quantitative analysis of the TUNEL assay. **l** qRT–PCR detection of VEGFA, bFGF, IGF-1, eNOS, and SDF-1 gene expression in HUVECs treated with different EVs under glucose and oxygen deprivation (*n* = 3, biological replicates per group). **m, n** The expression and quantitative analysis of the VEGFA, eNOS, and SDF-1 proteins in HUVECs treated with different EVs under glucose and oxygen deprivation (*n* = 3, biological replicates per group). *$p < 0.05$, **$p < 0.01$ vs. OGD; #$p < 0.05$, ##$p < 0.01$ vs. CS-EPC-EVs. Data are presented as the mean ± standard. Two-tailed Student's t-test was used to compare the differences between two groups. One-way ANOVA and post hoc Bonferroni tests were used to compare differences among more than two groups. Source data are provided as a Source Data file. Each experiment was repeated 3 times or more independently with similar results.

## Therapeutic effects of microsphere+CS-EPC-EVs on myocardial infarction

The microsphere+CS-EPC-EVs were injected into the infarct area of mice to evaluate the therapeutic function at a dose of 20 μg EVs based on the injection protocol reported in the literature (10–50 μg)[44,45]. However, importantly, given the difference between different EVs, 20 μg may not be the optimal dose for our EVs, and more studies are needed to determine the optimal doses in the future. To determine the retention rate of EVs in the myocardium, we used PKH26-labeled EVs to observe the fluorescence signals in cardiac slices at different time periods after injection. On the first day, red fluorescent signal-labeled EVs were uniformly dispersed in the infarcted area, the amount of red fluorescence decreased in both groups over time, and the fluorescence intensity in the microsphere-EV group was significantly higher than that in the EV group. On the 21st day, the fluorescent signal was no longer observed in the EV group, while it was still obvious in the microsphere-EV group (Fig. 5a). Quantitative analysis also confirmed that the fluorescence intensity decreased obviously from Day 0 to Day 21 in both the microsphere-EV group and the EV group. Moreover, at each time point, the fluorescence intensity was significantly higher in the microsphere-EV group than in the EV group (Supplementary Fig. 12). This result indicates that the prepared microspheres can indeed effectively deliver EVs in a sustained manner at the injection site. The cardiotoxicity of locally injected microspheres was evaluated by comparing the injection of phosphate-buffered saline (PBS) with microspheres without EVs. CD68 fluorescent staining was used to verify whether empty microspheres induced additional macrophage infiltration in myocardial tissue. The results showed that there was no significant difference in the number of macrophages between the microsphere group and the PBS group (Fig. 5b), indicating that the microspheres have good biocompatibility.

To evaluate the therapeutic function of CS-EPC-EVs (microsphere+CS-EPC-EVs), we first evaluated heart function evaluated by echocardiography examination at 0, 7, 14, and 21 days post-microsphere injection. When we compared the same group at different time points, the results showed that the cardiac function in the EV and EV-containing microsphere groups was significantly improved on Day 14 compared with that on Day 7 after myocardial infarction. Furthermore, with a prolonged time period up to 21 days, we found that cardiac function on Day 21 was only significantly improved in the EV-containing microsphere groups compared to that on Day 14, while no significant difference was found between the pure EV without microsphere treatment group. In contrast, the cardiac function in the PBS group declined with time up to 21 days (Fig. 5c; Supplementary Figs. 6 and 7). Interestingly, as shown by the specific values of left ventricular ejection fraction (EF), left ventricular fractional shortening (FS), left ventricular end-diastolic volume (LVVD) and left ventricular end-systolic volume (LVVS), microsphere+CS-EPC-EVs more significantly improved the cardiac systolic function of mice than microsphere+EPC-EVs (Fig. 5d and Supplementary Fig. 7), which indicates that CS-EPC-EVs have a higher therapeutic effect than EPC-EVs.

Further histological analysis (Masson staining, Fig. 5e) demonstrated that the microsphere+CS-EPC-EV group had significantly reduced infarct size and scar thickness compared with the PBS and microsphere+EPC-EV groups, revealing that CS-EPC-EVs could protect the infarcted heart from adverse remodeling (Fig. 5f). Sirius red staining further proved that these scars were fibrotic tissue (Supplementary Fig. 8a), and quantitative statistical analysis determined that microsphere+CS-EPC-EVs indeed had the best inhibitory effect on fibrotic scar tissue (Supplementary Fig. 8b, c). The microsphere+CS-EPC-EV group showed a significantly reduced fibrotic area and scar size compared with the microsphere+EPC-EV group, revealing that CS-EPC-EVs could more effectively protect the infarcted heart from undesirable remodeling than EPC-EVs.

Wheat germ agglutinin (WGA) staining was used to assess the degree of cell hypertrophy in the marginal zone of myocardial infarction. The results showed that the cross-sectional area of cardiomyocytes was markedly increased in the PBS group due to infarction. In contrast, the cross-sectional area of cardiomyocytes in the microsphere+EPC-EV and microsphere+CS-EPC-EV groups was significantly smaller, indicating a therapeutic effect of the EVs. More interestingly, the microsphere+CS-EPC-EV group showed a smaller cross-sectional area of cardiomyocytes than the microsphere+EPC-EV group (Fig. 5g). This result suggests that CS-EPC-EVs indeed have higher bioactivity than PEC-EVs and can effectively alleviate the pressure overload of cardiomyocytes in the infarct border zone. The apoptosis of cardiomyocytes after myocardial infarction is an important factor affecting the degree of ventricular remodeling after myocardial infarction. Therefore, to further verify the antiapoptotic effect of microsphere+CS-EPC-EVs in vivo, we performed TUNEL staining in the infarct margin area on Day 21. The results showed that the number of TUNEL-positive cells in the microsphere+EPC-EV and microsphere+CS-EPC-EV groups was significantly reduced in the border zone of myocardial infarction compared with that in the PBS group (Fig. 5h). Quantitative analysis further indicated that the antiapoptotic activity of microsphere+CS-EPC-EVs was significantly higher than that of microsphere+EPC-EVs (Fig. 5h). Then, caspase-3 staining was used to further evaluate the apoptotic cells. The results showed that the antiapoptotic effect of microsphere+CS-EPC-EVs was significantly higher than that of microsphere+EPC-EVs (Supplementary Fig. 16), which was consistent with the TUNEL results. Angiogenesis in vivo was assessed by α-SMA and CD31 staining (Fig. 5i). The results showed that compared with the PBS group, both the microsphere+EPC-EV and microsphere+CS-EPC-EV groups showed significantly promoted angiogenesis in the peripheral area of the myocardial infarction, and the microsphere+CS-EPC-EV group had significantly higher values than the microsphere+EPC-EV group, indicating higher bioactivity of CS-EPC-EVs in stimulating new blood vessel formation (Fig. 5j). As shown by the

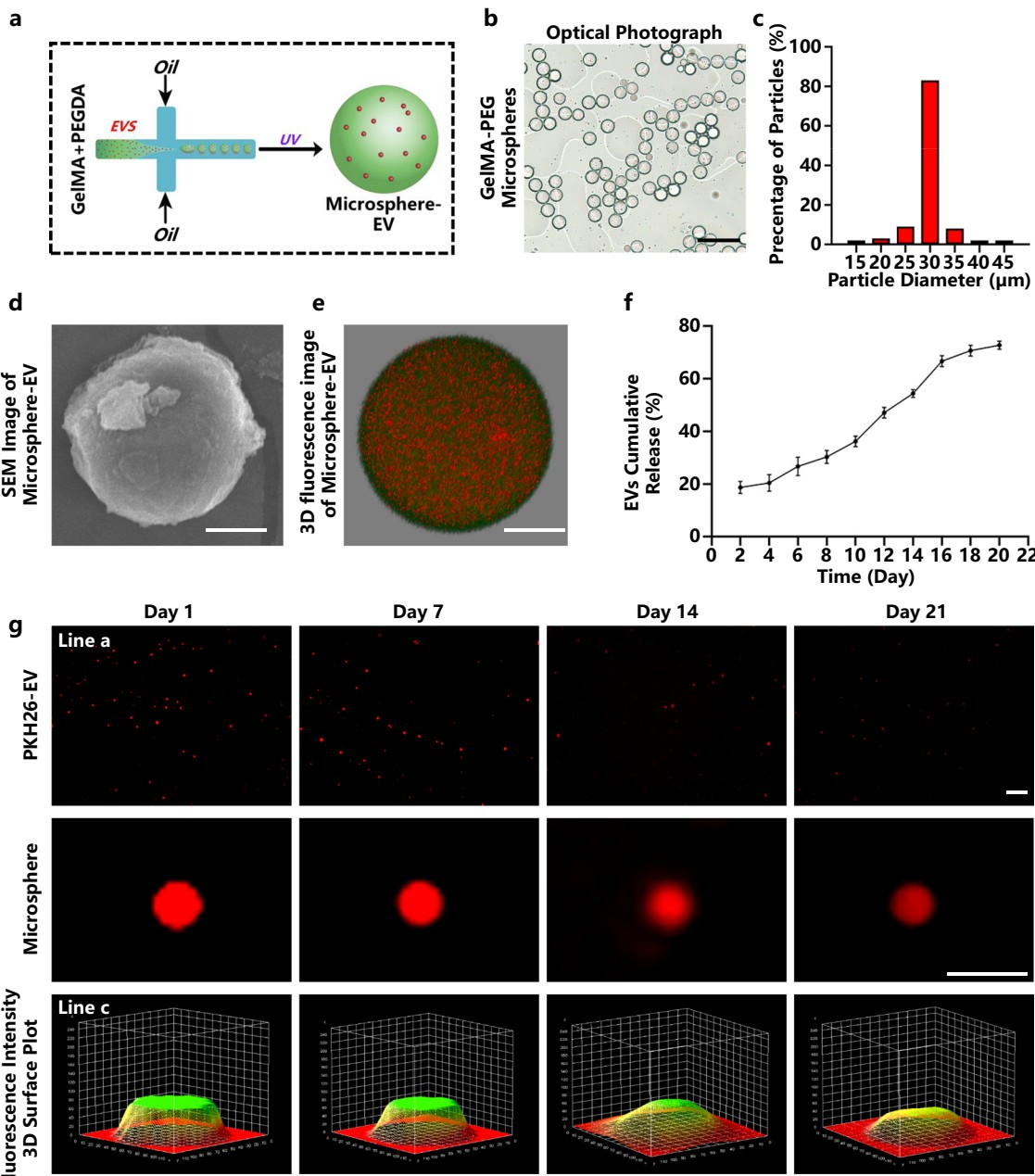

**Fig. 4 | Preparation and characterization of gelatin methacryloyl-polyethylene glycol (GelMA-PEG) microspheres. a** Schematic diagram of the preparation principle of GelMA-PEG microsphere-encapsulated EVs. This Figure was created with BioRender.com. **b** Optical photograph of GelMA-PEG microspheres. Scale bar = 100 μm. **c** Particle size analysis of the microspheres. **d** Scanning electron microscopy (SEM) image of GelMA-PEG microspheres. Scale bar = 10 μm. **e** Three-dimensional fluorescence image of EV-encapsulated microspheres taken by confocal microscopy. Microspheres (fluorescein isothiocyanate (FITC) labeled, green fluorescence), EVs (1,1′-dioctadecyl-3,3,3′,3′-tetramethylindocarbocyanine perchlorate-labeled (PKH26-labeled), red fluorescence), scale bar = 10 μm. **f** The release profile of extracellular vesicles from microspheres was detected by a spectrofluorometer ($n$ = 3, independent experiment replicates per group). **g** Fluorescence microscopy images of PKH26-EV-encapsulated microspheres after immersion in phosphate-buffered saline (PBS) for 1, 7, 14, and 21 days (line a, scale bar = 200 μm); magnified single microsphere images (line b, scale bar = 50 μm) and fluorescence intensity 3D surface plot (line c). Data are presented as the mean ± standard. Two-tailed Student's $t$-test was used to compare the differences between two groups. One-way ANOVA and post hoc Bonferroni tests were used to compare differences among more than two groups. Source data are provided as a Source Data file. Each experiment was repeated 3 times or more independently with similar results.

CD31/α-SMA colocalization staining images, microsphere+CS-EPC-EVs most significantly enhanced the regeneration of capillaries and arterioles (Supplementary Fig. 11). In addition, since the microspheres alone did not significantly promote angiogenesis, we further confirmed that CS-EPC-EVs were the key activator that induced angiogenesis (Supplementary Fig. 11). Moreover, by comparing microsphere+EPC-EV injection with the injection of pure EPC-EVs without microsphere encapsulation, we found that the microsphere-encapsulated EVs (microsphere+EPC-EVs) more effectively promoted the recovery of cardiac function (Supplementary Fig. 7), inhibited scarring and fibrosis (Supplementary Fig. 8), and enhanced angiogenesis (Supplementary Fig. 11), indicating that the microspheres could indeed improve the therapeutic effects of EPC-EVs possibly by sustained release of EVs for a longer period.

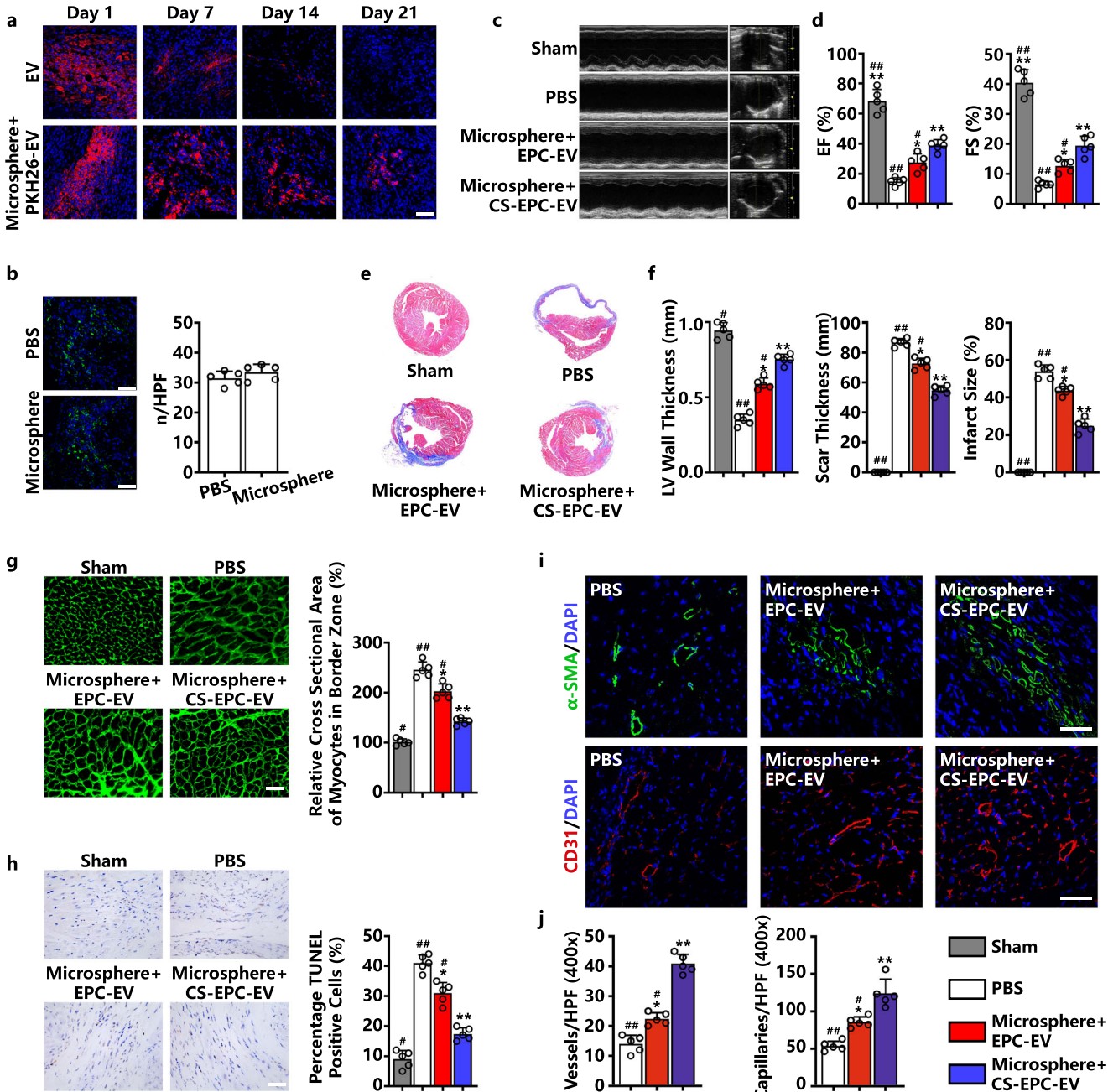

**Fig. 5 | Microsphere+CS-EPC-EVs can improve cardiac function after myocardial infarction, reduce cardiac remodeling, inhibit cardiomyocyte apoptosis, and increase angiogenesis (*n* = 5, biological replicates per group). a** Confocal fluorescence microscopy images of PKH26-EV-encapsulated microspheres in the area of mouse myocardial infarction at 1, 7, 14, and 21 days, Scale bar = 50 μm. PKH26-EV (red fluorescence), DAPI (blue fluorescence). **b** CD68 immuno-fluorescence staining of macrophages in cardiac tissue on Day 7 after myocardial infarction, Scale bar = 50 μm. CD68 (green fluorescence) and DAPI (blue fluores-cence). Quantitative analysis of the number of CD68-positive cells. **c** M-mode ultrasound images on Day 21 after myocardial infarction. **d** Quantification of the ejection fraction (EF) and fractional shortening (FS) of the animals on Day 21 after myocardial infarction. **e** Masson staining of hearts on Day 21 after myocardial infarction. **f** Statistical analysis of left ventricular wall thickness, scar thickness and infarct size. The number of hearts = 5. **g** Wheat germ agglutinin immuno-fluorescence staining of myocardial cells in the marginal zone of myocardial infarction on Day 21 after myocardial infarction, scale bar = 50 μm. Cross-sectional

area measurements of cardiomyocytes in the marginal zone of myocardial infarc-tion. **h** TUNEL staining in the infarct margin area on Day 21 after myocardial infarction, Scale bar = 100 μm. Total nuclei (DAPI staining, blue) and TUNEL-positive nuclei (tan). Quantitative analysis of TUNEL-positive cardiomyocytes. **i** Immunofluorescence staining pictures of small arteries in the peripheral area of myocardial infarction on Day 21 after myocardial infarction, scale bar = 50 μm. Alpha-smooth muscle actin (α-SMA, green fluorescence) and DAPI (blue fluores-cence). Immunofluorescence staining of capillaries in the peripheral area of myo-cardial infarction on Day 21 after infarction. CD31 (red fluorescence) and DAPI (blue fluorescence). **j** Quantitative analysis of immunofluorescence staining of arterioles and capillaries. \**p* < 0.05 and \*\**p* < 0.01 vs. the PBS group; #*p* < 0.05, ##*p* < 0.01 vs. microsphere+CS-EPC-EVs. Data are presented as the mean ± standard. Two-tailed Student's *t*-test was used to compare the differences between two groups. One-way ANOVA and post hoc Bonferroni tests were used to compare differences among more than two groups. Source data are provided as a Source Data file. Each experiment was repeated 3 times or more independently with similar results.

## CS-EPC-EVs promote EPC homing

Previous studies have shown that SDF-1 is a key factor for the migration of hematopoietic stem and progenitor cells between bone marrow and circulating blood and plays an important role in regulating the migration and homing of endothelial progenitor cells[26]. In the present study, we found that compared with EPC-EVs, CS-EPC-EVs not only contained more SDF-1 but also effectively activated HUVECs to express more SDF-1 under glucose and oxygen deprivation. Therefore, we speculate that CS-EPC-EVs can increase the concentration of SDF-1 in the myocardial infarction area, which results in the recruitment of EPCs from the blood circulation and enhances the homing of EPCs to the infarction area. To test this hypothesis, we first treated HUVECs with different EVs under glucose and oxygen deprivation conditions in vitro and collected the conditioned medium. EPCs were cultured with conditioned medium, and the ability of conditioned medium to recruit EPCs was observed by Transwell assays (Fig. 6a). The results showed that the conditioned medium produced by coincubating EPC-EVs or CS-EPC-EVs with HUVECs (EPC-EV+CM, CS-EV+CM) effectively

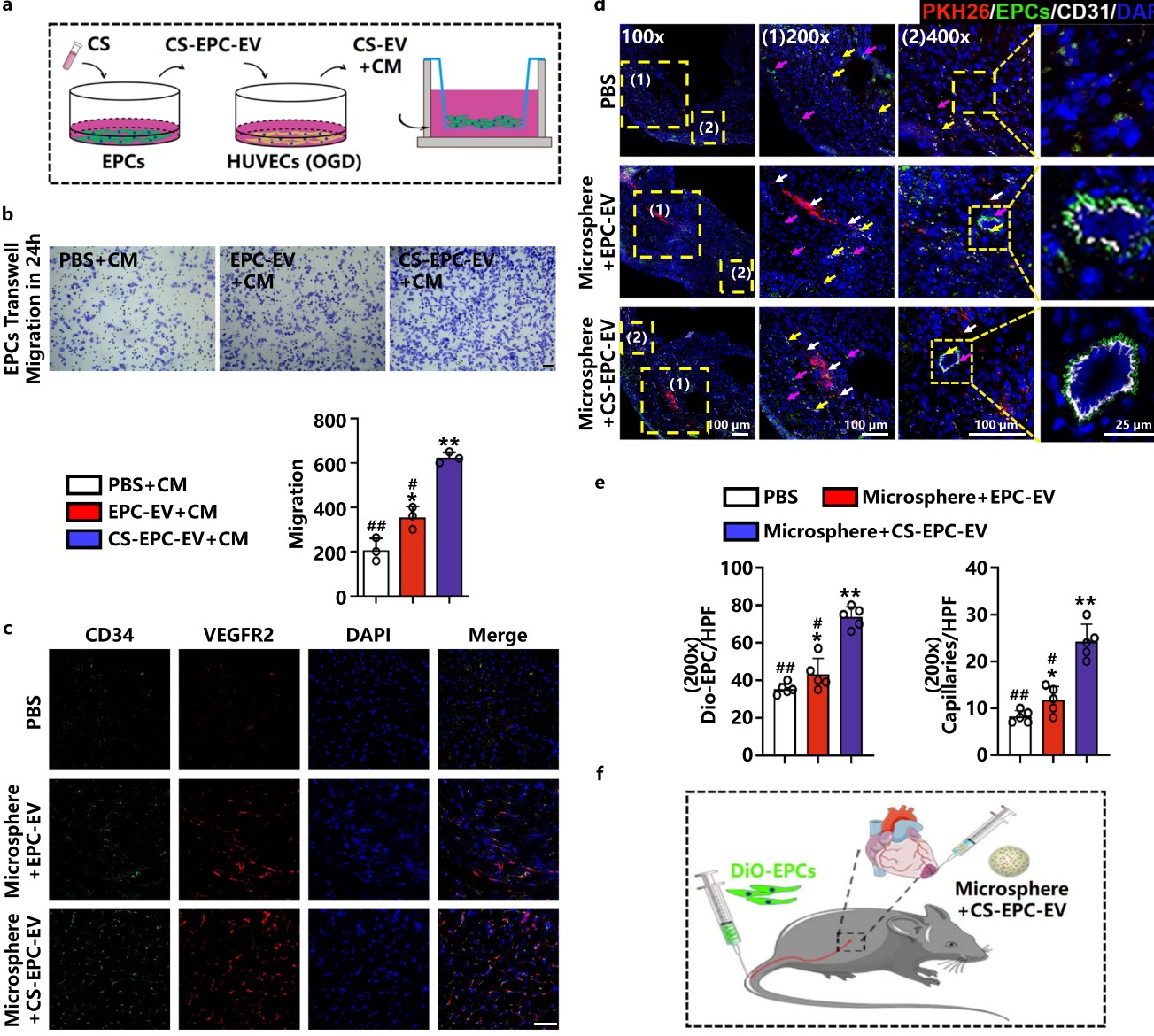

**Fig. 6 | CS-EPC-EVs promote EPCs migration in vitro and in vivo. a** Schematic of coincubation. This Figure was created with BioRender.com. **b** Transwell experiments and quantitative analysis of EPCs with different conditioned media under normoxia. *$p < 0.05$, **$p < 0.01$ vs. OGD; #$p < 0.05$, ##$p < 0.01$ vs. CS-EV+CM ($n = 3$, biological replicates per group), scale bar = 50 μm. **c** Cluster of Differentiation 34/ Vascular Endothelial Growth Factor Receptor 2 (CD34+/VEGFR2+) immunofluorescence staining of EPCs in the peripheral area of myocardial infarction on Day 7 after infarction. CD34+ (red), VEGFR2+ (green), DAPI (blue), CD34+/VEGFR2+ (orange), scale bar = 50 μm, specimens ($n = 5$, biological replicates per group). **d** Fluorescence images of 3,3'-Dioctadecyloxacarbocyanine Perchlorate-EPCs (DiO-EPCs) and CD31-positive capillaries in the peripheral area of myocardial infarction on Day 7 after infarction. (1) and (2) are images of the corresponding area zoomed in on the yellow dot box. PKH26-EV (red fluorescence, white arrow), DiO-EPC (green

fluorescence, purple arrow), CD31 (white fluorescence, yellow arrow), and DAPI (blue fluorescence) ($n = 5$, biological replicates per group). **e** Quantitative analysis of DiO-EPCs and the number of capillaries in the peripheral area of myocardial infarction (200 x magnification). *$p < 0.05$, **$p < 0.01$ vs. the PBS group; #$p < 0.05$, ##$p < 0.01$ vs. microsphere+CS-EPC-EVs ($n = 5$, biological replicates per group). **f** Schematic diagram of the DiO-EPCs injection and microsphere+CS-EPC-EV treatment of infarcted nude mice. This Figure was created with BioRender.com. Data are presented as the mean ± standard. Two-tailed Student's *t*-test was used to compare the differences between two groups. One-way ANOVA and post hoc Bonferroni tests were used to compare differences among more than two groups. Source data are provided as a Source Data file. Each experiment was repeated 3 times or more independently with similar results.

improved the migration of EPCs, while the CS-EV+CM group showed a higher migration rate than the EPC-EV+CM group (Fig. 6b). This result indicates that CS-EPC-EVs can substantially increase the migration of EPCs under glucose and oxygen deprivation and may enhance the homing of EPCs in vivo.

The in vivo experiment further confirmed the enhanced EPC recruitment, in which we used two EPC markers, CD34 and VEFGFR2, for colocalization staining to quantitatively analyze the number of EPCs in the peripheral area of the myocardial infarction. The results showed more CD34 and VEFGFR2 colabeled cells in both the microsphere+EPC-EV and microsphere+CS-EPC-EV groups than in the PBS group. In particular, significantly more colabeled cells were observed in the microsphere+CS-EPC-EV group than in the microsphere+EPC-EV group, indicating a stronger ability to recruit endogenous EPCs (Fig. 6c). Quantitative analysis also showed that microsphere+CS-EPC-EVs significantly promoted the expression of CD34 and VEFGFR2 compared with microsphere+EPC-EVs (Supplementary Fig. 15). To determine the role of EPC recruitment in the therapeutic effect of microsphere+CS-EPC-EVs, we labeled EPCs with DiO (DiO-EPCs) and injected them into myocardial infarcted nude mice treated with microsphere+CS-EPC-EVs. The correlation between EPC homing and vascular repair was demonstrated by CD31 staining. The results showed that compared with the PBS and microsphere+EPC-EV treatments, microsphere+CS-EPC-EV treatment indeed stimulated the homing of DiO-EPCs from the circulation in the peripheral area of myocardial infarction (Fig. 6d, e). In addition, after stimulation by microsphere+CS-EPC-EVs, the density of capillaries in the peripheral area of the myocardial infarction was significantly increased. Interestingly, some of the new blood vessels may be formed by homing EPCs, as the enrichment of DiO-EPCs was clearly observed in these vessels (Fig. 6d). From Supplementary Fig. 2, we found that 71.61 ± 2.84% of EPCs were EOCs, and 26.08 ± 2.68% were ECFCs. ECFCs may participate in neovascularization, while EOCs do not[8]. This result

indicates that a small number of ECFCs in the DiO-EPCs participated in the process of angiogenesis. Furthermore, as shown by the immuno-fluorescence staining images of CTNT, CD31 and Vimentin, CS-EPC-EVs were colocalized with cardiomyocytes, endothelial cells and fibro-blasts, although many large red spots were found in all the groups. In contrast, injection of PKH26 dye directly marked all the area of the sections (Supplementary Fig. 5a), which indicates that PKH26 dye could label all the cells on the section. According to these results, we speculated that the large red dots observed in the image may be the aggregation of many microsphere+CS-EPC-EVs (PKH26). Moreover, we counted the number of PKH26-EV colocalized cells in the high-magnification field (cardiomyocytes (CTAT and PKH26 colocalized): 19.6 ± 3.8; endothelial cells (CD31 and PKH26 colocalized): 6.6 ± 1.1; fibroblasts (vimentin and PKH26 colocalized): 8.8 ± 1.5), and the results showed that the cell uptake of the PKH26-EVs in cardiomyocytes was the highest compared to that of the other cells (Supplementary Fig. 5b). Our experiments confirmed that CS-EPC-EV treatment indeed enhanced the recruitment of EPCs in the ischemic area of myocardial infarction, which actively contributes to enhanced angiogenesis at the infarcted site.

## Effect of silicate ions on the expression profile of miRNA in EPC-derived EVs

To further explore the underlying mechanism of the therapeutic effect of CS-EPC-EVs, we analyzed the differential miRNA expression profiles of EPC-EVs and CS-EPC-EVs by high-throughput sequencing (EPCs were stimulated with diluted CS ion solution (1/128, silicate ions) or con-ventional culture medium for 48 h to obtain CS-EPC-EVs and EPC-EVs). The results showed that compared with EPC-EVs, CS-EPC-EVs upregu-lated 11 miRNAs and downregulated 42 miRNAs ($p < 0.05$) (Fig. 7a, b). Among these upregulated miRNAs, 5 highly differentially expressed miRNAs (miR-126a-3p, miR-486b-5p, miR-150-5p, miR-26a-5p and miR-142a-3p) were screened according to the fold change value ≥2 as the

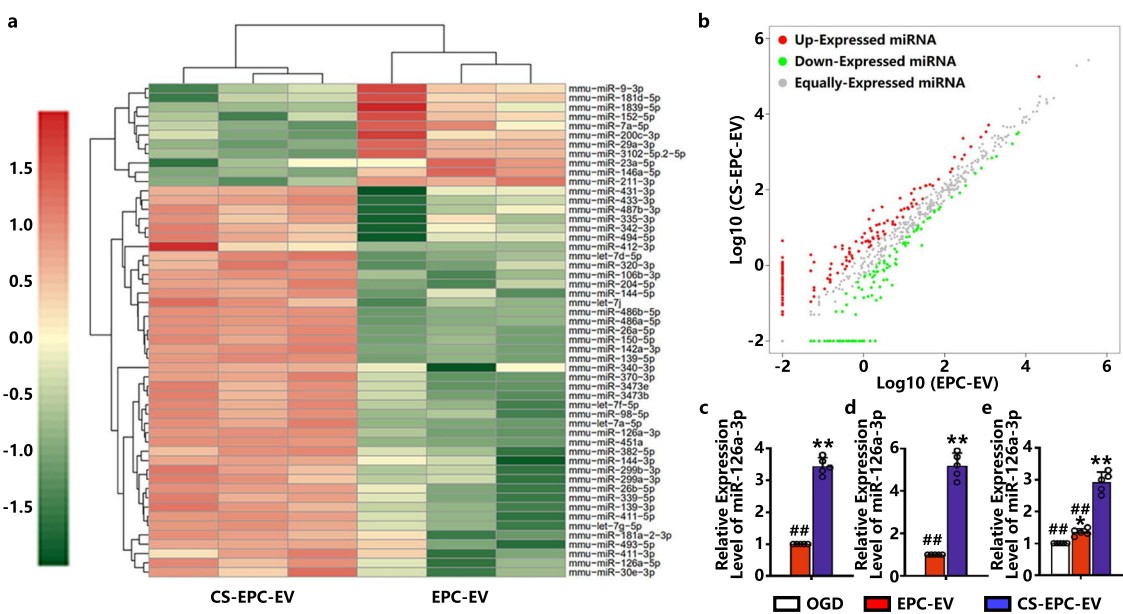

**Fig. 7 | Silicate ions from CS bioceramics promote the expression of angiogenesis-related factors and miRNAs in EPC-derived EVs. a** Heatmap of miRNA differential expression analysis between samples. Red represents upregu-lated genes, green represents downregulated genes, and color depth represents log10$^{(COUNT+1)}$ values. **b** Scatter plot of miRNA differential expression analysis between samples. Red dots indicate significantly upregulated miRNAs, and green dots indicate significantly downregulated miRNAs. **c** The expression of miR-126a-3p in the EVs was measured by qRT–PCR. **p < 0.01 vs. EPC-EVs, ##p < 0.01 vs. CS-EPC-EVs (n = 5, biological replicates per group). **d** qRT–PCR detection of miR-126a-3p

expression in EPCs after silicate ion treatments. *p < 0.05 and **p < 0.01 vs. EPC-EVs (n = 5, biological replicates per group). **e** The expression of miR-126a-3p in HUVECs treated with different EVs under glucose and oxygen deprivation (n = 5, biological replicates per group) **p < 0.01 vs. OGD, ##p < 0.01 vs. CS-EPC-EVs. Data are pre-sented as the mean ± standard. Two-tailed Student's t-test was used to compare the differences between two groups. One-way ANOVA and post hoc Bonferroni tests were used to compare differences among more than two groups. Source data are provided as a Source Data file. Each experiment was repeated 3 times or more independently with similar results.

criterion. To validate the results of miRNA sequencing, we further measured the expression of these 5 miRNAs in the two groups of EVs by qRT–PCR. The results showed that CS-EPC-EVs significantly upregulated these 5 genes compared with EPC-EVs, especially miR-126a-3p, which showed the highest upregulation (Supplementary Fig. 15 and Fig. 7c). Then, the expression of miR-126a-3p in EPCs was also detected, and the results showed that compared with that of the EPCs without any treatment, miR-126a-3p in the EPCs treated with silicate ions was significantly increased (Fig. 7d). Therefore, silicate ions are able to upregulate miR-126a-3p expression in EPCs and EPC-derived extracellular vesicles. Furthermore, we found that miR-126a-3p expression was also higher in the CS-EPC-EV-treated HUVECs than in the EPC-EV-treated HUVECs under glucose-oxygen deprivation (Fig. 7e). Studies have shown that overexpression of miR-126a-3p can increase the expression of VEGFA, bFGF, IGF-1, eNOS, and SDF-1 proteins in EPCs, which is consistent with our experimental results[46,47]. Furthermore, we used a miR-126-3p simulant with Cy5 fluorescence to transfect EPCs and collected the EVs secreted from these EPCs. Then, HUVECs were cultured with the obtained EVs. Interestingly, we found that EPCs can indeed transfer miR-126a-3p to HUVECs through EVs (Supplementary Fig. 14). Here, we hypothesized that CS-EPC-EVs promote angiogenesis mainly by transferring miR-126a-3p obtained from EPCs into HUVECs.

## CS-EPC-EVs promote the angiogenesis of HUVECs through the transfer of highly expressed miR-126a-3p

For verification of whether EVs mediate angiogenesis in HUVECs by transferring miR-126a-3p, EPCs were transfected with miR-126a-3p inhibitor (miR-126a-3p inhibitor NC was used as a negative control group) to silence the effect of miR-126a-3p and then cultured with silicate ions. Subsequently, qRT–PCR was used to verify the expression of miR-126a-3p in EPCs cultured with silicate ions (CS-EPCs) and their derived EVs. As shown in Fig. 8a, b, the expression of miR-126a-3p in the EPCs treated with miR-126a-3p inhibitor and cultured with silicate ions (CS-126IEPC) and EVs derived from miR-126a-3p inhibitor-transfected EPCs with silicate ion treatment (CS-126IEPC-EV) was significantly lower than that in the EPCs treated with inhibitor NC (CS-NCIEPC) and EVs derived from EPCs treated with inhibitor NC (CS-NCIEPC-EV), which demonstrated that CS-126IEPC-EVs could not stimulate miR-126a-3p expression after inhibitor silencing. Then, the effect of CS-126IEPC-EVs on the migration and vascular tube formation of HUVECs under glucose-oxygen deprivation was examined. Both wound healing and Transwell assays (Fig. 8c) showed that compared with CS-EPC-EVs and CS-NCIEPC-EVs, CS-126IEPC-EVs cannot promote the migration of HUVECs. Quantitative analysis showed that both the wound healing area and migration number (Fig. 8c) of HUVECs after CS-126IEPC-EV intervention were significantly lower than those of the CS-EPC-EV and CS-NCIEPC-EV groups. Similarly, CS-126IEPC-EVs failed to promote HUVEC formation of vascular tubes on Matrigel (Fig. 8d). The above results indicate that miR-126a-3p in CS-EPC-EVs was indeed the main bioactive molecule for its promotion of angiogenesis.

To further confirm the role of miR-126a-3p in CS-EPC-EVs in the recruitment of EPCs through activation of endothelial cells, we cultured HUVECs with CS-126IEPC-EVs, CS-NCIEPC-EVs and CS-EPC-EVs, and the respective conditioned media CS-NCIEV+CM, CS-NCIEV+CM, and CS-EV+CM were used to treat EPCs in a Transwell assay. The results showed that the mobility of the EPCs treated with CS-126IEV+CM was significantly decreased compared with that of the EPCs treated with CS-EV+CM and CS-NCIEV+CM (Fig. 8e). This finding further proves that miR-126a-3p is an important component in CS-EPC-EVs for promoting EPC homing.

Previous studies have shown that the SDF-1/CXCR4 signaling axis and its downstream PI3K/AKT/eNOS pathway play a key role in angiogenesis and recruiting circulating stem cells to injured sites. As an inhibitor of the SDF-1/CXCR4 signaling pathway, RGS16 is abundantly expressed in HUVECs and has been confirmed to be one of the downstream target genes of miR-126a-3p[48,49]. In this study, we observed that under glucose and oxygen deprivation, CS-EPC-EVs can stimulate HUVECs to express high levels of SDF-1 and eNOS proteins and mediate angiogenesis by transferring the expressed miR-126a-3p to HUVECs. Therefore, we speculated that CS-EPC-EVs may inhibit the expression of RGS16 in HUVECs and activate the CXCR4 receptor to facilitate miR-126a-3p transfer. The qRT–PCR results also showed that compared with those of CS-EPC-EVs and CS-NCIEPC-EVs, the levels of SDF-1 and eNOS in HUVECs cultured with CS+126IEPC-EVs were significantly decreased (Fig. 8f). Furthermore, under glucose and oxygen deprivation, CS-EPC-EVs indeed significantly inhibited the expression of RGS16 and stimulated the expression of CXCR4 in HUVECs compared with EPC-EVs and OGD (Fig. 8g). However, when miR-126a-3p was silenced, the RGS16 protein level was significantly increased and the protein levels of CXCR4, SDF-1, and p-AKT were significantly decreased in the CS-126IEPC-EV group compared to the CS-EPC-EV and CS-NCIEPC-EV groups (Fig. 8h). All these results demonstrated that under glucose and oxygen deprivation, miR-126a-3p carried by CS-EPC-EVs stimulated angiogenesis by activating the SDF-1/CXCR4 axis and its downstream PI3K/AKT/eNOS axis in HUVECs.

## The sorting of miR-126a-3p into EVs was mediated by hnRNPA2B1 and nSMase2

Previous studies have shown that cells can manipulate the content of miRNA in EVs through various sorting methods[50]. To further investigate the effect of silicate ions on the sorting and secretion mechanism of EPC-derived EVs and to explore the relationship between the content of miR-126a-3p and the sorting mechanism, we measured the mRNA expression level of 5 common sorting genes of EPCs cultured with silicate ions for 48 h. As shown in Fig. 9a, b, EPCs cultured with silicate ions showed significantly higher mRNA expression levels of hnRNPA2B1 and nSMase2 (SMPD3) than those cultured in normal medium. Previous studies have confirmed that hnRNPA2B1 and nSMase2 (SMPD3) are key regulators of RBP-dependent and ceramide-dependent pathways[50]. They are involved in EV biogenesis and secretion. However, no studies have confirmed whether hnRNPA2B1 and nSMase2 (SMPD3) mediate the sorting of miR-126a-3p. Therefore, in this study, we focused on exploring the interaction of these two proteins with miR-126a-3p. First, we knocked down hnRNPA2B1 and nSMase2 (SMPD3) expression in CS-EPCs with specific siRNAs, and Fig. 9c–e shows successful knockdown of both genes and proteins. Then, we examined the expression of miR-126a-3p in EPCs and derived EVs and found that knocking down the expression of hnRNPA2B1 and SMPD3 in EPCs significantly decreased the levels of miR-126a-3p in their derived EVs (Fig. 9f, g), while the levels of miR-126a-3p in cells were almost unchanged (Fig. 9h, i). This result indicates that miR-126a-3p is selectively loaded into EVs via hnRNPA2B1 and nSMase2 (SMPD3).

## Discussion

One of the challenges for clinical applications of EVs is the high-yield production of highly bioactive EVs. In this study, based on our previous finding of the bioactivity of silicate ions derived from silicate bioceramics for the regulation of stem cells[24], we proposed a material chemistry-based approach by using an ionic solution of calcium silicate bioceramics (CS) to regulate EV synthesis in EPCs. Considering the low solubility of most silicate compounds and materials, the advantage of this bioceramic-based approach to prepare silicate ion solutions is that the concentration of silicate ions can be well controlled while the effect of unwanted ions can be avoided. Our results demonstrate that CS bioceramic material-derived silicate ion solution effectively regulated EV secretion of EPCs, in which not only the amount of the synthesized EVs but also the bioactivity of the EVs was significantly increased. We show that these highly bioactive EVs more significantly promoted angiogenesis by stimulating endothelial cells and EPCs and provided more therapeutic benefits, including reduced cell apoptosis

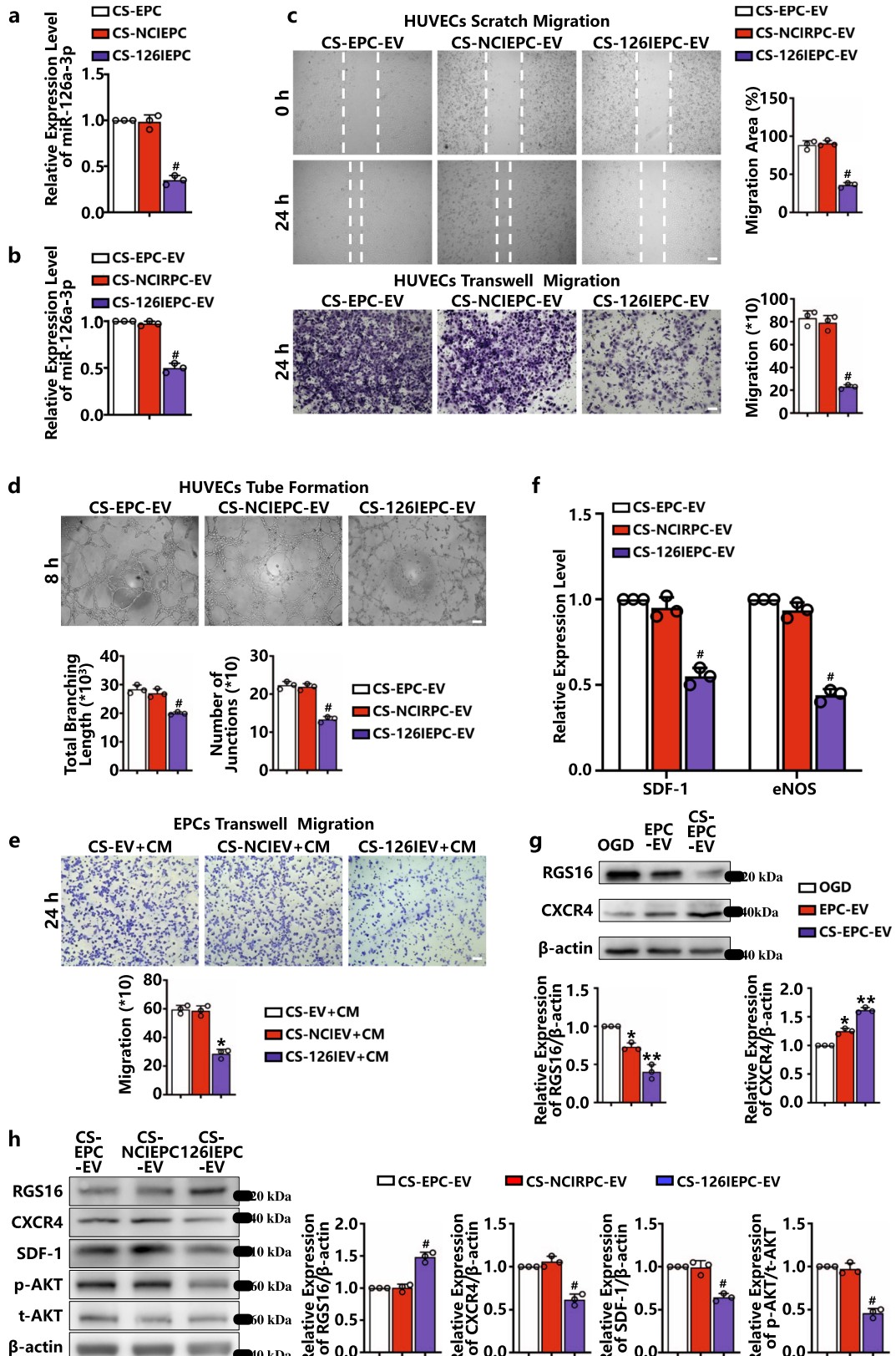

and heart fibrosis and recovery of heart function, than EVs obtained from normal cultured EPCs. Especially in promoting angiogenesis, the effect of CS-induced highly active extracellular vesicles is significantly higher than that of extracellular vesicles without CS stimulation. Notably, echocardiography was used to calculate ventricular volume in this study, and this technique assumes that the heart is spherical and then transforms the systolic and diastolic diameter measurements to volumes. In the infarct model, the heart is not spherical, and therefore, the volumes become less reliable. Thus, a pressure–volume catheter may be used to accurately measure ventricular volume in subsequent studies. Although more studies will be needed to elucidate the underlying mechanisms of silicate ion activation of cells for EV

**Fig. 8 | CS-EPC-EVs promote angiogenesis by transferring highly expressed miR-126a-3p to HUVECs** ($n = 3$, biological replicates per group). **a** qRT–PCR analysis of miR-126a-3p expression in CS-EPCs after miR-126a-3p inhibitor transfection. **b** qRT–PCR analysis of miR-126a-3p expression in EVs derived from miR-126a-3p inhibitor-transfected EPCs and verification of whether EVs mediate angiogenesis in HUVECs by transferring miR-126a-3p. EPCs were transfected with miR-126a-3p inhibitor (miR-126a-3p inhibitor NC was used as a negative control group) to silence the effect of miR-126a-3p and then cultured with silicate ions (CS +126IEPC-EV). **c**, **d** Qualitative and quantitative analysis of scratch migration experiments (scale bar = 150 μm), transwell migration experiments (scale bar = 100 μm) (**c**) and tube formation experiments (scale bar = 150 μm) (**d**) of HUVECs treated with different EVs under glucose and oxygen deprivation. **e** Qualitative and quantitative analysis of Transwell migration experiments of EPCs treated with different conditioned media under normoxic conditions, scale bar = 100 μm. **f** qRT– PCR analysis of the expression of SDF-1 and eNOS in HUVECs treated with different EVs under glucose and oxygen deprivation. **g** The expression and quantitative analysis of RGS16 and CXCR4 proteins in HUVECs treated with different EVs under glucose and oxygen deprivation. **h** The expression and quantitative analysis of Regulator of G protein Signaling 16 (RGS16), C-X-C Chemokine Receptor Type 4 (CXCR4), SDF-1, phosphorylated protein kinase B (p-AKT) and total protein kinase B (t-AKT) proteins in HUVECs treated with different EVs under glucose and oxygen deprivation. *$p < 0.05$ and **$p < 0.01$ vs. OGD, #$p < 0.05$ vs. the other two groups. Data are presented as the mean ± standard. Two-tailed Student's $t$-test was used to compare the differences between two groups. One-way ANOVA and post hoc Bonferroni tests were used to compare differences among more than two groups. Source data are provided as a Source Data file. Each experiment was repeated 3 times or more independently with similar results.

production, we believe that this bioceramic-based approach represents a promising advancement in EV engineering from EPCs for the treatment of ischemic diseases.

High-yield production of bioactive EVs is one of the key issues for successful cell-free EV therapies. Cells under conventional culture secrete limited amounts of EVs, which makes it difficult to meet the needs of clinical treatment. In addition, the bioactive components in EVs, such as proteins and RNA, have a decisive impact on the therapeutic effect. However, the increase in both the secretion amount and bioactivity of EVs is a challenge in EV preparation. In previous studies on the regulation of EV secretion, most approaches, such as physical conditions and drug treatments, either enhanced the secretion amount of EVs or increased the bioactivity of the obtained EVs but did not simultaneously stimulate both secretion and bioactivity. Studies have shown that various external stimuli (such as calcium induction, stress induction, chemical drug induction and growth factor intervention, etc.) can affect the EV secretion of cells[51]. Scheenen et al.[52] found that the exocytosis of cells is directly regulated by $Ca^{2+}$ and that optimal exocytosis can be maintained under stimulation with 80 ppm $Ca^{2+}$. Savina et al.[53] stimulated K562 cells with monensin (a kind of $Ca^{2+}$ carrier), which changed the $Ca^{2+}$ concentration inside cells and enhanced the secretion of EVs by increasing the exocytosis of cells. Shyong et al.[54] showed that calcium phosphate nanoparticles stimulated macrophages with increased EV secretion compared with untreated cells. However, although the stimulation of cells with $Ca^{2+}$ induction increased the yield of EVs, their biological activity was not increased compared to that of normal EVs. Stress induction by changing the culture environment of cells (such as temperature, oxygen, radiation and pH, etc.) may also affect EV secretion by placing cells under specific survival pressure. Although this method may increase the production of EVs in the short term, long-term production is questionable since abnormal conditions negatively affect cell growth for most types of cells[55]. Some chemical drugs such as palmitic acid, sitafloxacin, forskolin, SB218795, fenoterol, nitazole, and pentylenetetrazole may be used to stimulate EV secretion[56,57], but the EVs prepared by these chemical drug stimulations have proinflammatory and profibrotic properties, which are not suitable for tissue repair[57,58]. Growth factors such as HaR, TGF-β2, and PDGF have also been found to be able to regulate EV secretion[59–61], but the increase in the production yield is not high (approximately 1.2-fold)[62], and a specific growth factor such as PDGF activates cells to secrete EVs containing only higher amounts of this specific growth factor (PDGF), so the increase in EV bioactivity is limited[61]. CS is a unique biomaterial that has good anti-inflammatory and vascular regenerative effects[63]. Our previous studies have shown that ionic solutions derived from silicate bioceramics have bioactivity to regulate different cellular behaviors of many types of cells in tissue regeneration, such as fibroblasts, endothelial cells, cardiomyocytes and stem cells, and promote cell proliferation, migration, and gene expression[24,25,64], and the specific regulatory effect is dependent on the composition and concentration of the ionic

solutions. In our recent studies, we observed that lithium- and strontium-containing silicate bioceramics can improve vascularized bone regeneration by secreting bioactive EVs from bone marrow mesenchymal stem cells (BMSCs), but the activation mechanism of EVs is not clear[26,27]. Although the yield of EV production was not increased in these experiments, inspired by these findings, we hypothesized that bioceramic-derived ions may regulate both the secretion amount and composition of EVs if we select the right ceramics and ion concentrations. In this study, we demonstrated for the first time that the silicate ionic solution derived from CS bioceramics significantly activated EPCs to secrete large amounts of EVs (CS-EPC-EVs). Compared with that of conventionally cultured EPCs, the yield of EVs was increased by 1.74-fold without changing any other EV characteristics, including morphology, size distribution and EV-specific marker proteins. Furthermore, in addition to the increase in the yield of the EVs, the bioactivity of the obtained EVs was significantly higher than that of EVs derived from normal cultured EPCs without silicate ion treatment (EPC-EVs). The higher bioactivity of CS-EPC-EVs can be attributed to the significantly higher content of many different bioactive molecules, particularly angiogenesis-related molecules such as VEGFA, eNOS, SDF-1, IGF-1, and HGF, in CS-EPC-EVs than in EPC-EVs. This finding also explains the observed phenomenon of enhanced angiogenesis in mice with myocardial infarction treated with CS-EPC-EVs in our experiment. Interestingly, we found in our study that the silicate ionic solution promoted nSMase2 expression in EPCs. nSMase2 can promote ceramide formation by removing the phosphorylcholine moiety of sphingomyelin by hydrolysis, which can not only regulate the content of miRNA in EVs but also accelerate the formation of EV vesicles[65]. This finding explains the dual-promotion effect of silicate ions on both the secretion amount and bioactivity of CS-EPC-EVs in EPCs. However, more careful comparative studies will be required to further evaluate the applicability of CS-EPC-EV extracellular vesicles.

To explain why silicate ion stimulation results in higher bioactivity of the obtained EVs (CS-EPC-EVs), we analyzed the differential miRNA expression profiles of different EVs, which revealed a significant upregulation of miR-126a-3p, miR-486b-5p, miR-150a-5p, miR-26a-5p, and miR-142a-3p. In particular, the amount of miR-126a-3p is the highest among these miRNAs, and this molecule is known as a strong angiogenesis-promoting miRNA[66]. Therefore, we assumed that miR-126a-3p plays a key role in the therapeutic effects, and the higher bioactivity of CS-EPC-EVs in promoting angiogenesis of HUVECs may be mostly related to the upregulation of miR-126a-3p in CS-EPC-EVs. To further confirm the role of miR-126a-3p in CS-EPC-EVs, we silenced the expression of miR-126a-3p in EPCs and their EVs by transient transfection and found that the CS-EPC-EV-mediated angiogenesis and recruitment of EPCs were significantly reduced. The expression of angiogenesis-related genes such as eNOS and SDF-1 in EVs was also significantly decreased. This result suggests that miR-126a-3p plays an integral role in CS-EPC-EV-mediated angiogenesis. MiRNAs exert regulatory effects by repressing their target genes. As an inhibitor of the

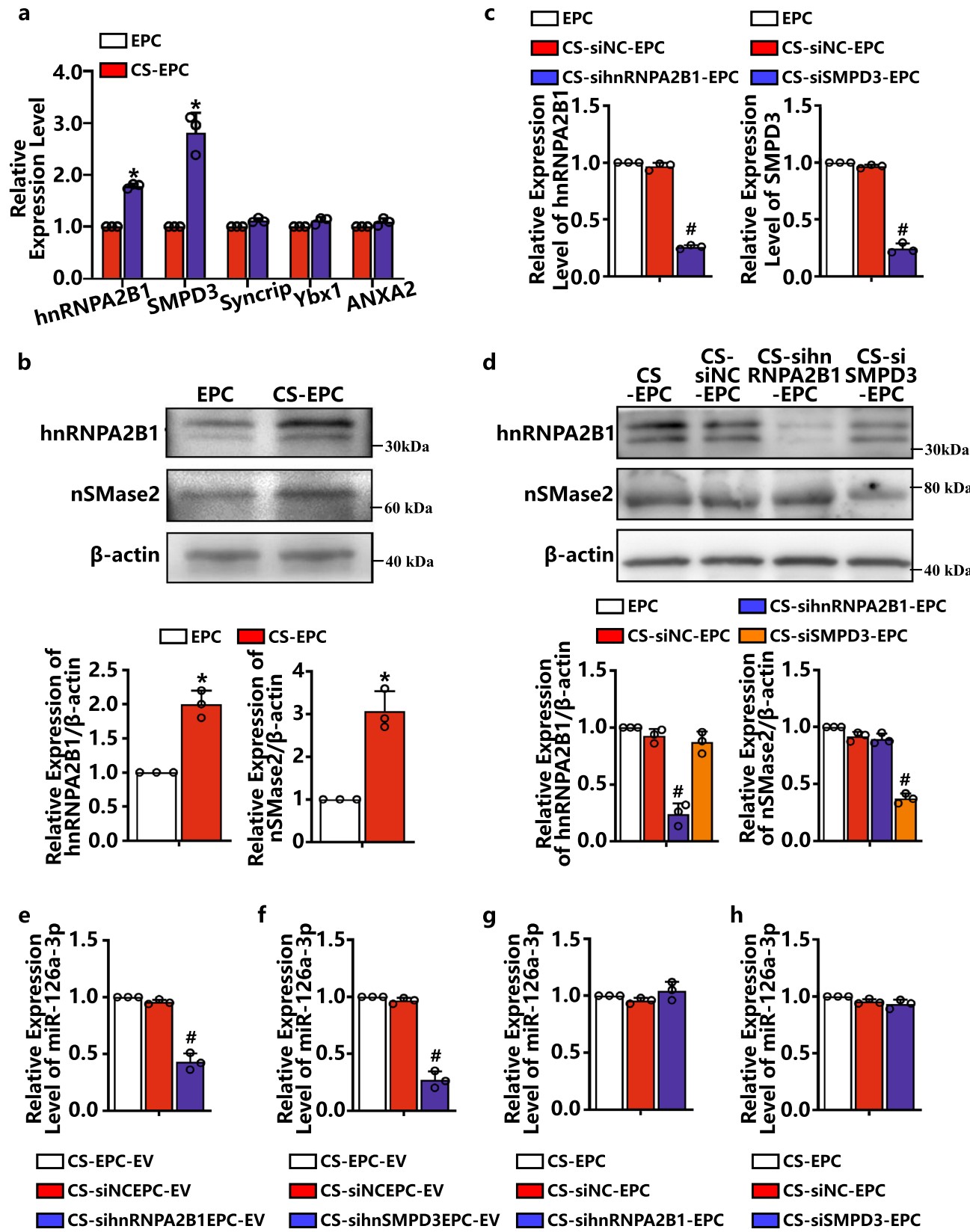

SDF-1/CXCR4 signaling pathway, RGS16 is abundantly expressed in HUVECs and has been confirmed to be one of the downstream target genes of miR-126a-3p[49]. Zernecke et al.[48] found that during atherosclerosis, as a self-protective mechanism of the human body, endothelial cells may deliver some miRNA-126-3p-enriched apoptotic bodies to recipient vascular cells to inhibit RGS16 expression and enhance CXCR4 signaling in recipient cells, which will significantly promote SDF-1 expression and recruit EPCs, thereby reducing atherosclerosis. Hiasa et al.[67] found that SDF-1 could enhance ischemia-induced neovascularization by activating AKT/eNOS-related pathways. Based on these previous findings, we believe that the SDF-1/CXCR4 axis and its downstream PI3K/AKT/eNOS signaling pathway may be

**Fig. 9 | miR-126a-3p was mediated by Heterogeneous Nuclear Ribonucleoprotein A2/B1 (hnRNPA2B1) and Neutral Sphingomyelinase 2 (nSMase2) to sort into EVs (** $n = 3$ **, biological replicates per group). a** The expression of hnRNPA2B1, Sphingomyelin Phosphodiesterase 3 (SMPD3), Syncrip, Y-box binding protein 1 (Ybx1), and Annexin A2 (ANXA2) in EPCs after culture with silicate ions (CS-EPCs) for 48 h. **b** Western blot analysis of hnRNPA2B1 and nSMase2 proteins in EPCs after culture with silicate ions for 48 h. *$p < 0.05$ vs. EPC. **c** The expression of hnRNPA2B1 and SMPD3 in CS-EPCs after knockdown with specific siRNA. **d** Protein analysis of hnRNPA2B1 and nSNase2 in CS-EPCs after knockdown with specific siRNA. **e, f** The

expression of miR-126a-3p in EVs after knockdown of hnRNPA2B1 (**e**) and SMPD3 (**f**). **g, h** The expression of miR-126a-3p in EPCs after knockdown of hnRNPA2B1 (**g**) and SMPD3 (**h**). #$p < 0.05$ vs. the other two groups. Data are presented as the mean ± standard. Two-tailed Student's $t$-test was used to compare the differences between two groups. One-way ANOVA and post hoc Bonferroni tests were used to compare differences among more than two groups. Source data are provided as a Source Data file. Each experiment was repeated 3 times or more independently with similar results.

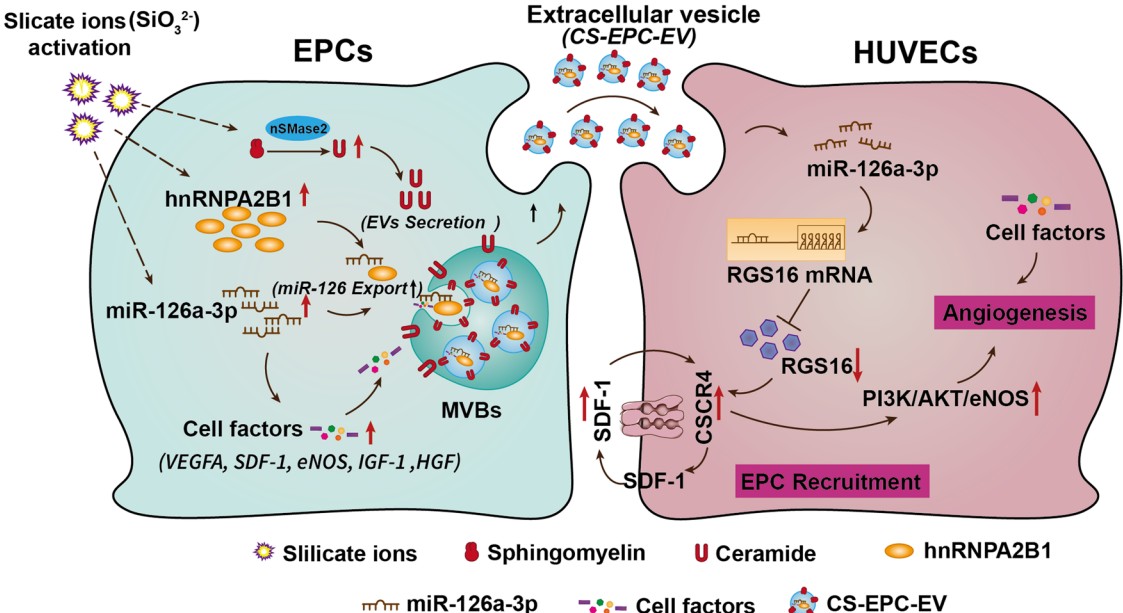

**Fig. 10 | Summary of the mechanism underlying the stimulatory effect of CS-EPC-EVs on the recruitment of EPCs and angiogenesis of HUVECs.** Silicate ions promote the expression of nSMase2, hnRNPA2B1 and miR-126a-3p in EPCs. The increased nSMase2 results in an increase in ceramide secretion in cells, thereby enhancing the production of extracellular vesicles. The higher hnRNPA2B1 and nSMase2 levels selectively enhanced the sorting of miR-126a-3p into multivesicular bodies (MVBs, sites of extracellular vesicle biogenesis). In addition, the high expression of miR-126a-3p promotes the expression of related angiogenic factors in EPCs, resulting in the increased content of related angiogenic factors in extracellular vesicles. After the highly bioactive extracellular vesicles with high contents of miR-126a-3p and angiogenic factors were transferred to recipient HUVECs, highly expressed miR-126a-3p could inhibit RGS16 and activate the SDF-1/CXCR4 axis and its downstream Phosphatidylinositol 3-kinase (PI3K)/AKT/eNOS axis of HUVECs, thereby mediating angiogenesis. Furthermore, a high content of angiogenic factors (such as VEGF) in extracellular vesicles contributes to the enhanced angiogenesis of HUVECs. This Figure was created with BioRender.com.

involved in miR-126a-3p-mediated angiogenesis by CS-EPC-EVs. Therefore, we further analyzed RGS16 expression. Our results showed that CS-EPC-EVs indeed significantly reduced the protein level of RGS16 in HUVECs and enhanced the activities of SDF-1, CXCR4 and eNOS compared with EPC-EVs. Consistently, after the expression of miR-126a-3p was silenced in EPC-derived EVs by transient transfection, the inhibitory effect of CS-EPC-derived EVs transfected with a miR-126a-3p inhibitor (CS-126IEPC-EVs) on RGS16 expression in HUVECs was significantly decreased, and the activation of the SDF-1/CXCR4 axis and its downstream PI3K/AKT/eNOS pathway was also significantly attenuated compared to those of CS-EPC-EVs. These data suggest that miR-126a-3p carried by CS-EPC-EVs can mediate the angiogenic process by activating the SDF-1/CXCR4 axis of endothelial cells and its downstream PI3K/AKT/eNOS axis.

Some studies have shown that the loading of miRNAs by cells does not occur randomly, and specific types of miRNAs may be preferentially allocated to EVs. One question to answer is how silicate ion-activated EPCs loaded more miR-126a-3p into EVs. In the process of miRNA sorting into EVs, hnRNPA2B1 and nSMase2 may play a major role. hnRNPA2B1 is a key mediator regulating miRNA loading into EVs and belongs to a class of RNA-binding proteins that control miRNA sorting into EVs in both a sequence-dependent and sequence-independent manner[68–70]. Moreover, nSMase2 is a key protein that

controls ceramide biosynthesis, which can regulate the content of miRNA in EVs and control the number of EVs[65,71]. To date, no study has confirmed whether hnRNPA2B1 and nSMase2 mediate the sorting of miR-126a-3p. Our data showed that silicate ions significantly upregulated hnRNPA2B1 and nSMase2 expression in EPCs. By knocking down hnRNPA2B1 and nSMase2 in EPCs, we found that the expression level of miR-126a-3p in EVs was significantly reduced, while the level of miR-126a-3p in cells was almost unchanged. This result confirmed that miR-126a-3p was selectively loaded into EVs through the interaction with hnRNPA2B1 and nSMase2, and silicate ions can significantly enhance this miRNA sorting. Interestingly, miR-126a-3p does not have a specific sequence that can be bound by hnRNPA2B1, which suggests that hnRNPA2B1 might be involved in the sorting of miR-126a-3p in a sequence-independent manner.

In addition to the activation of local endothelial cells in the myocardial infarction area, the recruitment of EPCs is critical for the effective repair of damaged myocardial tissue after myocardial infarction. Studies have shown that EPCs circulating in peripheral blood are recruited to the ischemic area within 72 h and form vascular clusters on Day 7 and vascular rings on Day 14 after myocardial infarction[72]. However, these self-recruited EPCs to the heart are far from sufficient for effective repair after myocardial infarction. Exogenous EPCs are injected intravenously as a therapeutic approach, but

only approximately 2% of EPCs can successfully reach the myocardial ischemia area, which is still not sufficient for revascularization in the myocardial infarction area[73]. The injection of growth factors such as SDF-1, VEGF, and PDGF directly into the site of myocardial infarction has also been applied, which did show the effect of EPC recruitment[74–76]. However, the half-life of these growth factors in vivo is very short, and a single injection into the myocardial infarction area cannot fully meet the therapeutic needs[74]. In our study, we found that microsphere+CS-EPC-EVs significantly increased the number of EPCs and capillary density in the peripheral area of myocardial infarction. More interestingly, some EPCs began to form vascular structures, indicating that CS-EPC-EVs not only activated the viability of endothelial cells in the infarction area but also recruited and promoted the homing and angiogenic function of EPCs, and ECFCs within the recruited EPC population may be directly involved in the formation of vascular structures. We speculate that one of the reasons for the formation of vascular structure by recruited EPCs on Day 7 after myocardial infarction may be related to the continuous release of SDF-1 and VEGFa from EVs, resulting in increased EPCs homing with increased differentiation[9,77]. In addition, although the expression of angiogenic factors by endothelial cells and recruited EPCs in the heart is rapidly upregulated after myocardial infarction[77], they may not be able to maintain higher concentrations in the tissue and drop significantly within 4–7 days[78]. Here, we used microsphere+CS-EPC-EVs to gradually release highly bioactive EVs, which ensures sustained delivery of sufficient active factors (SDF-1 and VEGFa) in the myocardial infarction area. After treatment for 21 days, the cardiac function and ventricular remodeling of the mice were significantly improved, while treatment using EPC-EVs alone only showed minor improvement compared to the PBS control (Supplementary Figs. 6–11), which confirms the effectiveness of microsphere+CS-EPC-EVs in the treatment of myocardial infarction and reflects the clinical value of encapsulation of extracellular vesicles in microspheres.

As summarized in Fig. 10, the possible underlying mechanisms of the activation of EPCs to secrete highly bioactive extracellular vesicles can be explained as follows. Silicate ions stimulated the expression of high levels of nSMase2, hnRNPA2B1 and miR-126a-3p, in which nSMase2 leads to increased ceramide secretion in cells and enhances the production of extracellular vesicles, and hnRNPA2B1 selectively enhances the sorting of miR-126a-3p into multivesicular bodies for extracellular vesicle biogenesis, while the enhanced expression of miR-126a-3p also promotes the expression of angiogenic factors in EPCs, resulting in an increased content of angiogenic factors in extracellular vesicles. When highly bioactive extracellular vesicles with high contents of miR-126a-3p and angiogenic factors were applied to the infarct site, they were transferred to recipient HUVECs, and miR-126a-3p-inhibited RGS16 and activated the SDF-1/CXCR4 axis and its downstream PI3K/AKT/eNOS axis in HUVECs to stimulate angiogenesis. Furthermore, a high content of angiogenic factors (such as VEGF) in extracellular vesicles contributes to enhanced angiogenesis in the infarct area. However, the activity of EVs in vitro is different from that in vivo, and more studies are needed to further explore the biological mechanism of CS-EPC-EVs in vivo.

In summary, we have demonstrated that silicate ions derived from CS bioceramics can effectively stimulate stem cells such as EPCs to secrete high-yield extracellular vesicles with high bioactivity, and the highly bioactive extracellular vesicles delivered to the infarct site through extracellular vesicle-loaded hydrogel microspheres can effectively promote recovery of the heart after myocardial infarction. Although more studies are required to further elucidate the mechanisms of bioceramic-derived silicate ions in the regulation of extracellular vesicle production in stem cells, we believe that silicate biomaterial-based extracellular vesicle engineering represents a new strategy to prepare highly bioactive extracellular vesicles for the treatment of heart infarction and other ischemic tissue/organ injuries.

## Methods

### Mice
To avoid the effect of the heredity of mice, such as breed, age, weight and sex, on the experimental results and considering the surgical injury for the mice in the myocardial infarction model and the higher tolerance of male mice, we selected male C57BL/6 mice and BALB/c nude mice at 8 weeks of age (average weight is 20 g) for this experiment. C57BL/6 mice and BALB/c nude mice were purchased from the Laboratory Animal Center of Southern Medical University. This experimental protocol was approved by the Animal Care and Use Committee of Southern Medical University (LAEC-2021-076) and complied with the Guide for the Care and Use of Laboratory Animals published by the National Institutes of Health (8th edition, 2011). The mice were grouped and housed in a controlled environment with a 12-h light-dark cycle, ambient temperature ranging from 18 to 22 °C, and 50–70% humidity. They were provided with unrestricted access to food and water. In this study, all experimental animals were euthanized by the injection of an overdose of anesthetic (1% pentobarbital sodium) at the conclusion of the experiment.

### Cell culture
Bone marrow-derived endothelial progenitor cells (EPCs) were isolated from C57BL/6 mice. In brief, adult mice (average weight 20 g) were anesthetized with 0.01 mg/g of pentobarbital administered via intraperitoneal injection. Upon the cessation of pedal reflex, euthanasia was performed by cutting the carotid artery, followed by induction of pneumothorax to confirm death. The long bones were then collected and the bone marrow was extracted. Bone marrow mononuclear cells were isolated using density-gradient centrifugation ($300 \times g$, 10 min) with Histopaque-1083 (Sigma-Aldrich) and plated on 10-cm culture dishes coated with fibronectin. EPCs were cultured as an adherent fraction in EGM-2 media (Lonza, Basel, Switzerland) without heparin or hydrocortisone. After 4 days, the media were changed to eliminate non-adherent cells and all EPCs were used 6 to 7 days after isolation. Then, EPCs were characterized by APC-A: CD133, PE-A: CD34 and FITC-A: VEGFR2 immunofluorescence staining and flow cytometry (Beckman)[34,35]. Furthermore, the subgroups of EPCs (EOCs or ECFCs) were characterized by flow cytometry (Beckman) analysis with triple fluorescent markers three times (APC-A: VEGFR2; PE-A: CD133; FITC-A: CD34) ($n = 3$). The percentages of the subgroups of EPCs were calculated as follows. The percentage of EOCs (CD133+/CD34+/VEGFR2+) = The percentage of VEGFR2+ * The percentage of CD133+/CD34+; The percentage of ECFCs (CD133-/CD34+/VEGFR2+) = The percentage of VEGFR2+(N) * The percentage of CD133-/CD34+; The percentage of ECFCs (CD133+/CD34−/VEGFR2+) = The percentage of VEGFR2+ * The percentage of CD133+/CD34−.

Human umbilical vein endothelial cells (HUVECs, iCellbioscience, HUM-iCell-e005) were cultured in ECM (Sciencell, USA) with 10% FBS, 1% endothelial cell growth supplement, and 1% penicillin/streptomycin. All cells were cultured at 37 °C in a humidified incubator with 5% $CO_2$.

### Preparation of silicate ion solutions
Calcium silicate (CS) powders (400 mesh) were purchased from Kunshan Huaqiao Technology New Materials Co., Ltd. A series of gradient-diluted silicate ion solutions were prepared. Briefly, 1 g of CS powder was soaked in 5 mL of serum-free EGM-2 and shaken in a humidified 37 °C/5% $CO_2$ incubator for 24 h. The supernatant was collected from the mixture, sterilized through a filter (Millipore, 0.22 mm) and stored at 4 °C. The stock solution was diluted with serum-free EGM-2 to prepare the gradient-diluted silicate ion solutions (1/2, 1/4, 1/8, 1/16, 1/32, 1/64, 1/128 and 1/256 dilutions), and all the solutions were stored at 4 °C for further experiments. The concentrations of Ca, Si, and P in the diluted silicate ion solutions were analyzed by ICP–AES (Optima 3000DV, PerkinElmer, USA).

## Isolation, characterization, and internalization of EVs

EPCs were stimulated with diluted silicate ion solution in culture medium (1/128) or conventional culture medium for 48 h, and EVs released from EPCs in the supernatant were isolated by ultra-centrifugation (CS-EPC-EV: stimulated with silicate ion solution; EPC-EV: stimulated with conventional culture medium). The specific extraction steps are as follows. The cell culture medium of EPCs was first centrifuged for 10 min at $300 \times g$ to remove the live cells. Then, the supernatant was centrifuged for 10 min at $2000 \times g$ to remove the dead cells followed by centrifugation for 30 min at $10,000 \times g$ to remove cellular debris. Finally, the supernatant was centrifuged for 70 min at $100,000 \times g$ to obtain EVs. PBS was used to wash EVs. The EVs obtained from silicate ion-stimulated EPCs were named CS-EPC-EVs, and the EVs obtained from EPCs cultured in conventional medium were named EPC-EVs. In addition, EVs labeled with PKH26 red fluorescence were also prepared for subsequent fluorescence imaging experiments. The morphology of EVs was characterized by transmission electron microscopy (TEM; Hitachi H800, Japan). The size, particle concentration, and video frames of EVs were analyzed by nanoparticle tracking analysis (NTA). CS ion solution was used as the control group. The total protein mass of EVs was analyzed by a BCA Protein Quantitative Kit (Thermo Fisher). The purity of EVs was calculated using the following equation: purity (particles/μg) = particles of EVs/total protein mass of EVs. EV markers such as ALIX(Abcam, ab275377), CD81(Abcam, ab109201), CD63(Abcam, ab217345) were detected by Western blot analysis. HUVECs were incubated with PKH26-labeled EVs for 12 h for the internalization assay.

## Cell proliferation

The effect of silicate ion solution on the proliferation of EPCs and the effect of EVs on the proliferation of HUVECs were examined by MTS (3-(4,5-dimethylthiazol-2-yl)-5-(3-carboxymethoxyphenyl)-2-(4-sulfophenyl)-2H-tetrazolium) assays (Dojindo, Japan). First, EPCs or HUVECs were seeded onto 96-well plates at a density of $3 \times 10^3$ cells/well. Then, cells were cultured with different concentrations of silicate ion solutions or different EVs. After 2 days of culture, different concentrations of silicate ion solutions or different EVs were replaced with an equal amount of medium containing 10% MTS, and the culture was continued for 3 h. Finally, the absorbance at 490 nm of the samples was measured using a Varioskan LUX Multimode Microplate Reader (Thermo Fisher, USA) to assess the proliferative capacity of the cells.

## Establishment of an oxygen-glucose deprivation (OGD) model

HUVECs were cultured in glucose-free EGM-2 medium (Gibco, USA) for 8 h at 37 °C in a hypoxic incubator (1% $O_2$, 5% $CO_2$) to induce ischemic injury.

## Wound healing and transwell migration of cells in vitro

For evaluation of the ability of EVs to promote cell migration, HUVECs were seeded in 6-well culture plates. After the cells reached more than 85% confluence, the monolayer of HUVECs was scraped using a 200 μL micropipette tip. Then, 3 mL of ECM containing different types of EVs was added to each well, and the cells were incubated for an additional 24 h. Cell images at 0 h and 24 h were taken using an inverted phase microscope (Leica, Japan). The migration rate (%) of cells was calculated as follows: Migration rate (%) = (A0 − An)/A0 × 100%. A0 represents the initial wound area, and An represents the final wound area.

Transwell (8.0 μm pore size, Corning Company) tests were used to assess the number of cells that migrated. HUVECs or EPCs in 200 μL of serum-free medium were seeded into the upper chamber. Complete medium (400 μL) with different types of EVs or PBS was added to the lower chamber. After 12 h, pictures of the five regions were taken with an inverted microscope (Leica, Japan). The number of migrated cells was measured by ImageJ (Version 1.5.3).

## Vascular tube formation of HUVECs

The formation of vascular tubes in vitro was performed by using Matrigel™ basement membrane matrix (356234, Corning, USA). First, Matrigel™ was used to coat plates (96 wells). Then, HUVECs ($3 \times 10^4$ per well) were incubated with EVs for 8 h. Five random pictures were taken using an inverted light microscope (Leica DMI 3000B, Germany), and the number of vascular tubes formed was counted with ImageJ (Version 1.5.3).

## Quantitative real-time PCR, western blot, EdU assays, apoptosis analysis, and enzyme-linked immunosorbent assays (ELISAs) of cells, and EVs

First, total RNA from EPCs, HUVECs, and EVs was extracted using RNAsio (TaKaRa, Japan). In particular, equal amounts of EPCs, HUVECs, and EVs were used to extract RNA. The concentration of total RNA was determined by a NanoDrop 1000 reader (Thermo Scientific). Then, cDNA was synthesized by Prime Script RT Master Mix (TaKaRa, Japan). All cDNAs of stemness marker genes were quantified using the Bio-Rad MyiQ Monochrome Real-Time PCR System. All experiments were performed in triplicate to obtain average data. The primer sequences are shown in Supplementary Table 2. Uncropped and unprocessed scans of Western Blot have been provided in "Data Availability".

Proteins in EPCs, HUVECs and EVs were extracted using RIPA solution (Sigma-Aldrich, USA) containing protease inhibitors and phosphatase inhibitors, and specific concentrations of proteins were measured by a BCA protein assay kit (Thermo Fisher, USA). Proteins were then separated using SDS–PAGE, transferred to PVDF membranes (Millipore, MA) and blocked with PBST containing 5% nonfat milk for 2 h at room temperature. Then, the PVDF membrane was incubated with primary antibodies for 12 h at 4 °C. The specific types of primary antibodies were as follows: eNOS (Abcam, 1:5000, ab300071), VEGFA (Abcam, 1:2000, ab214424), SDF-1 (Abcam, 1:2000, ab25117), RGS16 (Abcam, 1:2000, ab119424), CXCR4 (Abcam, 1:2000, ab181020), AKT (Cell Signaling Technology, 1:2000, 9272), p-AKT (Cell Signaling Technology, 1:2000,4060), GAPDH (Bioworld, 1:5000, ab8245), and β-actin (Abcam, 1:1000, ab8226). The PVDF membrane was then incubated with HRP-conjugated IgG (1:5000) for an additional 2 h at room temperature, and the labeled protein was visualized by ECL chemiluminescence reagent. In this study, the grayscale of immunoblots was measured by ImageJ (Version 1.5.3), and the relative expression of GAPDH or β-actin served as a loading control.

HUVECs were EdU-stained using an EdU Cell Proliferation Kit (Roche Applied Science, Germany) as directed by the manufacturer's instructions. Five pictures were randomly taken using an inverted optical microscope, and the number of EdU-positive cells and the total number of cells were counted.

TUNEL staining was performed on HUVECs and paraffin-embedded sections of mouse hearts by an in situ cell death detection kit (Roche Applied Science, Germany) following the manufacturer's instructions[24]. Five random pictures were taken by inverted light microscopy, and the number of TUNEL-positive cells and the total number of cells were counted.

The amounts of VEGF, eNOS, SDF-1, IGF-1, and HGF present in the different kinds of EVs were quantified by ELISA kits (Ray Biotec, China) according to the manufacturer's instructions.

Furthermore, the cytokine content in each EV particle was calculated according to the mass of cytokines (VEGF, eNOS, HGF, IGF, and SDF) measured by ELISAs and the total number of particles measured by NTA. Calculation formula: $C_{EVs} = M_{cyto}/N_{EVs}$, where $C_{EVs}$ is the mass of cytokine in one EV particle, $M_{cyto}$ is the mass of cytokines, and $N_{EVs}$ is the total number of particles.

## MicroRNA analysis

EPC-derived EVs were isolated as described above, the known microRNAs (miRNAs) in the two groups of EVs were identified, and their

expression patterns in different samples were analyzed by RiboBio (Guangzhou RiboBio Co., Ltd., China). Differentially expressed (DE) miRNAs were identified by a p value cutoff of 0.05 and a fold value change of ≥2 as filters.

## Function of miRNA-126a-3p

In this study, EPCs were treated with a miRNA inhibitor (Guangzhou RiboBio Co., Ltd., China) to obtain miRNA-126a-3p-inhibited EVs. Wound healing assays, transwell assays and tube formation experiments were used to verify the efficacy of miRNA-126a-3p in mediating angiogenesis in HUVECs. The function of hnRNPA2B1 (1:2000, Abcam, ab183654) and SMPD3 (1:2000, Abcam, ab85017) in the sorting of miRNA-126a-3p was studied with small interfering RNA (siRNA, Guangzhou RiboBio Co., Ltd., China). The sorting of miRNA-126a-3p in EVs treated with siRNA was assessed by qRT–PCR assay and Western blots. Uncropped and unprocessed scans of Western Blot have been provided in "Data Availability".

Furthermore, miR-126-3p simulant with Cy5 fluorescence was used to transfect EPCs, and EVs from these EPCs were collected to culture HUVECs for 24 h. The transfer of miR-126a-3p into HUVECs was observed by microscopy (Leica DMI 3000B, Germany).

## Preparation and characterization of microsphere-EVs

First, 5% PEG, 5% GelMA, and 0.2% lap (photoinitiator) solution were mixed in proportions, and EVs were added to the mixed solution (concentration of EVs in mixed solution: 1 mg/mL). Then, a water phase tube and an oil phase tube were connected to a microfluidic pressure pump. The prepared mixture solution was added dropwise to mineral oil by the microfluidic pressure pump and cured by crosslinking with UV light. The size and speed of microsphere formation were observed under a single lens reflex camera (Nikon F3), and the flow rates of the water phase and oil phase solution were adjusted accordingly. Next, some microspheres were placed in cell culture plates, and the particle size, morphology and dispersion of the microspheres were observed under a light microscope (Leica, Germany). The flow rate was further adjusted until the diameter of the microspheres was 30 μm. Finally, the mixture of oil and microspheres was centrifuged at 1500 rpm for 5 min. The mineral oil in the upper layer was carefully aspirated, and the microspheres were resuspended by adding an appropriate amount of PBS ($V_{Microsphere}$:$V_{PBS}$ = 2:1). The morphology of the microspheres was observed using optical microscopy (Leica, Japan) and scanning electron microscopy (SEM, SU-8010; Hitachi, Japan). Furthermore, PKH26 was used to label EVs, and fluorescein isothiocyanate (FITC) was used to label microspheres. The distribution of labeled EVs inside the microsphere was observed using a confocal laser scanning microscope (Leica TCS SP8, Germany). Five pictures were taken for each sample, and a representative picture was selected. The EVs containing microspheres were soaked in PBS solution, and the release of EVs was observed by fluorescence confocal microscopy on Days 1, 7 and 14. In addition, the PBS-containing microspheres were placed in a 37 °C incubator, 200 μL of supernatant was collected and resupplemented with 200 μL of PBS every two days until Day 20, and the fluorescence intensity of PKH26 in the supernatant was measured by fluorescence spectrophotometry to quantitatively analyze the release of EVs.

## Establishment of a mouse myocardial infarction model and treatment with EV-containing microspheres

For generation of an AMI model, C57BL/6 mice underwent permanent left anterior descending (LAD) ligation. First, mice were anesthetized with tribromoethanol (1.2%, 0.02 mL/g) and ventilated with a rodent ventilator (DW-3000B; Xinsida, Beijing, China) at 100 breaths per minute with a stroke volume of 100 mL. Then, thoracotomy was performed on the left side of the mouse and ligated with 8-0-prolene suture at the LAD. The survival rate of the animals was 80% throughout the experiment. In addition, the sham group underwent the same surgical procedure, except that the LAD was not ligated. Echocardiography, evaluated by a small animal cardiac ultrasound instrument (Vevo® 2100 system (Fujifilm Visual Sonics, Canada)), was performed immediately after LAD ligation (mice with ejection fractions (EFs) of approximately 50% and fractional shortening (FS) of approximately 25% were selected in this study to ensure that the initial myocardial infarction degree of each mouse was consistent (Supplementary Fig. 6), and animals were randomized to four groups primarily based on echocardiography (PBS group, EPC-EV, microsphere+EPC-EV group, and microsphere+CS-EPC-EV group; all groups were dispersed with PBS). There were 5 animals in each group. Furthermore, microspheres alone were used as a control group to observe the effect of promoting angiogenesis. For the microsphere+EPC-EV group and microsphere+CS-EPC-EV group, 20 μL of microsphere+EPC-EV or microsphere+CS-EPC-EV suspension containing 20 μg of EVs (1 mg/mL EVs) was injected immediately via 30 G needles at two symmetric sites in the infarct area located at the cardiac base after MI. The total number of EPC-EV and CS-EPC-EV particles in each injection was $(46.0 \pm 3.6) * 10^9$ and $(47.3 \pm 1.8) * 10^9$, respectively (Supplementary Table 4). The total amount of growth factor was more than $5 * 10^{-3}$ μg, which is known to be effective for the treatment of myocardial infarction (Supplementary Table 6).

## Release of microsphere-EVs in vivo

For evaluation of the degradation behavior of microsphere-EVs in mouse hearts in vivo, microspheres encapsulating PKH26-labeled EVs were injected into the infarcted myocardial border zone. EVs alone were used as a control group. Tissue samples were processed for cryosectioning, and EV retention in the heart at Days 1, 7, 14, and 21 after microsphere injection was visualized by fluorescence microscopy. The release of EVs in vivo was quantified according to the fluorescence intensity change.

## Assessment of heart function

On Days 0, 7, 14, and 21, cardiac function was measured by the Vevo® 2100 system (Fujifilm Visual Sonics, Canada). Cardiac ejection fraction (EF), fractional shortening (FS), left ventricular end-diastolic volume (LVVD) and left ventricular end-systolic volume (LVVS) were assessed according to M-mode and B-mode echocardiography.

## Histological staining and analysis

Mice with myocardial infarction were sacrificed after 21 days of microsphere-EV treatment. Heart tissues from different groups were fixed using 4% paraformaldehyde, embedded in paraffin and sliced (thickness = 4 μm), and then subjected to Masson staining (Sigma-Aldrich, USA), Sirius red staining (Sigma-Aldrich, USA), and Alexa Fluor 488 and wheat germ agglutinin staining (Thermo Fisher, USA). Then, ImageJ software (Version 1.5.3) was used to evaluate the parameters of left ventricular remodeling (LV wall thickness, relative scar thickness, and infarct size) from Masson (Masson positive-staining area represents scar) and Sirius red staining images (Sirius red-positive areas represent fibrosis)[79,80], and the lateral area of cardiomyocytes from Alexa Fluor 488 and wheat germ agglutinin staining images.

Furthermore, TUNEL (Sigma-Aldrich, USA) and caspase-3 (1:500, Abcam, ab184787) immunohistochemical staining were used to detect the apoptosis of cells in the heart. Quantitative analysis of apoptotic cells was calculated according to the TUNEL-positive cells and caspase-3-positive cells.

## Immunofluorescence staining

First, tissue paraffin sections were washed 3 times with PBS and blocked with 3% BSA for 40 min. Then, they were incubated with primary antibody solution overnight at 4 °C. The following primary antibodies were used: CD31 (1:500, Servicebio, GB13063), α-SMA (1:500, Abcam, ab124964), CD34 (1:500, Abcam, ab8158), VEGFR2 (1:500,

Abcam, ab214424), and CD68 (1:500, CST, 29176). TUNEL staining was performed with an in situ cell death detection kit (Sigma–Aldrich, St. Louis, MO, USA). Tissue sections were rinsed five times in PBS, titrated with the appropriate secondary antibody (1:800, Abcam) and incubated for 1 h at room temperature. Then, tissue paraffin sections were washed 5 times with PBS and stained with DAPI for 5 min. Finally, images were captured by a Leica fluorescence microscope and analyzed by the Leica Application Suite X.

### Evaluation of the internalization of EVs by cells in vivo
Male C57BL/6 mice were used to evaluate the internalization of EVs in vivo. First, 20 μL of microsphere+CS-EPC-EVs (PKH26) (1 mg/mL EVs) was injected immediately after LAD ligation via 30G needles at two symmetric sites in the infarct area located at the cardiac base. Mice were killed on Day 3, and the heart paraffin sections were stained with immunofluorescence staining (cardiomyocytes (CTNT, 1:500, Abcam, ab209813), endothelial cells (CD31, 1:500, Servicebio, GB13063) and fibroblasts (Vimentin, 1:500, Abcam, ab92547)). PKH26 dye injected directly was used as a control group. Five random pictures (high-magnification field, 60X) were taken by confocal laser scanning microscopy (Leica TCS SP8, Germany). The number of PKH26-EV-colocalized cells was counted, and 5 slides were counted in each group.

### EPC recruitment assay
First, myocardial infarction models were created in nude mice following the process for establishing myocardial infarction models of normal mice as described above, and different EV preparations were injected into the myocardium. Then, DiO staining of EPCs was performed using a Cell Plasma Membrane Staining Kit (Beyotime, China), and 200 μL of the DiO-stained cell suspension ($10^6$ EPCs) was injected into nude mice through the tail vein. On Day 7, frozen sections of cardiac tissue from different groups were prepared for immunofluorescence staining (DAPI, CD31). Five random pictures were taken using a fluorescence microscope, and the number of positive cells (DiO-EPCs) was counted.

### Statistical analysis
All the results were assessed blindly by three people in this study. The number of tissue specimens to analyze in each group was 5. The data in this study are expressed as the mean ± standard deviation. Two-tailed Student's $t$-test was used to compare the differences between two groups. One-way ANOVA and post hoc Bonferroni tests were used to compare differences among more than two groups. Two-way time-varying ANOVA and t tests were used to compare the time-related differences between groups. Significant differences were considered when $p < 0.05$.

### Statistics and reproducibility
Each experiment was repeated 3 times or more independently with similar results.

### Reporting summary
Further information on research design is available in the Nature Portfolio Reporting Summary linked to this article.

## Data availability
All data needed to evaluate the conclusions in the paper are present in the paper and/or the Supplementary Information. Source data are provided with this paper. Additional data have been uploaded to the science data bank (https://www.scidb.cn/s/Rjamqm) in its original form. The data includes statistical graphs related to cell viability, cell migration, cell ring formation, quantitative analysis of animal experiments, Western Blot bands, and Western Blot parallel samples. The sequencing results have been deposited to the NCBI SRA public database under the accession number: PRJNA944988, and CS-EPC-EV

SRA accession number: SRX19681860; SRX19681859; SRX19681858, and EPC-EV SRA accession number: SRX19681857; SRX19681856; SRX19681855. Source data are provided with this paper.

## Code availability
No custom computer code or custom algorithm was used in this study.

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

## Acknowledgements

This work was supported by the Strategic Priority Research Program of the Chinese Academy of Sciences (Grant No. XDA16010203 to J.C.), the National Natural Science Foundation of China (Grant Nos. 81871504 and 82172103 to C.O., 82102226 to H.L.), the Guangdong Basic and Applied Basic Research Foundation (Grant No. 2020A1515110728 to H.L.), the China Postdoctoral Science Foundation (Grant No. 2021M691477 to H.L.), and the Guangzhou Basic and Applied Basic Research Foundation (Grant No. 2023A04J2447 to B.Y.).

## Author contributions

B.Y., H.L., and Z.Z. contributed equally to this work. B.Y., H.L., Z.Z., P.C., L.W., X.F., X.N., F.Z., X.H., J.C., and C.O. designed the research; B.Y., H.L., Z.Z., P.C., L.W., X.F., X.N., Y.P., F.Z., and X.H. performed the research; B.Y., H.L., Z.Z., J.C., and C.O. analyzed the data; and B.Y., H.L., Z.Z., J.C., and C.O. wrote the paper.

## Competing interests

The Shanghai Institute of Ceramics Chinese Academy of Sciences has applied for two patents related to silicate bioceramic ion solutions for the treatment of myocardial infarction, in which one has been granted and another is pending. J.C. and C.O. are the inventors of these two patents. The remaining authors declare no other competing interests.
