## [Peer Review File · Nature Communications]

Extracellular vesicles engineering by silicates-activated endothelial progenitor cells for myocardial infarction treatment in male miceREVIEWER COMMENTS

Reviewer #1 (Remarks to the Author):

These studies demonstrated a biomaterial-based approach to prepare larger amount of exosomes with more contents from endothelial progenitor cells (EPCs) by bioactive ceramics stimulation. The exosomes were encapsulated in hydrogel microspheres, which effectively mitigate myocardial infarction in an animal model. The authors further demonstrated that the engineered exosomes enhanced vascular repair by increasing the contents of miR-126-3p and angiogenic factors such as VEGF, SDF-1, CXCR4 and eNOS, which activate endothelial cells and recruit EPCs. The authors did great work to generate engineered exosomes and show the therapeutic effects of the EPC exosomes. There are some concerns need to be addressed.

Specific Comments:

1. Using exosomes as a therapeutic agent is very promising, however the field has concerns that the contents of miRNAs or proteins in each exosome are very low, which may not be enough to induce robust effects. Did the authors determine and calculate copies of miR-126-3p and concentrations of angiogenic factors such as VEGF, SDF-1, CXCR4 and eNOS in each exosome? miRNAs levels were usually determined by PCR, where the miRNAs were amplified. Another approach should be used to obtain quantitative data on miRNA levels. Once miRNA and protein contents in each exosome were calculated, the authors can roughly calculate how many exosomes were injected in vivo and convince the readers that enough cargos were delivered in vivo.
2. I would suggest the authors show each individual data point in all the figures.
3. Figure 1B-F should be labeled with each angiogenic factors.
4. For in vivo studies such as the figure 5D, I expected to see the injected exosomes were taken up by endothelial cells. Please clarify or show additional data that the injected exosomes were taken up by ECs.
5. The authors did great work to identify hnRNPA1 and nSMase2 expression were increased in CS stimulated EPC and they are responsible to sort miR-126-3p into exosomes. It is still not clear increase of hnRNPA1 and nSMase2 selectively increase miR-126-3p or few miRNAs or generally increase all the miRNA contents. If several miRNAs or many miRNAs were all increased in EPC exosome, then why only miR-126-3p exert robust effects?
6. In results section for figure 8, line 489 should be Figure 8C, D, line 492 should be Figure 8E, F, line 493 should be Figure 8G,H. They were mislabeled.
7. Why only male mice were used for in vivo studies. This needs appropriate justification.
8. In the method section, 20ul of 1mg/ml microsphere+EPC-Exo were injected. Please clarify how many exosomes were injected. This is important for readers to understand what contents were delivered in vivo and exerted the protective effects.
9. Microsphere+EPC-EXO was injected immediately after injury, please consider post treatment as therapeutic strategy.

Reviewer #2 (Remarks to the Author):

In this interesting study, the authors have used calcium silicate ions to induce endothelial progenitor cells to secrete exosomes (CS-EPC-EXO) which enhanced the proliferation of EPCs in comparison to normal media and promoted the expression of angiogenesis-related genes. Silicate ions also improved the yield of EXOs in EPC culture which were found to express key angiogenic factors as well as SDF-1 which is involved in the recruitment of hematopoietic stem cells. In vitro, these exosomes significantly enhanced the vascular tube-formation of HUVECs and prevented their apoptosis under conditions of oxygen and glucose deprivation. Next, the authors used a microfluidics-based technology to encapsulate these exosomes in microspheres made of methacrylated PEG which, in vitro, was able to continuously release exosomes for at least 21 days. In an acute myocardial infarction model in mice, it was then shown that these

microspheres prolonged the residency time of the encapsulated exosomes, improved left ventricular function, reduced fibrosis, scar size, cardiomyocyte hypertrophy and apoptosis and promoted angiogenesis in the peripheral area of myocardial infarction, in part through an enhanced recruitment of EPCs. The transcriptomic analysis of the CS-EPC-Exo showed a differential regulation of genes compared with the EPC-Exo, with a strong upregulation of miR-126a-3p whose enrichment in exosomes was triggered by the silicate ion-induced higher mRNA expression levels of hnRNPA2B1 and nSMase2 (SMPD3).

This paper addresses a clinically relevant topic. The experiments have been carefully designed and executed and the results look straightforward. Some issues, though, need to be addressed:

1- Isolation of exosomes: exosomes released from EPCs cultured in the silicate ion solution were isolated by ultracentrifugation. No details are provided regarding the method which was used precisely (speed, number, duration). The medium was prepared by diluting silicate ions in a serum-free Endothelial Cell Growth Medium-2. This medium, however, is likely to contain growth factors which may "contaminate" the EXO yield and consequently bias the results. Thus, NTA data should definitely include a true control showing that the silicate ion-EGF mix was free from particles that could have been misinterpreted by NTA as exosomes.

2- Experimental groups : The authors primarily document the superiority of CS-CPC-Exo over CPC-Exo but in both cases, exosomes were encapsulated in microspheres. They should have also included a group with the exclusive injection of exosomes (without microspheres) to better assess the added value of encapsulation which, in the perspective of a clinical translation, would increase the complexity of manufacturing. Likewise, a true control group should have consisted of the virgin medium (i.e., the silicate ion-EGF medium alone) instead of PBS to eliminate any potential effect of the medium on the outcomes. In addition to ejection fraction values, left ventricular volumes should be reported. Otherwise, it is incorrect to conclude that the treatment prevented adverse remodeling. Finally, the number of animals in each group is missing.

3- Scar size was assessed by Masson staining, which is appropriate. However, it is not correct to mention a reduction of fibrosis (likely interstitial fibrosis) without performing Sirius Red stainings. These stainings should be performed or the mention of fibrosis has to be deleted.

4- It is unclear whether outcomes were assessed blindly or not.

5- Internalization of exosomes: The authors report that exosomes mediated angiogenesis in HUVECs by transferring miR-126a-3p. On the other hand, they used PKH26-labeled EXOs to monitor the fluorescence signals in cardiac slices at different time periods after injection. In which cell type were exosomes internalized? This identification should be made possible by co-staining exosomes with cell-specific (cardiomyocyte, fibroblast, endothelial cell) markers.

6- Exosomes harbor few copies of miRNAs. In their experiments, the authors show different expression levels of miR-126a-3p and conclude to its predominant role on the basis of indirect findings (migration and vascular tube formation of HUVECs under glucose-oxygen deprivation). However, they do not provide direct evidence for a transfer of this miRNA from the exosomes to the recipient cells and the increased levels of miR-126a-3p in these cells might well reflect an increased endogenous synthesis triggered by one of the components of the exosome cargo. In the absence of a direct visualisation of the transfer of miR-126a-3p into HUVEC, the authors cannot definitely conclude to this transfer. Thus, they should either provide direct evidence for this transfer or reformulate the paper accordingly. In line with this statement, the discussion highlights the role of miR126a-3 but it should also be mentioned that other components of the EXO cargo (miRNAs, proteins) are likely involved in the cardiac protection observed in these experiments.

Reviewer #3 (Remarks to the Author):

Review of:

High efficiency exosomes engineering by activation of endothelial progenitor cells using silicate bioceramics for myocardial infarction treatment

Introduction:

Yu and colleagues describe an extensive evaluation of an innovative intervention to induce angiogenesis after coronary ligation in a mouse model. A variety of angiogenic therapies for a myocardial infarction have been reported in both animal models as well as in humans in the last 20 years. The effects of protein, gene and cell therapies have been controversial and no angiogenic therapy is currently recommended for patients after a myocardial infarction. The authors suggest that their new, complex intervention may have advantages over other approaches. However, a more extensive comparison with other approaches may be necessary to confirm their suggestion. Previous studies have demonstrated that the injection of EPCs and other types of stem cells into the peri-infarct region increased new blood vessel formation and improved heart function in both animal models and in some clinical trials. EXOs from a variety of cell sources have also been employed to increase angiogenesis and restore function after coronary occlusion in animal models (Eur Heart J 2017;38:201–211, J Cardiovasc Transl Res 2019;12:5–17, Cardiovasc Res 2021;117:292–307 for example). Although the authors are correct that the injection of EXOs into the infarcted heart may have limited retention, these studies demonstrated that they are effective in inducing angiogenesis and preserving ventricular function. The authors are requested to compare their microsphere+EPC-EXO studies with studies which employed cells or cell derived EXOs to put their studies into the context of the current literature.

Although the authors declare no competing interests, do they have a patent on the use of silicate bioceramics ionic solutions?

The authors employed bone marrow derived EPCs for their studies but did not further characterize these cells or indicate the spectrum of the cell types in their preparation. How many CD34 or CD133 cells are contained in their mixture? Previous studies have demonstrated that the isolation techniques employed greatly influence the angiogenic potential of the final mixture. The authors suggest that they followed the technique described in their reference #63. However, that excellent study from Philadelphia employed rats rather than mice. That study also isolated extracellular vesicles from EPCs (which were well characterized) and then incorporated into a shear thinning gel and injected into the infarct region. In each of their figures those authors depicted the results of each animal or experiment (which Yu et al should consider) and they carefully evaluated serial ventricular function with a pressure volume catheter.

The concept that exosomes (EXOs) may provide advantages over cell therapy is very attractive and has been under investigation by many groups for many years. However, the benefits of EXOs compared to cell therapy has not yet been established and clinical trials are very preliminary.

The stimulation of EPCs to produce more and better functioning EXOs has also been proposed by many investigators, but the benefits compared to the risks associated with this process have not been adequately described. The use of calcium silicate (CS) may be as beneficial as other stimulants, but careful comparative studies will be required to determine the differential effects.

The encapsulation of the CS stimulated EXOs from EPCs in hydrogel microspheres to increase their retention is novel and similar to other studies which have employed hydrogels (such as their reference #63).

Results:

The authors should further characterize the EPC mixtures they employed to indicate the variation in cells types contained in the mixture.

What was the ultracentrifugation protocol for the isolation of EXO from EPCs? The sizes in Figure 1G and H represent how many TEM images? What was the purity of the preparation? How does this process compare to alternative approaches? Was the stimulation with CS compared to stimulation

with any other approach? Is CS the best method to stimulate EPCs? Was the content of the angiogenic factors normalized to the number of EXOs?

The authors did not comment on the relevance and in vivo significance of the in vitro HUVEC experiments. The authors determined that 50µg/mL CS-EPC-EXO was the optimal dose in HUVECs. They also need to determine the optimal doses of microsphere-EPC-EXO and microsphere-CS-EPC-EXO for their in vivo studies in their mouse coronary ligation model. Although the CS-EPC-EXOs stimulated cell expansion and tube formation in vitro more than EPC-EXO, a similar benefit cannot be assumed in vivo. Additional studies will be required to elucidate the mechanisms for any proposed in vivo benefits.

The mouse coronary ligation model employed for these studies should be better described. The authors reported that they had an 80% mortality after anterior coronary ligation. This is a very high mortality. Did they ligate the anterior coronary artery proximal to the first septal perforator? Then they state: "Echocardiography was performed immediately after LAD ligation, and animals were randomized primarily based on echocardiography" into the two groups. How was randomization performed? Sealed envelopes? Were the investigators blinded to the group assignments. What echocardiographic criteria were employed to include the animals in the study? Did they employ a specific range of ejection fractions (or fractional shortening)? What were the results of the echocardiographic measurements at time 0 (immediately after coronary ligation)? Were there differences between the groups?

Immediately after ligation the blanched epicardial area in the mouse is very small. The authors said: "20 µL of 1 mg/mL microsphere+EPC-EXO or microsphere+CS-EPC-EXO suspension was injected via 30 G needles at two symmetric sites in the peri-infarct area immediately after MI." What areas were injected? Were the two areas at the cardiac base and apex? The volume of the injectate likely expanded to the entire infarct region on the anterior aspect of the heart. How did the investigators keep the injectate in the peri-infarct region?

The description of the preparation and characterization of the microspheres requires additional information. Are figures 3B, 3D and 3F representative of multiple microscopic examinations? In figure 3D the labelled EXOs could be only on the outside of the microspheres. Were they uniformly distributed? Figure 3F does not adequately describe the EXO release from the microspheres. How many experiments are described in figure 3F? What was the fluorescence intensity of PKH26 red-labeled EXOs (PKH26-EXO) at each time and was the supernatant intensity significantly greater after 14 days? The release of EXOs into PBS may not represent the release rate in vivo. Further experiments in their mouse model are required to document duration of the EXO release in vivo.

The number of animals and specimens examined with each outcome described in figure 4 should be reported. Differences between microsphere+EPC-EXO and microsphere+CS-EPC-EXO on ventricular function change with time following coronary ligation and should be evaluated with a time varying two-way analysis of variance reporting the overall effect (of time and group) and then specifying differences with a multiple range t test. Heart function should be evaluated at multiple time intervals after the infarct and not just at 0 and 21 days. Estimation of systolic and diastolic volumes should be reported as well as EF and FS. Were the differences in EF and FS due to differences in preload and afterload or were they due to differences in muscle contraction? Estimates of ventricular volumes from the echocardiographic measurements provide a guide to differences in the loading conditions which could have resulted from the differences in the cytokine release from the EXOs.

In figures 4A and 4B were the differences between the groups significant? In figure 4F how many hearts were evaluated in each group? TUNEL staining (in figure 4H) may not accurately detect apoptotic cells. Was another index employed? In figure 4I and 4J, how were clumps of smooth muscle cells differentiated from capillaries or arterioles? What were the criteria used to characterize labelled cell clusters as capillaries or arterioles? How many slides were examined in each group and were they evaluated by a blinded observer?

The in vitro cell migration experiments may not reflect in vivo events. Does CD34 and VEGFR2 co-localization staining accurately determine cell recruitment to the infarct region? Double stained cells may also be endogenous cells. How many specimens were examined at 7 days after the infarct (figure 5C)? Were the differences between microsphere+CS-EPC-EXO and microsphere+EPC-EXP statistically significant? Does the assessment of Dio-EPCs injected into the tail vein predict the recruitment of the endogenous bone marrow cells to the infarct? What is the time course of the homing of the Dio-EPCs to the infarct region? Previous studies have demonstrated that injecting EPCs into the circulation does not improve the recovery after a myocardial infarction in humans and the studies in animals are contradictory. Previous studies with labelled bone marrow progenitor cells demonstrated that enhancement of endogenous bone marrow and cardiac resident stem cell homing improves recovery from an infarct, but that intravenous cell injection had no benefits (European Heart Journal 2013;34:1157-1167).

For the results presented in figure 6, please describe the culture condition of the EPCs with and without CS stimulation. The sequencing studies were confirmed with only 3 studies using qRT-PCR. Is that a sufficient number of studies to confirm the up-regulation of the 5 genes? Are the effects in HUVEC cells the same as in in vivo heart tissue or even in vitro EPCs? Angiogenesis was not established in HUVEC cells since capillaries and arterioles were not formed. The evaluation of the migration and vascular tube formation of HUVECs under glucose-oxygen deprivation does not reflect the effects of EXO in vivo.

Discussion

The angiogenic effects of microsphere-CS-EPC-EXOs were greater than microsphere-EPC-EXOs, but the differences were small. How do these interventions compare to angiogenic enhancement by cell, gene or protein therapy? Were the benefits greater in these studies than those obtained by injecting EPCs or EXOs (with or without microspheres) into the infarct region?

The authors demonstrated that CS stimulation of EPCs produced more particles (exosomes?) than unstimulated EPCs. When the concentration of angiogenic cytokines (VEGF, eNOS, HGF, IGF and SDF) is divided by the number of particles, were there more cytokines/particle after CS stimulation or only more particles? Are the CS stimulated EXOs more biologically active or only at a higher concentration? As previously noted, CS stimulation may have advantages and disadvantages if CS induces adverse side effects. To answer this question, the authors should compare CS-EPC-EXO to EPC-EXO in an in vivo model (such as their mouse infarct model) and to the results obtained with stem cells and stem cell derived EXOs. Other authors have reported that other stimuli increase both the secretion and the bioactivity, but the two properties must be studied separately. The authors should also indicate what they believe would be the effects of CS injection into the infarcted heart. Previous studies have indicated that most stimulants enhance angiogenesis and improve ventricular function. However, some agents which stimulate inflammation have been found to induce heart failure and death.

The microsphere encapsulated EPC-EXO may have prolonged the duration of the EXO in the infarct region in the infarct mouse model, but the duration of the EXO secretion was not evaluated in vivo. The microspheres alone may also have induced angiogenesis and a control group with only microspheres without EPC-EXOs was not studied. What is the time course of the benefit of these interventions? Will the benefit last more than 21 days?

Summary:

The authors evaluated an innovative (but complex) intervention with microsphere encapsulated CS stimulated EPC derived EXOs. The results are encouraging. However, the authors should place their results in context of other studies of cells, genes, proteins and extracellular vesicles injected into the infarct region after coronary ligation. In addition, the authors need to include additional controls for their studies.

Response to reviewer #1:

These studies demonstrated a biomaterial-based approach to prepare larger amount of exosomes with more contents from endothelial progenitor cells (EPCs) by bioactive ceramics stimulation. The exosomes were encapsulated in hydrogel microspheres, which effectively mitigate myocardial infarction in an animal model. The authors further demonstrated that the engineered exosomes enhanced vascular repair by increasing the contents of miR-126-3p and angiogenic factors such as VEGF, SDF-1, CXCR4 and eNOS, which activate endothelial cells and recruit EPCs. The authors did great work to generate engineered exosomes and show the therapeutic effects of the EPC exosomes. There are some concerns need to be addressed.

Re: We thank the reviewer for the comments on our work. We have revised the full text according to the suggestions of the reviewer.

1. Using exosomes as a therapeutic agent is very promising, however the field has concerns that the contents of miRNAs or proteins in each exosome are very low, which may not be enough to induce robust effects. Did the authors determine and calculate copies of miR-126-3p and concentrations of angiogenic factors such as VEGF, SDF-1, CXCR4 and eNOS in each exosome? miRNAs levels were usually determined by PCR, where the miRNAs were amplified. Another approach should be used to obtain quantitative data on miRNA levels. Once miRNA and protein contents in each exosome were calculated, the authors can roughly calculate how many exosomes were injected in vivo and convince the readers that enough cargos were delivered in vivo.

Re: According to the reviewer's suggestion, we have calculated the content of angiogenic factors in a single EXOs particle, and the results have been added in Table S5. In our in vivo experiment, the amount of EXOs injected into the mice was 20 µg, and the corresponding amount of different angiogenic factor proteins have been listed in Table S6.

Some studies have shown that $>5 \times 10^{-3} \mu\text{g}$ of growth factors (such as VEGF and HGF) have therapeutic effect on myocardial infarction^{84, 85}. Considering that the amount of HGF in our EXOs was higher than $5 \times 10^{-3} \mu\text{g}$, we believe that 20 µg EXOs could meet the needs of treating myocardial infarction.

In the Methods section, the sentence “The total amount of growth factor is more than $5 \times 10^{-3} \mu\text{g}$, which is known to be effective for the treatment of myocardial infarction (Table S6)^{78,84,85}.” (Page 42 Line 2-3) and the sentence “Furthermore, the cytokine content in each EXOs particle was calculated according to the mass of cytokine (VEGF, eNOS, HGF, IGF and SDF) measured by ELISA and the total number of particles measured by NTA. Calculation formula: $C_{\text{EXOs}} = M_{\text{cyto}} / N_{\text{EXOs}}$; where C_{EXOs} is the mass of cytokine in one EXOs particle, M_{cyto} is the mass of cytokines, N_{EXOs} is the total number of particles.” have been added in the revised manuscript. (Page 39 Line 19-22)

The newly added references in the revised manuscripts are as follows:

78 Chen, C. W. et al. Sustained release of endothelial progenitor cell-derived extracellular vesicles from shear-thinning hydrogels improves angiogenesis and promotes function after myocardial infarction. *Cardiovascular research* 114, 1029-1040, doi:10.1093/cvr/cvy067 (2018).

84 Wu, Y. et al. Release of VEGF and BMP9 from injectable alginate based composite hydrogel for treatment of myocardial infarction. *Bioactive Materials* 6, 520-528, doi:10.1016/j.bioactmat.2020.08.031 (2021).

85 Wang, X. H. et al. Intra-Myocardial Injection of Both Growth Factors and Heart Derived

Sca-1(+)/CD31(-) Cells Attenuates Post-MI LV Remodeling More Than Does Cell Transplantation Alone: Neither Intervention Enhances Functionally Significant Cardiomyocyte Regeneration. Plos One 9, doi:10.1371/journal.pone.0095247 (2014).

2. I would suggest the authors show each individual data point in all the figures.

Re: According to the reviewer's suggestion, we have added each individual data point in all the figures.

3. Figure 1B-F should be labeled with each angiogenic factors.

Re: Thanks to the reviewer, we have labeled each angiogenic factors in Figure 1B-F.

4. For in vivo studies such as the figure 5D, I expected to see the injected exosomes were taken up by endothelial cells. Please clarify or show additional data that the injected exosomes were taken up by ECs.

Re: According to reviewer's comments, we have performed immunofluorescence staining of endothelial cells (CD31), cardiomyocytes (CTNT) and fibroblasts (Vimentin) and examined the co-localization of the cells with PKH26 labeled EXOs, the results showed that the EXOs were taken by these cells.

Therefore, in the results section, the new results have been added in Figure S4 and the sentences "Furthermore, from the immunofluorescence staining images of CTNT, CD31 and Vimentin, it could be clearly observed that CS-EPC-EXO were co-localized with these cells, indicating the up-take of the exosomes by cardiomyocytes, endothelial cells and fibroblasts in infarct heart (Figure S4)." have been added in the revised manuscript. (Page 20 Line 16-18)

In the methods section, the sentences "Evaluation of the internalization of EXOs by cells in vivo. Male C57BL/6 mice were used to evaluated the internalization of EXOs in vivo. First, 20 μ L of microsphere+CS-EPC-EXO (PKH26) (1mg/mL EXOs) was injected immediately after LAD ligation via 30 G needles at two symmetric sites in the infarct area located at the cardiac base. Mice were killed on day 3, and the hearts paraffin sections were stained with immunofluorescence staining (cardiomyocytes (CTNT, 1:500, Abcam), endothelial cells (CD31, 1:500, Servicebio) and fibroblasts (Vimentin, 1:500, Abcam))." have been added in the revised manuscript. (Page 43 Line 11-17)

5. The authors did great work to identify hnRNPA1 and nSMase2 expression were increased in CS stimulated EPC and they are responsible to sort miR-126-3p into exosomes. It is still not clear increase of hnRNPA1 and nSMase2 selectively increase miR-126-3p or few miRNAs or generally increase all the miRNA contents. If several miRNAs or many miRNAs were all increased in EPC exosome, then why only miR-126-3p exert robust effects?

Re: We believe it is possible that miR-126a-3p is not the only miRNA sorted by hnRNPA1 and nSMase2. We have found that other miRNAs related to angiogenesis include miR-486b-5p, miR-150a-5p, miR-26a-5p and miR-142a-3p also increased in the CS-EPC exosomes, which may also be affected by hnRNPA1 and nSMase2. However, we found that the amount of miR-126a-3p is the highest among these miRNAs, and which is known as one of the strongest angiogenesis promoting miRNA among these miRNAs⁶¹, so it is possible that only miR-126a-3p exerts robust effects. But we will consider review's comments and look at the role of other

up-regulated miRNAs and their relation with hnRNPA1 and nSMase2 in our future study.

In order to avoid misunderstanding, we have revised the Discussion section. The sentences “To explain why silicate ion stimulation results in higher bioactivity of obtained EXOs (CS-EPC-EXO), we analyzed differential miRNA expression profiles of different EXOs, which revealed a significant up-regulation of miR-126a-3p, miR-486b-5p, miR-150a-5p, miR-26a-5p and miR-142a-3p. In particular, the amount of miR-126a-3p is the highest among these miRNAs, and which is known as one of the strongest angiogenesis promoting miRNA among these miRNAs⁶¹. Therefore, it is assumed that miR-126a-3p plays a key role for the therapeutic effects, and the higher bioactivity of CS-EPC-EXO in promoting angiogenesis of HUVECs may be mostly related to the up-regulation of miR-126a-3p in CS-EPC-EXO.” have been added in the revised manuscript (page 31 line 27-29 and Page 32 Line 1-5).

The newly added references in the revised manuscripts are as follows:

61 Yan, Y. et al. Transcriptional regulation of microRNA-126a by farnesoid X receptor in vitro and in vivo. *Biotechnology Letters* 42, 1327-1336, doi:10.1007/s10529-020-02864-7 (2020).

6. In results section for figure 8, line 489 should be Figure 8C, D, line 492 should be Figure 8E, F, line 493 should be Figure 8G,H. They were mislabeled.

Re: We have corrected the errors in Figure 8 and checked the full text to avoid similar errors.

7. Why only male mice were used for in vivo studies. This needs appropriate justification.

Re: We have two reasons for using male mice. The first is that we would like to avoid effect of the heredity of mice such as breed, age, weight and sex on the experimental results, so we followed similar protocols reported in the literature for most studies on myocardial infarction, which used same sex mice for experiments^{1,2}. The second is that, considering the surgical injury for the mice in myocardial infarction model and the higher tolerance of male mice³, we used male mice and hope it can improve the success rate during the construction of myocardial infarction model.

Therefore, in Methods section, the sentence “To avoid the effect of the heredity of mice such as breed, age, weight and sex on the experimental results, and considering the surgical injury for the mice in myocardial infarction model and the higher tolerance of male mice, male C57BL/6 mice of 8 weeks of age (average weight is 20g) were selected for this experiment.” was added in the revised manuscript (Page 41 Line 7-10).

Reference:

1 Hoffmann, J. et al. Post-myocardial infarction heart failure dysregulates the bone vascular niche. *Nat. Commun.* 12, doi:10.1038/s41467-021-24045-4 (2021).

2 Park, B. W. et al. In vivo priming of human mesenchymal stem cells with hepatocyte growth factor-engineered mesenchymal stem cells promotes therapeutic potential for cardiac repair. *Science Advances* 6, doi:10.1126/sciadv.aay6994 (2020).

3 Mouton, A. J. et al. Interaction of Obesity and Hypertension on Cardiac Metabolic Remodeling and Survival Following Myocardial Infarction. *Journal of the American Heart Association* 10, doi:10.1161/jaha.120.018212 (2021).

8. In the method section, 20ul of 1mg/ml microsphere+EPC-Exo were injected. Please clarify how many exosomes were injected. This is important for readers to understand what contents were

delivered in vivo and exerted the protective effects.

Re: In this study, 20 μ L solution contains 20 μ g of EXOs. According to reviewer's comments, we have calculated the number of EXOs particles in 1 μ g EXOs, which contains 2.44×10^9 particles. Therefore, 48.8×10^9 particles have been injected into the myocardial infarction site. In the Methods section, the sentences "20 μ L of 1 mg/mL microsphere+EPC-EXO or microsphere+CS-EPC-EXO suspension was injected via 30 G needles at two symmetric sites in the peri-infarct area immediately after MI." have been changed as "20 μ L of microsphere+EPC-EXO or microsphere+CS-EPC-EXO suspension containing 20 μ g of EXOs (1mg/mL EXOs) was injected immediately via 30 G needles at two symmetric sites in the infarct area located at the cardiac base after MI. The total amount of growth factor is more than 5×10^{-3} μ g, which is known to be effective for the treatment of myocardial infarction (Table S6)^{78,84,85}." in the revised manuscript. (Page 41 Line 28-29 and Page 42 Line 1-3)

The newly added references in the revised manuscripts are as follows:

78 Chen, C. W. et al. Sustained release of endothelial progenitor cell-derived extracellular vesicles from shear-thinning hydrogels improves angiogenesis and promotes function after myocardial infarction. *Cardiovascular research* 114, 1029-1040, doi:10.1093/cvr/cvy067 (2018).

84 Wu, Y. et al. Release of VEGF and BMP9 from injectable alginate based composite hydrogel for treatment of myocardial infarction. *Bioactive Materials* 6, 520-528, doi:10.1016/j.bioactmat.2020.08.031 (2021).

85 Wang, X. H. et al. Intra-Myocardial Injection of Both Growth Factors and Heart Derived Sca-1(+)/CD31(-) Cells Attenuates Post-MI LV Remodeling More Than Does Cell Transplantation Alone: Neither Intervention Enhances Functionally Significant Cardiomyocyte Regeneration. *Plos One* 9, doi:10.1371/journal.pone.0095247 (2014).

9. Microsphere+EPC-EXO was injected immediately after injury, please consider post treatment as therapeutic strategy.

Re: We thank for reviewer's suggestion. In this study, the left anterior descending branch (LAD) of the mouse heart was ligated to build the animal model, and the survival rate of the mouse was only 80%, which was similar to the literature report¹. Considering the risk of the second chest open surgery for the post-treatment treatment, which may significantly increase the mortality of mice, we did not include the post treatment in our study this time. But we agree with the reviewer, that the post treatment is also important to verify the applicability of therapeutic strategy. We will consider to investigate therapeutic effects of post treatment at different time points by determining mortality rate of different second operation time in the future.

Reference:

1 Nogami, Y. et al. Cardiac dysfunction induced by experimental myocardial infarction impairs the host defense response to bacterial infection in mice because of reduced phagocytosis of Kupffer cells. *Journal of Thoracic and Cardiovascular Surgery* 140, 624-U185, doi:10.1016/j.jtcvs.2009.11.005 (2010).

Response to reviewer #2:

Remarks to the Author:

In this interesting study, the authors have used calcium silicate ions to induce endothelial progenitor cells to secrete exosomes (CS-EPC-EXO) which enhanced the proliferation of EPCs in comparison to normal media and promoted the expression of angiogenesis-related genes. Silicate ions also improved the yield of EXOs in EPC culture which were found to express key angiogenic factors as well as SDF-1 which is involved in the recruitment of hematopoietic stem cells. In vitro, these exosomes significantly enhanced the vascular tube-formation of HUVECs and prevented their apoptosis under conditions of oxygen and glucose deprivation. Next, the authors used a microfluidics-based technology to encapsulate these exosomes in microspheres made of methacrylated PEG which, in vitro, was able to continuously release exosomes for at least 21 days. In an acute myocardial infarction model in mice, it was then shown that these microspheres prolonged the residency time of the encapsulated exosomes, improved left ventricular function, reduced fibrosis, scar size, cardiomyocyte hypertrophy and apoptosis and promoted angiogenesis in the peripheral area of myocardial infarction, in part through an enhanced recruitment of EPCs. The transcriptomic analysis of the CS-EPC-Exo showed a differential regulation of genes compared with the EPC-Exo, with a strong upregulation of miR-126a-3p whose enrichment in exosomes was triggered by the silicate ion-induced higher mRNA expression levels of hnRNPA2B1 and nSMase2 (SMPD3). This paper addresses a clinically relevant topic. The experiments have been carefully designed and executed and the results look straightforward. Some issue, though, need to be addressed:

1. a. Isolation of exosomes: exosomes released from EPCs cultured in the silicate ion solution were isolated by ultracentrifugation. No details are provided regarding the method which was used precisely (speed, number, duration).

Re: According to the reviewer's suggestion, the details about isolation of exosomes have been provided in the methods section. The sentences "The specific extraction steps are as follows. The cell culture medium of EPCs was first centrifuged for 10 minutes at 300 g to remove the live cells. Then, the supernatant was centrifuged for 10 minutes at 2000 g to remove the dead cells followed by centrifugation for 30 minutes at 10000 g to remove cellular debris. Finally, the supernatant was centrifuged for 70 minutes at 100000 g to obtain EXOs. PBS was used to wash EXOs." have been added in the revised manuscript. (Page 36 Line 26-29 and Page 37 Line 1)

b. The medium was prepared by diluting silicate ions in a serum-free Endothelial Cell Growth Medium-2. This medium, however, is likely to contain growth factors which may "contaminate" the EXO yield and consequently bias the results. Thus, NTA data should definitely include a true control showing that the silicate ion-EGF mix was free from particles that could have been misinterpreted by NTA as exosomes.

Re: According to reviewer's suggestion, CS ion solution has been analyzed by nanoparticle tracking analysis (NTA), and found that there were no particles in it. This result indicates that CS ion solution did not contaminate the EXOs yield in our experiments.

In the results section, the sentence "In addition, no particles were found in CS ion solution, suggesting that the CS ion solution does not affect the determination of EXOs yield (Figure S2)." has been added in the revised manuscript. (Page 7 Line 2-4)

In the Methods section, the sentence “CS ion solution was used as the control group.” has been added in the revised manuscript. (Page 37 Line 7)

2. a. Experimental groups : The authors primarily document the superiority of CS-CPC-Exo over CPC-Exo but in both cases, exosomes were encapsulated in microspheres. They should have also included a group with the exclusive injection of exosomes (without microspheres) to better assess the added value of encapsulation which, in the perspective of a clinical translation, would increase the complexity of manufacturing.

Re: According to reviewer’s suggestion, different exosomes without microspheres encapsulation have been injected to evaluate the treatment effect, and the results showed that the therapeutic effect of EPC-EXO alone was significantly lower than that of Microsphere+EPC-EXO (Figure S5-S10). This is mainly because the exosomes can only stay in the heart for 7 days after a single injection, while the exosomes encapsulated in microspheres can stay in the heart for more than 21 days (Figure 4A). This result proofed the clinical value of encapsulation of exosomes in microspheres.

Therefore, in the results section, the sentences “Moreover, by comparing Microsphere+EPC-EXO injection with the injection of pure EPC-EXO without hydrogel encapsulation, we found that the hydrogel encapsulated exosomes (Microsphere+EPC-EXO) more effectively promote the recovery of cardiac function (Figure S6), inhibit scar and fibrosis (Figure S7), and enhance angiogenesis (Figure S10), indicating that the microspheres could indeed improve the therapeutic effects of EPC-EXO possibly by sustained release of exosomes for longer period.” have been added in the revised manuscript. (Page 17 Line 10-15)

In the discussion section, the sentences “Here, we used Microsphere+CS-EPC-EXO to gradually release highly bioactive EXOs, which ensures sustained delivery of sufficient active factors (SDF-1 and VEGFa) in the myocardial infarction area. After the treatment for 21 days, the cardiac function and ventricular remodeling of the mice were significantly improved, which confirms the effectiveness of Microsphere+CS-EPC-EXO in the treatment of myocardial infarction.” have been changed as “Here, we used Microsphere+CS-EPC-EXO to gradually release highly bioactive EXOs, which ensures sustained delivery of sufficient active factors (SDF-1 and VEGFa) in the myocardial infarction area. After the treatment for 21 days, the cardiac function and ventricular remodeling of the mice were significantly improved, while the treatment using EPC-EXO alone only showed minor improvement as compared to the PBS control group (Figure S5-S10), which confirms the effectiveness of Microsphere+CS-EPC-EXO in the treatment of myocardial infarction, and reflected the clinical value of encapsulation of exosomes in microspheres.” in the revised manuscript. (Page 34 Line 10-17)

In the methods section, the sentences “...animals were randomized primarily based on echocardiography (PBS group, microsphere+EPC-EXO group, and microsphere+CS-EPC-EXO group...” has been changed as “...animals were randomized to four groups primarily based on echocardiography (PBS group, EPC-EXO, microsphere+EPC-EXO group, and microsphere+CS-EPC-EXO group...” in the revised manuscript. (Page 41 Line 24-26)

b. Likewise, a true control group should have consisted of the virgin medium (i.e., the silicate ion-EGF medium alone) instead of PBS to eliminate any potential effect of the medium on the outcomes.

Re: In this study, EPC-EXO, Microsphere+EPC-EXO and Microsphere+CS-EPC-EXO were

dispersed by PBS and then injected into the myocardial infarction area of mice. Therefore, there was no culture medium to interfere with experimental results.

In the Methods section, the sentence “All groups were dispersed with PBS.” has been added in the revised manuscript. (Page 41 Line 26)

c. In addition to ejection fraction values, left ventricular volumes should be reported. Otherwise, it is incorrect to conclude that the treatment prevented adverse remodeling. Finally, the number of animals in each group is missing.

Re: We thank for the reviewer’s suggestion. Here, left ventricular end-diastolic volume (LVVD) and left ventricular end-systolic volume (LVVS) of the animals on day 7, 14 and 21 after myocardial infarction have been supplemented in Figure S6. It could be found that LVVS and LVVD increased gradually in PBS group, but decreased gradually in EPC-EXO, Microsphere+EPC-EXO and Microsphere+CS-EPC-EXO groups. In particular, the reduction was most obvious in the Microsphere+CS-EPC-EXO group. This result suggests that Microsphere+CS-EPC-EXO could effectively treat and prevent the adverse remodeling of myocardial infarction.

In the results section, the sentence “...from the specific values of left ventricular ejection fraction (EF) and left ventricular fractional shortening (FS), it can be clearly seen that Microsphere+CS-EPC-EXO more significantly improved the cardiac systolic function of mice as compared with Microsphere+EPC-EXO (Figure 4D)...” has been changed as “...Interestingly, from the specific values of left ventricular ejection fraction (EF), left ventricular fractional shortening (FS), left ventricular end-diastolic volume (LVVD) and left ventricular end-systolic volume (LVVS), it can be clearly seen that Microsphere+CS-EPC-EXO more significantly improved the cardiac systolic function of mice as compared with Microsphere+EPC-EXO (Figure 4D and Figure S6)...” in the revised manuscript. (Page 15 Line 25-29 and Page 16 Line 1)

In the methods section, the sentence “On day 0 and 21, cardiac function was measured by the Vevo® 2100 system (FUJIFILM Visual Sonics, Canada). Cardiac ejection fraction (EF) and fractional shortening (FS) were assessed according to M-mode and B-mode echocardiography.” has been changed as “On day 0, 7, 14 and 21, cardiac function was measured by the Vevo® 2100 system (FUJIFILM Visual Sonics, Canada). Cardiac ejection fraction (EF), fractional shortening (FS), left ventricular end-diastolic volume (LVVD) and left ventricular end-systolic volume (LVVS) were assessed according to M-mode and B-mode echocardiography.” in the revised manuscript. (Page 42 Line 12-15)

3. Scar size was assessed by Masson staining, which is appropriate. However, it is not correct to mention a reduction of fibrosis (likely interstitial fibrosis) without performing Sirius Red stainings. These stainings should be performed or the mention of fibrosis has to be deleted.

Re: According to the reviewer’s suggestion, Sirius Red staining has been performed to evaluate fibrosis in the infarcted area. The results showed that Sirius red staining was obvious in the area positive-stained by Masson staining, and the quantitative statistical analysis of Sirius Red revealed that Microsphere+CS-EPC-EXO had the best inhibitory effect on fibrotic scar tissue (Figure S7).

In the results section, the sentence “Sirius red staining further proved that these scars were

fibrosis tissue (Figure S7A), and quantitative statistical analysis determined that Microphere+CS-EPC-EXO indeed had the best inhibitory effect on fibrotic scar tissue (Figure S7B and C).” has been added in the revised manuscript. (Page 16 Line 5-7)

In the methods section, the sentences “Heart tissues from different groups were fixed using 4% paraformaldehyde and then embedded in paraffin and sliced (thickness = 4 μm), which were subjected to Masson staining (Sigma Aldrich, USA), and Alexa Fluor 488 and wheat germ agglutinin staining (ThermoFisher, USA). Then, Image-Pro Plus software was used to evaluate the parameters of left ventricular remodeling (LV wall thickness, relative scar thickness, and infarct size) from Masson staining images...” have been changed as “Heart tissues from different groups were fixed using 4% paraformaldehyde and then embedded in paraffin and sliced (thickness = 4 μm), which were subjected to Masson staining (Sigma Aldrich, USA), Sirius red staining (Sigma Aldrich, USA), and Alexa Fluor 488 and wheat germ agglutinin staining (ThermoFisher, USA). Then, Image-Pro Plus software was used to evaluate the parameters of left ventricular remodeling (LV wall thickness, relative scar thickness, and infarct size) from Masson (Masson positive staining area represents scar.) and Sirius red staining images (Sirius red positive areas represent fibrosis.)^{86,87}...” in the revised manuscript. (Page 42 Line 18-24)

The newly added references in the revised manuscripts are as follows:

86 Fan, G. P. et al. Pharmacological Inhibition of Focal Adhesion Kinase Attenuates Cardiac Fibrosis in Mice Cardiac Fibroblast and Post-Myocardial-Infarction Models. *Cellular Physiology and Biochemistry* 37, 515-526, doi:10.1159/000430373 (2015).

87 Li, X. et al. Activation of Cannabinoid Receptor Type II by AM1241 Ameliorates Myocardial Fibrosis via Nrf2-Mediated Inhibition of TGF-beta 1/Smad3 Pathway in Myocardial Infarction Mice. *Cellular Physiology and Biochemistry* 39, 1521-1536, doi:10.1159/000447855 (2016).

4. It is unclear whether outcomes were assessed blindly or not.

Re: All the results were assessed blindly by three people in this study. We have further improved the Method section. The sentence “All the results were assessed blindly by three people in this study.” has been added in the revised manuscript (page 43 line 28).

5. Internalization of exosomes: The authors report that exosomes mediated angiogenesis in HUVECs by transferring miR-126a-3p. On the other hand, they used PKH26-labeled EXOs to monitor the fluorescence signals in cardiac slices at different time periods after injection. In which cell type were exosomes internalized? This identification should be made possible by co-staining exosomes with cell-specific (cardiomyocyte, fibroblast, endothelial cell) markers.

Re: According to reviewer’s comments, we have performed immunofluorescence staining of endothelial cells (CD31), cardiomyocytes (CTNT) and fibroblasts (Vimentin) and examined the co-localization of the cells with PKH26 labeled EXOs, the results showed that the EXOs were taken by these cells.

Therefore, in the results section, the new results have been added in Figure S4 and the sentences “Furthermore, from the immunofluorescence staining images of CTNT, CD31 and Vimentin, it could be clearly observed that CS-EPC-EXO were co-localized with these cells, indicating the up-take of the exosomes by cardiomyocytes, endothelial cells and fibroblasts in

infarct heart (Figure S4).” have been added in the revised manuscript. (Page 20 Line 16-18)
In the methods section, the sentences “Evaluation of the internalization of EXOs by cells in vivo. Male C57BL/6 mice were used to evaluate the internalization of EXOs in vivo. First, 20µL of microsphere+CS-EPC-EXO (PKH26) (1mg/mL EXOs) was injected immediately after LAD ligation via 30 G needles at two symmetric sites in the infarct area located at the cardiac base. Mice were killed on day 3, and the hearts paraffin sections were stained with immunofluorescence staining (cardiomyocytes (CTNT, 1:500, Abcam), endothelial cells (CD31, 1:500, Servicebio) and fibroblasts (Vimentin, 1:500, Abcam)).” have been added in the revised manuscript. (Page 43 Line 11-17)

6. Exosomes harbor few copies of miRNAs. In their experiments, the authors show different expression levels of miR-126a-3p and conclude to its predominant role on the basis of indirect findings (migration and vascular tube formation of HUVECs under glucose-oxygen deprivation). However, they do not provide direct evidence for a transfer of this miRNA from the exosomes to the recipient cells and the increased levels of miR-126a-3p in these cells might well reflect an increased endogenous synthesis triggered by one of the components of the exosome cargo. In the absence of a direct visualisation of the transfer of miR-126a-3p into HUVEC, the authors cannot definitely conclude to this transfer. Thus, they should either provide direct evidence for this transfer or reformulate the paper accordingly. In line with this statement, the discussion highlights the role of miR126a-3 but it should also be mentioned that other components of the EXO cargo (miRNAs, proteins) are likely involved in the cardiac protection observed in these experiments.

Re: According to reviewer’s comments, we have evaluated the direct visualisation of the transfer of miR-126a-3p, and found that miR-126-3p could indeed transfer from EPCs to HUVECs by EXOs. In this experiment, we used miR-126-3p simulant with cy5 fluorescence to transfect EPCs, and collected the EXOs secreted from these EPCs. Then, HUVECs were cultured with the obtained EXOs, and the localization of the fluorescence labeled miR-126-3p in HUVECs was observed (Figure S13).

Therefore, in the Methods section, the sentences “Furthermore, miR-126-3p simulant with cy5 fluorescence was used to transfect EPCs, and EXOs from these EPCs were collected to culture HUVECs for 24 hours. The transfer of miR-126a-3p into HUVECs were observed by microscopy (Leica DMI 3000B, Germany).” have been added in the revised manuscript. (Page 40 Line 7-9)

In the Results section, the sentences “Furthermore, we used miR-126-3p simulant with cy5 fluorescence to transfect EPCs, and collected the EXOs secreted from these EPCs. Then, HUVECs were cultured with the obtained EXOs. Interestingly, we found that EPCs can indeed transfer miR-126a-3p to HUVECs through EXOs (Figure S13).” have been added in the revised manuscript. (Page 22 Line 21-24)

Response to reviewer #3:

1. Yu and colleagues describe an extensive evaluation of an innovative intervention to induce angiogenesis after coronary ligation in a mouse model. A variety of angiogenic therapies for a myocardial infarction have been reported in both animal models as well as in humans in the last 20 years. The effects of protein, gene and cell therapies have been controversial and no angiogenic therapy is currently recommended for patients after a myocardial infarction. The authors suggest that their new, complex intervention may have advantages over other approaches. However, a more extensive comparison with other approaches may be necessary to confirm their suggestion. Previous studies have demonstrated that the injection of EPCs and other types of stem cells into the peri-infarct region increased new blood vessel formation and improved heart function in both animal models and in some clinical trials. EXOs from a variety of cell sources have also been employed to increase angiogenesis and restore function after coronary occlusion in animal models (Eur Heart J 2017;38:201–211, J Cardiovasc Transl Res 2019;12:5–17, Cardiovasc Res 2021;117:292–307 for example). Although the authors are correct that the injection of EXOs into the infarcted heart may have limited retention, these studies demonstrated that they are effective in inducing angiogenesis and preserving ventricular function. The authors are requested to compare their microsphere+EPC-EXO studies with studies which employed cells or cell derived EXOs to put their studies into the context of the current literature.

Re: According to reviewer’s comments, we have also supplemented experiments to compare the therapeutic effects of EPC-EXO and Microsphere+EPC-EXO, and found that Microsphere+EPC-EXO more effectively promoted the recovery of cardiac function as compared with EPC-EXO (Figure S6 and S7). This result revealed that the injection of microsphere encapsulated exosomes are more effective than the injection of pure exosomes. Moreover, as the reviewer said, exosomes from various cell sources have been used to treat myocardial infarction. However, most studies only observed the effect of exosomes in the early stage of myocardial infarction as reported in the literature. For example, in the three representative works recommended by reviewer²⁷⁻²⁹, the authors euthanized the animals after 48 or 72 hours to observe the effect of the exosomes, and the long-term therapeutic effect is not clear. In contrast, our results demonstrated that microspheres can improve the retention rate of exosomes and increase the therapeutic effect as compared to the injection of pure exosomes, which disappeared after 7 days.

Therefore, in the Introduction section, the sentences “Furthermore, considering the treatment of myocardial infarction using EXOs, most studies using EXOs injection to treat myocardial infarction only observed the therapeutic effect within 72 hours²⁷⁻²⁹ possibly due to the low retention rate of EXOs in vivo.” have been added in the revised manuscript. (Page 4 Line 22-25)

In the results section, the sentences “Moreover, by comparing Microsphere+EPC-EXO injection with the injection of pure EPC-EXO without microsphere encapsulation, we found that the microsphere encapsulated exosomes (Microsphere+EPC-EXO) more effectively promoted the recovery of cardiac function (Figure S6), inhibited scar and fibrosis (Figure S7), and enhanced angiogenesis (Figure S10), indicating that the microspheres could indeed improve the therapeutic effects of EPC-EXO possibly by sustained release of exosomes for longer period.” have been added in the revised manuscript. (Page 17 Line 10-15)

In the methods section, the sentences "...animals were randomized to primarily based on echocardiography (PBS group, microsphere+EPC-EXO group, and microsphere+CS-EPC-EXO group..." has been changed as "...animals were randomized to four groups primarily based on echocardiography (PBS group, EPC-EXO, microsphere+EPC-EXO group, and microsphere+CS-EPC-EXO group..." in the revised manuscript. (Page 41 Line 24-26)

The newly added references in the revised manuscripts are as follows:

27 Correa, B. L. et al. Extracellular vesicles from human cardiovascular progenitors trigger a reparative immune response in infarcted hearts. *Cardiovascular Research* 117, 292-307, doi:10.1093/cvr/cvaa028 (2021).

28 Maring, J. A. et al. Cardiac Progenitor Cell-Derived Extracellular Vesicles Reduce Infarct Size and Associate with Increased Cardiovascular Cell Proliferation. *Journal of Cardiovascular Translational Research* 12, 5-17, doi:10.1007/s12265-018-9842-9 (2019).

29 Gallet, R. et al. Exosomes secreted by cardiosphere-derived cells reduce scarring, attenuate adverse remodelling, and improve function in acute and chronic porcine myocardial infarction. *Eur. Heart J.* 38, 201-211, doi:10.1093/eurheartj/ehw240 (2017).

2. Although the authors declare no competing interests, do they have a patent on the use of silicate bioceramics ionic solutions?

Re: Yes, we have applied for two patents on silicate bioceramic ion solutions for the treatment of myocardial infarction (ZL201810290955.3 and PCT/CN2018/113043), which do not use exosomes and are different with the technique in this manuscript. In addition, these two patents are owned by our institute and some of authors in this manuscript are only inventors of the patents. Therefore, there is no conflict of interest in this study.

3. The authors employed bone marrow derived EPCs for their studies but did not further characterize these cells or indicate the spectrum of the cell types in their preparation. (a) How many CD34 or CD133 cells are contained in their mixture? Previous studies have demonstrated that the isolation techniques employed greatly influence the angiogenic potential of the final mixture. The authors suggest that they followed the technique described in their reference #63. However, that excellent study from Philadelphia employed rats rather than mice. That study also isolated extracellular vesicles from EPCs (which were well characterized) and then incorporated into a shear thinning gel and injected into the infarct region. (b) In each of their figures those authors depicted the results of each animal or experiment (which Yu et al should consider) and (c) they carefully evaluated serial ventricular function with a pressure volume catheter.

a. How many CD34 or CD133 cells are contained in their mixture?

Re: According to reviewer's suggestion, the immunofluorescence staining and flow cytometry have been used to identify the cell types in the isolated EPCs. The results showed that 98.6% of the cells expressed CD133, 74.5% expressed CD34, and 87.4% expressed VEGFR2.

As described in literature that CD133 (>86.1%)¹, CD34 (>60%)² and VEGFR2 (>75%)³ are the main cell markers of EPCs^{30,31}, our results indicate that the majority of the cells in our preparation are EPCs.

Therefore, in the methods section the sentence "Bone marrow-derived endothelial progenitor cells (EPCs) were isolated from C57Bl/6 mouse." has been changed as "Bone marrow-derived

endothelial progenitor cells (EPCs) were isolated from C57Bl/6 mouse according to a published method^{78,79}, and characterized by CD133, CD34 and VEGFR2 immunofluorescence staining and flow cytometry^{30,31}.” in the revised manuscript. (Page 36 Line 5-7)

In the results section, the sentences “First, EPCs isolated from C57Bl/6 mouse was characterized by CD133, CD34 and VEGFR2 immunofluorescence staining and flow cytometry^{30,31}. The results showed that 98.6% of the cells expressed CD133, 74.5% expressed CD34, and 87.4% expressed VEGFR2. Since CD133, CD34 and VEGFR2 are the main cell markers of EPCs^{30,31}, this result suggests that EPCs were successfully isolated and could be used for subsequent experiments (Figure S1).” have been added in the revised manuscript. (Page 6 Line 2-6)

The newly added references in the revised manuscripts are as follows:

30 Khan, S. S., Solomon, M. A. & McCoy, J. P. Detection of circulating endothelial cells and endothelial progenitor cells by flow cytometry. *Cytometry Part B-Clinical Cytometry* 64B, 1-8, doi:10.1002/cyto.b.20040 (2005).

31 Masouleh, B. K., Baraniskin, A., Schmiegel, W. & Schroers, R. Quantification of circulating endothelial progenitor cells in human peripheral blood: Establishing a reliable flow cytometry protocol. *Journal of Immunological Methods* 357, 38-42, doi:10.1016/j.jim.2010.03.015 (2010).

78 Chen, C. W. et al. Sustained release of endothelial progenitor cell-derived extracellular vesicles from shear-thinning hydrogels improves angiogenesis and promotes function after myocardial infarction. *Cardiovascular research* 114, 1029-1040, doi:10.1093/cvr/cvy067 (2018).

79 Carneiro, G. D., Godoy, J. A. P., Werneck, C. C. & Vicente, C. P. Differentiation of C57/BL6 mice bone marrow mononuclear cells into early endothelial progenitors cells in different culture conditions. *Cell Biology International* 39, 1138-1150, doi:10.1002/cbin.10487 (2015).

Reference:

1 Peichev, M. et al. Expression of VEGFR-2 and AC133 by circulating human CD34(+) cells identifies a population of functional endothelial precursors. *Blood* 95, 952-958, doi:10.1182/blood.V95.3.952.003k27_952_958 (2000).

2 Zayed, S. A. et al. Production of endothelial progenitor cells obtained from human Wharton's jelly using different culture conditions. *Biotechnic & Histochemistry* 91, 532-539, doi:10.1080/10520295.2016.1250284 (2016).

3 Van Craenenbroeck, E. M. F. et al. Quantification of circulating endothelial progenitor cells: A methodological comparison of six flow cytometric approaches. *Journal of Immunological Methods* 332, 31-40, doi:10.1016/j.jim.2007.12.006 (2008).

b. In each of their figures those authors depicted the results of each animal or experiment (which Yu et al should consider) and c. they carefully evaluated serial ventricular function with a pressure volume catheter.

Re: According to the reviewer's suggestion, each individual data point has been added in all the figures.

In addition, since we do not have a pressure volume catheter, we used ultrasound to

supplement the cardiac function of mice on day 0, 7, and 14, including ejection fraction (EF), fractional shortening (FS), left ventricular end-diastolic volume (LVVD) and left ventricular end-systolic volume (LVVS) (Figure S5 and S6).

4. The concept that exosomes (EXOs) may provide advantages over cell therapy is very attractive and has been under investigation by many groups for many years. However, the benefits of EXOs compared to cell therapy has not yet been established and clinical trials are very preliminary.

Re: We agree with the reviewer and will consider comparative studies to compare our bioactive EXOs approach with cell therapy in the future.

5. The stimulation of EPCs to produce more and better functioning EXOs has also been proposed by many investigators, (a) but the benefits compared to the risks associated with this process have not been adequately described. The use of calcium silicate (CS) may be as beneficial as other stimulants, (b) but careful comparative studies will be required to determine the differential effects. The encapsulation of the CS stimulated EXOs from EPCs in hydrogel microspheres to increase their retention is novel and similar to other studies which have employed hydrogels (such as their reference #63).

Re: We agree with the reviewer that many researches are devoted to improving the activity and yield of exosomes by using different stimulants, such as calcium ion stimulation, chemical drug stimulation and growth factor stimulation. However, some of these stimulants can only increase the yield of exosomes, some can only increase the activity of exosomes, and some even have proinflammatory effect. One of the advantages of the CS ionic solution is the promotion of both the secretion amount and bioactivity of exosomes in EPCs. More importantly, the stimulation of CS seems more cell friendly and have been found to inhibit inflammation and promote angiogenesis^{57,58}, and have a unique protective effect on myocardium¹. However, indeed as indicated by the reviewer, more careful comparative studies will be required to evaluate the applicability of the CS-EPC-EXO exosomes. So, we will consider reviewer's suggestion in our future studies by comparing our technique with different exosomes preparation approaches.

Therefore, in the discussion section, the sentences "CS is a unique biomaterial, which has good anti-inflammatory and vascular regeneration effects^{57,58}." have been added in the revised manuscript. (Page 30 Line 28-29)

The newly added references in the revised manuscripts are as follows:

57 Wang, X. T. et al. Chitosan/Calcium Silicate Cardiac Patch Stimulates Cardiomyocyte Activity and Myocardial Performance after Infarction by Synergistic Effect of Bioactive Ions and Aligned Nanostructure. *Acs Applied Materials & Interfaces* 11, 1449-1468, doi:10.1021/acsami.8b17754 (2019).

58 Que, Y. et al. Silicate ions as soluble form of bioactive ceramics alleviate aortic aneurysm and dissection. *Bioactive Materials*, doi:https://doi.org/10.1016/j.bioactmat.2022.07.005 (2022).

Reference:

1 Yi, M. et al. Ion Therapy: A Novel Strategy for Acute Myocardial Infarction. *Adv. Sci.* 6, 1801260, doi:10.1002/advs.201801260 (2019).

6. The authors should further characterize the EPC mixtures they employed to indicate the variation in cells types contained in the mixture.

Re: According to reviewer's suggestion, the immunofluorescence staining and flow cytometry have been used to identify the cell types in the isolated EPCs. The results showed that 98.6% of the cells expressed CD133, 74.5% expressed CD34, and 87.4% expressed VEGFR2.

As described in literature that CD133 (>86.1%)¹, CD34 (>60%)² and VEGFR2 (>75%)³ are the main cell markers of EPCs^{30,31}, our results indicate that the majority of the cells in our preparation are EPCs.

Therefore, in the methods section the sentence "Bone marrow-derived endothelial progenitor cells (EPCs) were isolated from C57Bl/6 mouse." has been changed as "Bone marrow-derived endothelial progenitor cells (EPCs) were isolated from C57Bl/6 mouse according to a published method^{78,79}, and characterized by CD133, CD34 and VEGFR2 immunofluorescence staining and flow cytometry^{30,31}." in the revised manuscript. (Page 36 Line 5-7)

In the results section, the sentences "First, EPCs isolated from C57Bl/6 mouse was characterized by CD133, CD34 and VEGFR2 immunofluorescence staining and flow cytometry^{30,31}. The results showed that 98.6% of the cells expressed CD133, 74.5% expressed CD34, and 87.4% expressed VEGFR2. Since CD133, CD34 and VEGFR2 are the main cell markers of EPCs^{30,31}, this result suggests that EPCs were successfully isolated and could be used for subsequent experiments (Figure S1)." have been added in the revised manuscript. (Page 6 Line 2-6)

The newly added references in the revised manuscripts are as follows:

30 Khan, S. S., Solomon, M. A. & McCoy, J. P. Detection of circulating endothelial cells and endothelial progenitor cells by flow cytometry. *Cytometry Part B-Clinical Cytometry* 64B, 1-8, doi:10.1002/cyto.b.20040 (2005).

31 Masouleh, B. K., Baraniskin, A., Schmiegel, W. & Schroers, R. Quantification of circulating endothelial progenitor cells in human peripheral blood: Establishing a reliable flow cytometry protocol. *Journal of Immunological Methods* 357, 38-42, doi:10.1016/j.jim.2010.03.015 (2010).

78 Chen, C. W. et al. Sustained release of endothelial progenitor cell-derived extracellular vesicles from shear-thinning hydrogels improves angiogenesis and promotes function after myocardial infarction. *Cardiovascular research* 114, 1029-1040, doi:10.1093/cvr/cvy067 (2018).

79 Carneiro, G. D., Godoy, J. A. P., Werneck, C. C. & Vicente, C. P. Differentiation of C57/BL6 mice bone marrow mononuclear cells into early endothelial progenitors cells in different culture conditions. *Cell Biology International* 39, 1138-1150, doi:10.1002/cbin.10487 (2015).

Reference:

1 Peichev, M. et al. Expression of VEGFR-2 and AC133 by circulating human CD34(+) cells identifies a population of functional endothelial precursors. *Blood* 95, 952-958, doi:10.1182/blood.V95.3.952.003k27_952_958 (2000).

2 Zayed, S. A. et al. Production of endothelial progenitor cells obtained from human Wharton's jelly using different culture conditions. *Biotechnic & Histochemistry* 91, 532-539, doi:10.1080/10520295.2016.1250284 (2016).

3 Van Craenenbroeck, E. M. F. et al. Quantification of circulating endothelial progenitor cells: A methodological comparison of six flow cytometric approaches. *Journal of Immunological Methods* 332, 31-40, doi:10.1016/j.jim.2007.12.006 (2008).

7. (a) What was the ultracentrifugation protocol for the isolation of EXO from EPCs? (b) The sizes in Figure 1G and H represent how many TEM images? (c) What was the purity of the preparation? How does this process compare to alternative approaches? (d) Was the stimulation with CS compared to stimulation with any other approach? (e) Is CS the best method to stimulate EPCs? (f) Was the content of the angiogenic factors normalized to the number of EXOs?

a. What was the ultracentrifugation protocol for the isolation of EXO from EPCs?

Re: According to the reviewer's suggestion, the ultracentrifugation protocol for the isolation of EXOs from EPCs have been supplemented in the Methods section. The sentences "The specific extraction steps are as follows. The cell culture medium of EPCs was first centrifuged for 10 minutes at 300 g to remove the live cells. Then, the supernatant was centrifuged for 10 minutes at 2000 g to remove the dead cells followed by centrifugation for 30 minutes at 10000 g to remove cellular debris. Finally, the supernatant was centrifuged for 70 minutes at 100000 g to obtain EXOs. PBS was used to wash EXOs." have been added in the revised manuscript. (Page 36 Line 26-29 and Page 37 Line 1)

b. The sizes in Figure 1G and H represent how many TEM images?

Re: Figure 1G is a representative image of 3 TEM images. Therefore, we have added n=3 in the caption of Figure 1G. Figure 1H were analyzed by nanoparticle tracking analysis (NTA, ZetaView Instrument), which was not associated with TEM images.

c. What was the purity of the preparation? How does this process compare to alternative approaches?

Re: We have calculated the purity of the EXOs. The purity of the CS-EPC-EXO was 2.32×10^9 particles/ μg , and the purity of the EPC-EXO was 2.43×10^9 particles/ μg (Table S7), which was in the same level as the purity of exosomes prepared by most commercial kits ($>1.5 \times 10^9$)^{32,33}. In the Method section, the sentences "The total protein mass of EXOs were analyzed by BCA Protein Quantitative Kit (ThermoFisher). The purity of EXOs were calculated using the following Equation: Purity (Particles/ μg) = Particles of EXOs/ The total protein mass of EXOs." have been added in the revised manuscript. (Page 37 Line 7-10)

In the results section, the sentences "The purity of the CS-EPC-EXO was 2.32×10^9 particles/ μg , and the purity of the EPC-EXO was 2.43×10^9 particles/ μg (Table S7), which was in the same level as the purity of exosomes separated by most commercial kits ($>1.5 \times 10^9$)^{32,33}." have been added in the revised manuscript. (Page 7 Line 6-8)

The newly added references in the revised manuscripts are as follows:

32 Cai, S. et al. Immuno-modified superparamagnetic nanoparticles via host-guest interactions for high-purity capture and mild release of exosomes. *Nanoscale* 10, 14280-14289, doi:10.1039/c8nr02871k (2018).

33 Forteza-Genestra, M. A. et al. Purity Determines the Effect of Extracellular Vesicles Derived from Mesenchymal Stromal Cells. *Cells* 9, doi:10.3390/cells9020422 (2020).

d. Was the stimulation with CS compared to stimulation with any other approach? e. Is CS the best

method to stimulate EPCs?

Re: As indicated by the reviewer, different stimulation approaches have been investigated, such as calcium ion stimulation, chemical drug stimulation and growth factor stimulation. However, some of these stimulants can only increase the yield of exosomes, some can only increase the activity of exosomes, and some even have proinflammatory effect. One of the advantages of the CS ionic solution is the promotion of both the secretion amount and bioactivity of exosomes in EPCs. More importantly, the stimulation of CS seems more cell friendly and have been found to inhibit inflammation and promote angiogenesis^{1,2}, and have a unique protective effect on myocardium³. Therefore, based on our results, we believe that the CS stimulation is a better preparation method than other stimulation approaches reported in the literature. However, more careful comparative studies will be required to evaluate the applicability of the CS-EPC-EXO exosomes. So, we will consider reviewer's suggestion in our future studies by comparing our technique with different exosomes preparation approaches.

Reference:

1 Wang, X. T. et al. Chitosan/Calcium Silicate Cardiac Patch Stimulates Cardiomyocyte Activity and Myocardial Performance after Infarction by Synergistic Effect of Bioactive Ions and Aligned Nanostructure. *Acs Applied Materials & Interfaces* 11, 1449-1468, doi:10.1021/acsami.8b17754 (2019).

2 Que, Y. et al. Silicate ions as soluble form of bioactive ceramics alleviate aortic aneurysm and dissection. *Bioactive Materials*, doi:https://doi.org/10.1016/j.bioactmat.2022.07.005 (2022).

3 Yi, M. et al. Ion Therapy: A Novel Strategy for Acute Myocardial Infarction. *Adv. Sci.* 6, 1801260, doi:10.1002/advs.201801260 (2019).

f. Was the content of the angiogenic factors normalized to the number of EXOs?

Re: Yes, when we used PCR or ELISA to detect angiogenic cytokines, the content of the angiogenic factors was normalized to the number of EXOs.

In the methods section, the sentence "In particular, equal amounts of EPCs, HUVECs, and EXOs were used to extract RNA." has been added in the revised manuscript. (Page 38 Line 19)

8. The authors did not comment on the relevance and in vivo significance of the in vitro HUVEC experiments. The authors determined that 50µg/mL CS-EPC-EXO was the optimal dose in HUVECs. They also need to determine the optimal doses of microsphere-EPC-EXO and microsphere-CS-EPC-EXO for their in vivo studies in their mouse coronary ligation model. Although the CS-EPC-EXOs stimulated cell expansion and tube formation in vitro more than EPC-EXO, a similar benefit cannot be assumed in vivo. Additional studies will be required to elucidate the mechanisms for any proposed in vivo benefits.

Re: We agree with the reviewer that the in vitro activity effect of the exosomes will be different than the in vivo one. We selected the primary doses of the exosomes for in vivo experiment based on the literature³⁵, which showed that the appropriate internal dosage of exosomes for mice experiments was in the range of 10-50µg, so we used 20 µg for our experiment. The primary purpose of the in vivo experiment is to compare the microsphere-EPC-EXO and microsphere-CS-EPC-EXO. We will further investigate the optimal dosage for the best

therapeutic effect in the future for potential clinical applications.

In the Results section, the sentence “The Microsphere+CS-EPC-EXO microspheres were injected into the infarct area of mice to evaluate the therapeutic function at a dosage of 20µg EXOs based on a standard injection protocol reported in the literature³⁵.” has been added in the revised manuscript. (Page 15 Line 4-6)

The newly added references in the revised manuscripts are as follows:

35 Kennedy, T. L., Russell, A. J. & Riley, P. Experimental limitations of extracellular vesicle-based therapies for the treatment of myocardial infarction. *Trends in Cardiovascular Medicine* 31, 405-415, doi:10.1016/j.tcm.2020.08.003 (2021).

9. (a) The mouse coronary ligation model employed for these studies should be better described. The authors reported that they had an 80% mortality after anterior coronary ligation. This is a very high mortality. Did they ligate the anterior coronary artery proximal to the first septal perforator? (b) Then they state: “Echocardiography was performed immediately after LAD ligation, and animals were randomized primarily based on echocardiography” into the two groups. How was randomization performed? Sealed envelopes? Were the investigators blinded to the group assignments. (c) What echocardiographic criteria were employed to include the animals in the study? (d) Did they employ a specific range of ejection fractions (or fractional shortening)? What were the results of the echocardiographic measurements at time 0 (immediately after coronary ligation)? Were there differences between the groups?

a. The mouse coronary ligation model employed for these studies should be better described. The authors reported that they had an 80% mortality after anterior coronary ligation. This is a very high mortality. Did they ligate the anterior coronary artery proximal to the first septal perforator?

Re: According to reviewer’s comments, we have checked the description and found that it was a writing error about the mortality. In this study, the mortality rate of mice was about 20%, and the survival rate was 80%.

In the method section, the sentence “The mortality rate of animals was 80% throughout the experiment.” has been changed as “The survival rate of animals was 80% throughout the experiment.” in the revised manuscript. (Page 41 Line 18)

b. Then they state: “Echocardiography was performed immediately after LAD ligation, and animals were randomized primarily based on echocardiography” into the two groups. How was randomization performed? Sealed envelopes? Were the investigators blinded to the group assignments.

Re: First, we selected all mice with ejection fractions (EF) about 50% and fractional shortening (FS) about 25% (Figure S5) after modeling for further experiments according to literature⁸³ to ensure that the degree of heart damage in each mouse was consistent. Based on this criterion, we randomly selected 5 mice for each group.

In the methods section, the sentence “Mice with ejection fractions (EF) about 50% and fractional shortening (FS) about 25% was selected in this study according to literature⁸³ to ensure the initial myocardial infarction degree of each mouse is consistent (Figure S5).” (Page 41 Line 21-24) and the sentence “The number of animals in each group was 5.” (Page 41 Line 26-27) have been added in the revised manuscript.

The newly added references in the revised manuscripts are as follows:

83 Li, H. K. et al. Injectable AuNP-HA matrix with localized stiffness enhances the formation of gap junction in engrafted human induced pluripotent stem cell-derived cardiomyocytes and promotes cardiac repair. *Biomaterials* 279, doi:10.1016/j.biomaterials.2021.121231 (2021).

c. What echocardiographic criteria were employed to include the animals in the study?

Re: In this study, we selected mice with ejection fractions (EF) about 50% and fractional shortening (FS) about 25% after modeling according to literature⁸³ and randomly assigned them to each experimental group (Figure S5).

In the methods section, the sentence “Mice with ejection fractions (EF) about 50% and fractional shortening (FS) about 25% was selected in this study according to literature⁸³ to ensure the initial myocardial infarction degree of each mouse is consistent (Figure S5).” have been added in the revised manuscript. (Page 41 Line 21-24)

The newly added references in the revised manuscripts are as follows:

83 Li, H. K. et al. Injectable AuNP-HA matrix with localized stiffness enhances the formation of gap junction in engrafted human induced pluripotent stem cell-derived cardiomyocytes and promotes cardiac repair. *Biomaterials* 279, doi:10.1016/j.biomaterials.2021.121231 (2021).

d. Did they employ a specific range of ejection fractions (or fractional shortening)? What were the results of the echocardiographic measurements at time 0 (immediately after coronary ligation)? Were there differences between the groups?

Re: In this study, mice with ejection fractions (EF) about 50% and fractional shortening (FS) about 25% was selected immediately after coronary ligation (Day 0) according to literature⁸³, and the degree of myocardial infarction of mice in each group was consistent according to echocardiography on day 0. Therefore, we have supplemented the cardiac ultrasound of mice in each group on day 0 in the supporting information (Figure S5).

In the methods section, the sentence “Mice with ejection fractions (EF) about 50% and fractional shortening (FS) about 25% was selected in this study according to literature⁸³ to ensure the initial myocardial infarction degree of each mouse is consistent (Figure S5).” have been added in the revised manuscript. (Page 41 Line 21-24)

The newly added references in the revised manuscripts are as follows:

83 Li, H. K. et al. Injectable AuNP-HA matrix with localized stiffness enhances the formation of gap junction in engrafted human induced pluripotent stem cell-derived cardiomyocytes and promotes cardiac repair. *Biomaterials* 279, doi:10.1016/j.biomaterials.2021.121231 (2021).

10. Immediately after ligation the blanched epicardial area in the mouse is very small. The authors said: “20 μ L of 1 mg/mL microsphere+EPC-EXO or microsphere+CS-EPC-EXO suspension was injected via 30 G needles at two symmetric sites in the peri-infarct area immediately after MI.” (a) What areas were injected? Were the two areas at the cardiac base and apex? (b) The volume of the injectate likely expanded to the entire infarct region on the anterior aspect of the heart. How did the investigators keep the injectate in the peri-infarct region?

a. What areas were injected? Were the two areas at the cardiac base and apex?

Re: After ligating the coronary artery, we selected two symmetrical points located at the cardiac base for injection.

We have modified the Method section. The sentence “20 μ L of 1 mg/mL microsphere+EPC-EXO or microsphere+CS-EPC-EXO suspension was injected via 30 G needles at two symmetric sites in the peri-infarct area immediately after MI.” has been changed as “20 μ L of microsphere+EPC-EXO or microsphere+CS-EPC-EXO suspension containing 20 μ g of EXOs (1mg/mL EXOs) was injected immediately via 30 G needles at two symmetric sites in the infarct area located at the cardiac base after MI.” in the revised manuscript (page 41 line 28-29 and Page 42 Line 1-2).

b. The volume of the injectate likely expanded to the entire infarct region on the anterior aspect of the heart. How did the investigators keep the injectate in the peri-infarct region?

Re: Indeed, as indicated by the reviewer, if we inject pure exosomes in PBS, they will expand quickly and disappear after 7 days. One of the main advantages of our microspheres encapsulation of exosomes is to increase the stay of the exosomes in the infarcted area for longer period. Our results showed that the fluorescence labeled exosomes were maintained in the infarction area for up to 21 days while the pure exosomes injection in PBS disappeared after 7 days (Figure 4A).

11. (a) The description of the preparation and characterization of the microspheres requires additional information. (b) Are figures 3B, 3D and 3F representative of multiple microscopic examinations? (c) In figure 3D the labelled EXOs could be only on the outside of the microspheres. Were they uniformly distributed? (d) Figure 3F does not adequately describe the EXO release from the microspheres. How many experiments are described in figure 3F? (e) What was the fluorescence intensity of PKH26 red-labeled EXOs (PKH26-EXO) at each time and was the supernatant intensity significantly greater after 14 days? The release of EXOs into PBS may not represent the release rate in vivo. (f) Further experiments in their mouse model are required to document duration of the EXO release in vivo.

a. The description of the preparation and characterization of the microspheres requires additional information.

Re: According to the reviewer’s suggestion, the sentences “Then, a water phase tube and an oil phase tube are connected to a microfluidic pressure pump. The prepared mixture solution was added dropwise to mineral oil by the microfluidic pressure pump and cured by cross-linking with UV light. The size and speed of the microsphere formation were observed under a single lens reflex camera (Nikon F3), and the flow rate of the water phase and oil phase solution were adjusted accordingly. Next, some microspheres were placed in cell culture plates, and the particle size, morphology and dispersion of the microspheres were observed under a light microscope (Leica, Germany), and the flow rate was further adjusted until the diameter of the microspheres was 30 μ m.” (Page 40 Line 13-20) have been added in the revised manuscript.

b. Are figures 3B, 3D and 3F representative of multiple microscopic examinations?

Re: Yes, five pictures were taken for each sample in this study, and no significant difference was observed among the five pictures, so representative picture was selected from 5 pictures.

In the methods section, the sentence “The morphology of the microspheres was observed using a scanning electron microscope (SEM, SU-8010; Hitachi, Japan). PKH26 was used to label EXOs, and fluorescein isothiocyanate (FITC) was used to label microspheres.” has been changed as “The morphology of the microspheres was observed using optical microscope (Leica, Japan) and scanning electron microscope (SEM, SU-8010; Hitachi, Japan). Furthermore, PKH26 was used to label EXOs, and fluorescein isothiocyanate (FITC) was used to label microspheres. The distribution of labeled EXOs inside the microsphere was observed using confocal laser scanning microscope (Leica TCS SP8, Germany). Five pictures were taken for each sample, and representative picture was selected.” in the revised manuscript. (Page 40 Line 22-27)

c. In figure 3D the labelled EXOs could be only on the outside of the microspheres. Were they uniformly distributed?

Re: Yes, Figure 3D was taken by scanning electron microscope, and the EXOs inside the microspheres could not be observed. However, Figure 3E was taken by a confocal laser scanning microscope, which revealed a cross section of the microspheres and showed uniform distribution of labeled EXOs inside the microsphere. According to reviewer’s comments, we have also added a video of the fluorescence scanning of one microsphere, which demonstrate the uniform distribution of the EXOs inside a microsphere (Video S1).

d. Figure 3F does not adequately describe the EXO release from the microspheres. How many experiments are described in figure 3F?

Re: The experiments were repeated 3 times. According to the reviewer’s suggestion, we have re-written the description of the EXO release from the microspheres represented in the Figure 3F.

In the results section, the sentences “As shown in Figure 3F, the measurement of the fluorescence intensity of PKH26 in the supernatant showed that the release rate of Microsphere-EXO was faster in the first 16 days with a linear trend, and started to slow down after 16 days. On day 20, the release rate of Microsphere-EXO reached 72.7%.” have been changed as “As shown in Figure 3F, the measurement of the fluorescence intensity of PKH26 in the supernatant showed that the release rate of Microsphere-EXO was faster in the first 16 days with a linear trend, especially on day 2, the release rate of exosomes was 18.6%. Then, the release rate started to slow down after 16 days (Day 16: 66.7%; Day 18: 70.7%). Finally, on day 20, the release rate of Microsphere-EXO reached 72.7%.” in the revised manuscript. (Page 12 Line 27-29 and Page 13 Line 1-2)

Furthermore, 3 experiments are described in Figure 3F, and n=3 has been added in Figure 3F.

e. What was the fluorescence intensity of PKH26 red-labeled EXOs (PKH26-EXO) at each time and was the supernatant intensity significantly greater after 14 days?

Re: As shown in FigureS3, the fluorescence intensity of the supernatant was the highest on the second day, which was mainly due to the rapid release of EXOs in the early stage. Then, the fluorescence intensity gradually increased from the 4th day to the 16th day, and began to decline after the 16th day. Since Gelma microsphere degrade 50% within 20 days³⁴, this might be the main reason for the increase of EXOs release amount in the microspheres from day 4.

In the results section, the sentences “Moreover, from the fluorescence intensity of the supernatant, it is clear to see that EXOs release amount was the highest on the 2nd day (Figure S3), which was mainly due to the rapid release of EXOs in the early stage, and the phenomenon that the fluorescence intensity gradually increased from the 4th day to the 16th day, and began to decline after the 16th day (Figure S3) might be due to the degradation of Gelma microspheres³⁴.” have been added in the revised manuscript. (Page 13 Line 3-7)

The newly added references in the revised manuscripts are as follows:

34 Zhang, Q. et al. Platelet lysate functionalized gelatin methacrylate microspheres for improving angiogenesis in endodontic regeneration. *Acta Biomaterialia* 136, 441-455, doi:10.1016/j.actbio.2021.09.024 (2021).

f. The release of EXOs into PBS may not represent the release rate in vivo. Further experiments in their mouse model are required to document duration of the EXO release in vivo.

Re: We agree with the reviewer. Therefore, we have quantitatively measured the release of EXOs *in vivo* according to the fluorescence intensity of PKH26 red-labeled EXOs in the heart. The results showed that the microspheres gradually released EXOs within 21 days. The new result has been added as Figure S11.

In the results section, the sentence “Quantitative analysis also confirmed significantly higher fluorescence intensity in microspheres group than that in PBS group (Figure S11).” has been added in the revised manuscript. (Page 15 Line 13-14)

In the methods section, the sentence “The release of EXOs *in vivo* was quantified according to fluorescence intensity change.” has been added in the revised manuscript. (Page 42 Line 9-10)

12. (a) The number of animals and specimens examined with each outcome described in figure 4 should be reported. (b) Differences between microsphere+EPC-EXO and microsphere+CS-EPC-EXO on ventricular function change with time following coronary ligation and should be evaluated with a time varying two-way analysis of variance reporting the overall effect (of time and group) and then specifying differences with a multiple range t test. (c) Heart function should be evaluated at multiple time intervals after the infarct and not just at 0 and 21 days. (d) Estimation of systolic and diastolic volumes should be reported as well as EF and FS. Were the differences in EF and FS due to differences in preload and afterload or were they due to differences in muscle contraction? Estimates of ventricular volumes from the echocardiographic measurements provide a guide to differences in the loading conditions which could have resulted from the differences in the cytokine release from the EXOs.

a. The number of animals and specimens examined with each outcome described in figure 4 should be reported.

Re: According to reviewer’s suggestion, the number of animals or specimens examined in each group (n=5) has been added in the caption of Figure 4.

In the methods section, the sentence “The number of animals in each group was 5.” (Page 41 Line 26-27) and “The number of tissue specimens to analyze in each group was 5.” (Page 43 Line 28-29) has been added in the revised manuscript.

b. Differences between microsphere+EPC-EXO and microsphere+CS-EPC-EXO on ventricular

function change with time following coronary ligation and should be evaluated with a time varying two-way analysis of variance reporting the overall effect (of time and group) and then specifying differences with a multiple range t test. c. Heart function should be evaluated at multiple time intervals after the infarct and not just at 0 and 21 days. Estimation of systolic and diastolic volumes should be reported as well as EF and FS.

Re: According to the reviewer's suggestion, heart function have been evaluated at 0, 7, 14 and 21 days (Figure S5 and S6), and the difference between microsphere+EPC-EXO and microsphere+CS-EPC-EXO on ventricular function change with time has been calculated with a multiple range t test in Figure S6. It is found that the ejection fraction (EF) and fractional shortening (FS) value of EPC-EXO, Microsphere+EPC-EXO and Microsphere+CS-EPC-EXO groups increased gradually with time, and the left ventricular end-diastolic volume (LVVD) and left ventricular end-systolic volume (LVVS) value decreased gradually with time. In contrast, the cardiac function of PBS group became worse with time. We also found that, although EPC-EXO alone has a better effect on improving cardiac function as compared to PBS group, the Microsphere+EPC-EXO group showed higher improvement, indicating the important role of microsphere encapsulation. Furthermore, by comparison of Microsphere+EPC-EXO with Microsphere+CS-EPC-EXO, the results confirmed the higher activity of CS-EPC-EXO in improving cardiac function of MI mice than EPC-EXO.

In the results section, the sentences "To evaluate the therapeutic function of the CS-EPC-EXO (Microsphere+CS-EPC-EXO), the heart function was first evaluated by echocardiography examination after 21 days post microspheres injection....." have been changed as "To evaluate the therapeutic function of the CS-EPC-EXO (Microsphere+CS-EPC-EXO), the heart function was first evaluated by echocardiography examination after 0, 7, 14 and 21 days post microspheres injection. The result showed that the EXOs and EXOs-containing microsphere groups significantly improved the cardiac function after myocardial infarction, while the cardiac function of PBS group declined gradually with time (Figure 4C and Figure S5 and Figure S6). Interestingly, from the specific values of left ventricular ejection fraction (EF), left ventricular fractional shortening (FS), left ventricular end-diastolic volume (LVVD) and left ventricular end-systolic volume (LVVS), it can be clearly seen that Microsphere+CS-EPC-EXO more significantly improved the cardiac systolic function of mice as compared with Microsphere+EPC-EXO (Figure 4D and Figure S6), which indicates that CS-EPC-EXO has higher therapeutic effect than EPC-EXO." in the revised manuscript. (Page 15 Line 21-29 and Page 16 Line 1)

d. Were the differences in EF and FS due to differences in preload and afterload or were they due to differences in muscle contraction? Estimates of ventricular volumes from the echocardiographic measurements provide a guide to differences in the loading conditions which could have resulted from the differences in the cytokine release from the EXOs.

Re: We believed the differences in EF and FS was due to the difference in muscle contraction. After EXOs treatment, the left ventricular end-diastolic volume (LVVD) and left ventricular end-systolic volume (LVVS) of the heart was significantly smaller than that of the PBS group, while EF and FS was higher than that of the PBS group, which was only possible by muscle contraction according to Franking-starling mechanism¹. We also agree that these differences

may be caused by cytokine release from EXOs.

Reference:

1 Jacob, R., Dierberger, B. & Kissling, G. FUNCTIONAL-SIGNIFICANCE OF THE FRANK-STARLING MECHANISM UNDER PHYSIOLOGICAL AND PATHOPHYSIOLOGICAL CONDITIONS. *European Heart Journal* 13, 7-14, doi:10.1093/eurheartj/13.suppl_E.7 (1992).

13. (a) In figures 4A and 4B were the differences between the groups significant? (b) In figure 4F how many hearts were evaluated in each group? (c) TUNEL staining (in figure 4H) may not accurately detect apoptotic cells. Was another index employed? (d) In figure 4I and 4J, how were clumps of smooth muscle cells differentiated from capillaries or arterioles? What were the criteria used to characterize labelled cell clusters as capillaries or arterioles? (e) How many slides were examined in each group and were they evaluated by a blinded observer?

a. In figures 4A and 4B were the differences between the groups significant?

Re: We have supplemented the quantitative statistics of EXOs release *in vivo* according to Figure 4A, and found that the difference between two groups were significant (Figure S11). However, in Figure 4B, the difference of the CD68 expression of was not significant between microspheres group and PBS group indicating no excessive inflammatory reaction in the myocardium after microsphere injection.

Therefore, in the results section, the sentence “Quantitative analysis also confirmed significantly higher fluorescence intensity in microspheres group than that in PBS group (Figure S11).” has been added in the revised manuscript. (Page 15 Line 13-14)

b. In figure 4F how many hearts were evaluated in each group?

Re: The number of hearts in each group is 5, and “The number of hearts = 5.” have been added in the Figure 4. (Page 18 Line 12)

c. TUNEL staining (in figure 4H) may not accurately detect apoptotic cells. Was another index employed?

Re: According to the reviewer’s suggestion, Caspase 3 staining has been used to further evaluated the apoptotic cells. The result showed that EXOs did significantly inhibit apoptosis of cells in heart. Especially the anti-apoptosis effect of Microsphere+CS-EPC-EXO was indeed higher than that of Microsphere+EPC-EXO (Figure S15).

In the results section, the sentences “Then, Caspase 3 staining was used to further evaluated the apoptotic cells. The results showed that the anti-apoptosis effect of Microsphere+CS-EPC-EXO was significantly higher than that of Microsphere+EPC-EXO (Figure S15), which was consistent with the trend of TUNEL.” have been added in the revised manuscript. (Page 16 Line 27-29 and Page 17 Line 1)

In the methods section, the sentences “Furthermore, TUNEL (Sigma Aldrich, USA) and Caspase 3 (Abcam) immunohistochemical staining were used to detect the apoptosis of cells in heart. Quantitative analysis of apoptotic cells was calculated according to the TUNEL positive cells and Caspase 3 positive cells.” have been added in the revised manuscript. (Page 42 Line 26-29)

d. In figure 4I and 4J, how were clumps of smooth muscle cells differentiated from capillaries or arterioles? What were the criteria used to characterize labelled cell clusters as capillaries or arterioles?

Re: Thank for the reviewer's suggestion. CD31 is a marker of endothelial cells, which mainly exists in the capillary or inner layer of arterioles¹, and α -SMA is a marker of smooth muscle cells, mainly located in the outer layer of arterioles². Therefore, α -SMA positive vessels are generally considered arterioles, while CD31 positive but α -SMA negative vessels are considered capillaries. In order to show the regeneration of capillaries and arterioles more clearly, we have supplemented CD31/ α -SMA co-localization staining images as shown in Figure S10.

In the results section, the sentences "From CD31/ α -SMA co-localization staining images, it is clear to see that Microsphere+CS-EPC-EXO most significantly enhanced the regeneration of capillaries and arterioles (Figure S10)." have been added in the revised manuscript. (Page 17 Line 6-8)

Reference:

- 1 Lee, C. et al. NEU1 Sialidase Regulates the Sialylation State of CD31 and Disrupts CD31-driven Capillary-like Tube Formation in Human Lung Microvascular Endothelia. *Journal of Biological Chemistry* 289, 9121-9135, doi:10.1074/jbc.M114.555888 (2014).**
- 2 Hansen-Smith, F., Egginton, S. & Hudlicka, O. Growth of arterioles in chronically stimulated adult rat skeletal muscle. *Microcirculation-London* 5, 49-59, doi:10.1080/713773811 (1998).**

e. How many slides were examined in each group and were they evaluated by a blinded observer?

Re: 5 slides have been examined in each group, and all the results were evaluated by blinded observers.

Therefore, n=5 has been added in the Figure 4.

The sentence "All the results were assessed blindly by three people in this study." has been added in Methods section in the revised manuscript (page 43 line 28).

14. The in vitro cell migration experiments may not reflect in vivo events. (a) Does CD34 and VEGFR2 co-localization staining accurately determine cell recruitment to the infarct region? Double stained cells may also be endogenous cells. (b) How many specimens were examined at 7 days after the infarct (figure 5C)? (c) Were the differences between microsphere+CS-EPC-EXO and microsphere+EPC-EXP statistically significant? (d) Does the assessment of Dio-EPCs injected into the tail vein predict the recruitment of the endogenous bone marrow cells to the infarct? (e) What is the time course of the homing of the Dio-EPCs to the infarct region? Previous studies have demonstrated that injecting EPCs into the circulation does not improve the recovery after a myocardial infarction in humans and the studies in animals are contradictory. Previous studies with labelled bone marrow progenitor cells demonstrated that enhancement of endogenous bone marrow and cardiac resident stem cell homing improves recovery from an infarct, but that intravenous cell injection had no benefits (*European Heart Journal* 2013;34:1157-1167).

a. The in vitro cell migration experiments may not reflect in vivo events. Does CD34 and VEGFR2 co-localization staining accurately determine cell recruitment to the infarct region? Double stained cells may also be endogenous cells.

Re: We agree with the reviewer, that the in vitro cell migration experiments may not reflect

the *in vivo* events. We used cell culture experiments to primarily assess the activity of CS-EPC-EXO in enhancing the migration ability of EPCs under the condition of glucose and oxygen deprivation, which may partially reflect the homing ability of the EPCs *in vivo*.

We also agree with the reviewer that the CD34 and VEGFR2 co-localization staining showed both recruited and endogenous cells. However, considering the lower concentration of the EPCs in bone marrow and labeling difficulty to visualize EPCs from bone marrow, a direct detection of the recruited EPCs from bone marrow might be difficult¹. Therefore, we first chose the injection of fluorescence labeled EPCs to evaluate the recruitment activity of the CE-EPC-EXO and confirmed its recruitment ability. We will consider reviewer's comments and try to further evaluate the recruitment of the EPCs from bone marrow in our future study.

Reference:

1 Aicher, A. et al. Assessment of the tissue distribution of transplanted human endothelial progenitor cells by radioactive labeling. *Circulation* 107, 2134-2139, doi:10.1161/01.Cir.0000062649.63838.C9 (2003).

b. How many specimens were examined at 7 days after the infarct (figure 5C)?

Re: For the results in Figure 5C, 5 specimens in each groups were used. It has been supplemented in the annotation.

In the results section, the sentence “(C) CD34+/VEGFR2+ immunofluorescence staining of EPCs in the peripheral area of myocardial infarction on day 7 after infarction. CD34+ (red), VEGFR2+ (green), DAPI (blue), CD34+/VEGFR2+ (orange), scale bar=50 μm” has been changed as “(C) CD34+/VEGFR2+ immunofluorescence staining of EPCs in the peripheral area of myocardial infarction on day 7 after infarction. CD34+ (red), VEGFR2+ (green), DAPI (blue), CD34+/VEGFR2+ (orange), scale bar=50 μm, specimens n=5.” in the revised manuscript (page 21 line 7).

c. Were the differences between microsphere+CS-EPC-EXO and microsphere+EPC-EXP statistically significant?

Re: As shown in quantitative analysis of Figure S14, the difference between Microsphere+CS-EPC-EXO and Microsphere+EPC-EXO was statistically significant.

According to reviewer's comments, in the Result section, the sentence “Quantitative analysis also showed that Microsphere+CS-EPC-EXO significantly promoted the expression of CD34 and VEGFR2 as compared with Microsphere+EPC-EXO (Figure S14).” has been added in the revised manuscript (page 20 line 4-6).

d. Does the assessment of Dio-EPCs injected into the tail vein predict the recruitment of the endogenous bone marrow cells to the infarct?

Re: The experiment of the tail vein injection of Dio-EPCs could only reflect the activity of the exosomes to recruit EPCs from the peripheral blood, but not direct from bone marrow. Studies have shown that endogenous bone marrow cells (EPCs) first entered the peripheral blood from the bone marrow, and then entered the myocardium from the peripheral blood^{1,2}. We think that, due to the lower concentration of the EPCs in bone marrow, a direct detection of the recruited EPCs from bone marrow might be difficult¹. Therefore, we first chose this method to evaluate the recruitment activity of the CE-EPC-EXO and confirmed its

recruitment ability. We will consider reviewer's comments and try to further evaluate the direct recruitment of the EPCs in our future study.

Reference:

1 Tepper, O. M. et al. Adult vasculogenesis occurs through in situ recruitment, proliferation, and tubulization of circulating bone marrow-derived cells. *Blood* 105, 1068-1077, doi:10.1182/blood-2004-03-1051 (2005).

2 Aicher, A. et al. Assessment of the tissue distribution of transplanted human endothelial progenitor cells by radioactive labeling. *Circulation* 107, 2134-2139, doi:10.1161/01.Cir.0000062649.63838.C9 (2003).

e. What is the time course of the homing of the Dio-EPCs to the infarct region?

Re: First of all, our study observed the recruitment of Dio-EPCs after the injection for 7 days, and we confirmed that Dio-EPCs could indeed be recruited to the myocardial infarction area for 7 days. Therefore, we believe that Dio-EPC may be recruited to the infarcted area within 7 days.

f. Previous studies have demonstrated that injecting EPCs into the circulation does not improve the recovery after a myocardial infarction in humans and the studies in animals are contradictory. Previous studies with labelled bone marrow progenitor cells demonstrated that enhancement of endogenous bone marrow and cardiac resident stem cell homing improves recovery from an infarct, but that intravenous cell injection had no benefits (*European Heart Journal* 2013;34:1157-1167).

Re: We thank the reviewer for these comments. The beneficial functions of stem cells decline with age, limiting their therapeutic efficacy for myocardial infarction (MI)¹. This may be the main reason why researchers sometimes obtain inconsistent experimental results when using cell therapy. Although the bioactivity of exosomes will also decrease significantly with cell aging², we demonstrated that CS ionic solution activates cells to secrete highly active exosomes, which might provide a possibility to overcome the problem associated with cell therapy, although further comparative studies are required.

Reference:

1 Zhang, Y. L. et al. Macrophage migration inhibitory factor rejuvenates aged human mesenchymal stem cells and improves myocardial repair. *Aging-Us* 11, 12641-12660, doi:10.18632/aging.102592 (2019).

2 Ahmadi, M. & Rezaie, J. Ageing and mesenchymal stem cells derived exosomes: Molecular insight and challenges. *Cell Biochemistry and Function* 39, 60-66, doi:10.1002/cbf.3602 (2021).

15. (a) For the results presented in figure 6, please describe the culture condition of the EPCs with and without CS stimulation. (b) The sequencing studies were confirmed with only 3 studies using qRT-PCR. Is that a sufficient number of studies to confirm the up-regulation of the 5 genes? (c) Are the effects in HUVEC cells the same as in in vivo heart tissue or even in vitro EPCs? (d) Angiogenesis was not established in HUVEC cells since capillaries and arterioles were not formed. The evaluation of the migration and vascular tube formation of HUVECs under glucose-oxygen deprivation does not reflect the effects of EXO in vivo.

a. For the results presented in figure 6, please describe the culture condition of the EPCs with and

without CS stimulation.

Re: According to reviewer's suggestion, in the Results section, the sentence "EPCs were stimulated with diluted CS ion solution (1/128, silicate ions) or conventional culture medium for 48 h to obtain CS-EPC-EXO and EPC-EXO." has been added in the revised manuscript. (Page 22 Line 4-5)

Furthermore, in the methods section, the sentence "EPCs were stimulated with diluted silicate ion solution (1/128) for 48 h, and EXOs released from EPCs in the supernatant were isolated by ultracentrifugation." has been changed as "EPCs were stimulated with diluted silicate ion solution in culture medium (1/128) or conventional culture medium for 48 h, and EXOs released from EPCs in the supernatant were isolated by ultracentrifugation (CS-EPC-EXO: stimulated with silicate ion solution; EPC-EXO: stimulated with conventional culture medium)." in the revised manuscript. (Page 36 Line 23-26)

b. The sequencing studies were confirmed with only 3 studies using qRT-PCR. Is that a sufficient number of studies to confirm the up-regulation of the 5 genes?

Re: We thank for the reviewer's suggestion. We repeated the PCR experiment here and increased the number of parallel samples to 5 in Figure 6C-E and Figure S12.

c. Are the effects in HUVEC cells the same as in *in vivo* heart tissue or even *in vitro* EPCs? d. Angiogenesis was not established in HUVEC cells since capillaries and arterioles were not formed. The evaluation of the migration and vascular tube formation of HUVECs under glucose-oxygen deprivation does not reflect the effects of EXO *in vivo*.

Re: Regarding angiogenesis, we found the stimulation effects of our CS-EXO-EPC for *in vitro* HUVECs cells and *in vivo* blood vessel formation are consistent. In addition, the stimulation effects of CS-EXO-EPC on the migration of both HUVECs and EPCs *in vitro* are similar.

We agree with the reviewer that the migration and vascular tube formation assay of HUVECs do not fully reflect the formation of capillaries and arterioles *in vivo*, they just reflect the bioactivity of CS-EXO-EPC in activating HUVECs for angiogenesis (we also analyzed expression of VEGFA, eNOS and other angiogenic related factors in HUVECs by PCR, Figure 2L-N). However, these results are the bases for our *in vivo* experiment. Based on these preliminary evaluations, we further designed *in vivo* study, and proof the activity of CS-EXO-EPC in promoting both capillaries and arterioles regeneration *in vivo* (Figure S10).

16. The angiogenic effects of microsphere-CS-EPC-EXOs were greater than microsphere-EPC-EXOs, but the differences were small. How do these interventions compare to angiogenic enhancement by cell, gene or protein therapy? Were the benefits greater in these studies than those obtained by injecting EPCs or EXOs (with or without microspheres) into the infarct region?

Re: According to the literature, after injecting MSCs cells or human cord blood cells, the number of new blood vessels was 6/HPF~20/HPF (high-power field: HPF)^{1,42}. Furthermore, many studies have revealed that the therapeutic effect of cell derived exosomes is almost the same as that of cell therapy^{42,43}. A study has also shown that the angiogenic effect of cell therapy is significantly higher than that of gene or protein therapy⁴⁴. In contrast, in our experiment, the number of new vessels in CS-EPC-EXO group was 40/HPF. Therefore, we think that the highly bioactive exosomes prepared by CS stimulation may be more effective in promoting angiogenesis in the myocardial infarction region as compare to cell, gene or protein

therapy, although further comparative experiments are required to confirm this statement, and we will consider this in our future study.

Regarding the comparison between the microsphere encapsulated exosomes and without microsphere encapsulation, according to reviewer's comments, we have supplemented the experiment of EPC-EXO injection without microsphere encapsulation, and proved that the therapeutic effect of Microsphere+EPC-EXO was clearly higher than that of EPC-EXO.

Therefore, in the Discussion section, the sentences "Especially in promoting angiogenesis, the effect of CS induced highly active exosomes is not only higher than that of the exosomes without CS stimulation, it seems also more effective than other approaches for the treatment of myocardial infarction such as cell, gene or protein therapies reported in the literature⁴²⁻⁴⁴, but this needs to be proofed by careful comparative studies." has been added in the revised manuscript. (Page 29 Line 22-25)

The newly added references in the revised manuscripts are as follows:

42 Rani, S., Ryan, A. E., Griffin, M. D. & Ritter, T. Mesenchymal Stem Cell-derived Extracellular Vesicles: Toward Cell-free Therapeutic Applications. *Molecular Therapy* 23, 812-823, doi:10.1038/mt.2015.44 (2015).

43 Bian, S. Y. et al. Extracellular vesicles derived from human bone marrow mesenchymal stem cells promote angiogenesis in a rat myocardial infarction model. *Journal of Molecular Medicine-Jmm* 92, 387-397, doi:10.1007/s00109-013-1110-5 (2014).

44 Shyu, K. G., Wang, B. W., Hung, H. F., Chang, C. C. & Shih, D. T. B. Mesenchymal stem cells are superior to angiogenic growth factor genes for improving myocardial performance in the mouse model of acute myocardial infarction. *Journal of Biomedical Science* 13, 47-58, doi:10.1007/s11373-005-9038-6 (2006).

Reference:

1 Ma, N. et al. Human cord blood cells induce angiogenesis following myocardial infarction in NOD/scid-mice. *Cardiovascular Research* 66, 45-54, doi:10.1016/j.cardiores.2004.12.013 (2005).

17. (a) The authors demonstrated that CS stimulation of EPCs produced more particles (exosomes?) than unstimulated EPCs. When the concentration of angiogenic cytokines (VEGF, eNOS, HGF, IGF and SDF) is divided by the number of particles, were there more cytokines/particle after CS stimulation or only more particles? Are the CS stimulated EXOs more biologically active or only at a higher concentration? (b) As previously noted, CS stimulation may have advantages and disadvantages if CS induces adverse side effects. To answer this question, the authors should compare CS-EPC-EXO to EPC-EXO in an in vivo model (such as their mouse infarct model) and to the results obtained with stem cells and stem cell derived EXOs. (c) Other authors have reported that other stimuli increase both the secretion and the bioactivity, but the two properties must be studied separately. (d) The authors should also indicate what they believe would be the effects of CS injection into the infarcted heart. Previous studies have indicated that most stimulants enhance angiogenesis and improve ventricular function. However, some agents which stimulate inflammation have been found to induce heart failure and death.

a. The authors demonstrated that CS stimulation of EPCs produced more particles (exosomes?) than

unstimulated EPCs. When the concentration of angiogenic cytokines (VEGF, eNOS, HGF, IGF and SDF) is divided by the number of particles, were there more cytokines/particle after CS stimulation or only more particles? Are the CS stimulated EXOs more biologically active or only at a higher concentration?

Re: According to the reviewer's comments, we have measured the mass of cytokine (VEGF, eNOS, HGF, IGF and SDF) and the total number of particles in 1 μ g exosomes by ELISA and NTA, respectively. Then, we calculated the cytokine content in one exosome particle (Calculation formula: The cytokine content in one particle = The mass of cytokine/The total number of particles). The results showed that the content of angiogenic cytokines in each CS-EPC-EXO particle was indeed higher than that in each EPC-EXO particle (Table S5). This result indicates that CS stimulation not only increased the number of exosomes secretion, but also enhanced the biological activity of exosomes by increasing the cytokines contents in each exosome particle.

In the methods section, the sentence "Furthermore, the cytokine content in each EXOs particle was calculated according to the mass of cytokine (VEGF, eNOS, HGF, IGF and SDF) measured by ELISA and the total number of particles measured by NTA. Calculation formula: $C_{EXOs} = M_{cyto} / N_{EXOs}$; where C_{EXOs} is the mass of cytokine in one EXOs particle, M_{cyto} is the mass of cytokines, N_{EXOs} is the total number of particles." has been added in the revised manuscript. (Page 39 Line 19-22)

In the results section, the sentence "...we assessed the expression levels of key angiogenic factors (VEGFA, eNOS, HGF, IGF-1 and SDF-1) in CS-EPC-EXO and EPC-EXO by ELISA (Figure 1K). Interestingly, the content of angiogenic factors (VEGFA, eNOS, HGF, IGF-1) in CS-EPC-EXO was indeed significantly higher than that in EPC-EXO." have been changed as "...we assessed the expression levels of key angiogenic factors (VEGFA, eNOS, HGF, IGF-1 and SDF-1) in CS-EPC-EXO and EPC-EXO by ELISA (Figure 1K), and calculated the content of angiogenic factors in a single EXOs particle (Table S5). Interestingly, the content of angiogenic factors (VEGFA, eNOS, HGF, IGF-1) in CS-EPC-EXO was indeed significantly higher than that in EPC-EXO." in the revised manuscript. (Page 7 Line 12-16)

b. As previously noted, CS stimulation may have advantages and disadvantages if CS induces adverse side effects. To answer this question, the authors should compare CS-EPC-EXO to EPC-EXO in an in vivo model (such as their mouse infarct model) and to the results obtained with stem cells and stem cell derived EXOs.

Re: In our previous studies, we have evaluated the safety of the CS ionic solution in the treatment of myocardial infarction and did not observe any adverse side effects. However, we agree with the reviewer that, although we have confirmed the higher therapeutic effect of CS-EPC-EXO as compared to EPC-EXO, we should further investigate the possible side effects in the treatment using CS-EPC-EXO and compared with EPC-EXO and other stem cell derived EXOs. So, we will consider this in our future research plan.

c. Other authors have reported that other stimuli increase both the secretion and the bioactivity, but the two properties must be studied separately.

Re: We agree with the reviewer, for further optimization of the CS stimulation in bioactive exosomes preparation, determination of optimal stimulation of both secretion amount and

bioactivity is important, which requires separate investigation. We will consider reviewer's good suggestion in our future experimental design.

d. The authors should also indicate what they believe would be the effects of CS injection into the infarcted heart. Previous studies have indicated that most stimulants enhance angiogenesis and improve ventricular function. However, some agents which stimulate inflammation have been found to induce heart failure and death.

Re: Thank for reviewer's suggestion. In our study, we did not observe inflammatory reaction in the treatment of Microsphere+CS-EPC-EXO. Furthermore, in our previous study, CS injection did not cause any inflammatory reaction, and in fact it even showed inhibitory effect on inflammation⁵⁸. Indeed, in the literature, some researchers have found that some exosomes derived from adipocytes could cause early inflammatory reaction to promote vascularization¹, although these exosomes are usually not used for myocardial repair.

According to reviewer's comments, in the Discussion section, the sentences "CS is a unique biomaterial, which has good anti-inflammatory and vascular regeneration effects^{57,58}." have been added in the revised manuscript. (Page 30 Line 28-29)

The newly added references in the revised manuscripts are as follows:

57 Wang, X. T. et al. Chitosan/Calcium Silicate Cardiac Patch Stimulates Cardiomyocyte Activity and Myocardial Performance after Infarction by Synergistic Effect of Bioactive Ions and Aligned Nanostructure. *Acs Applied Materials & Interfaces* 11, 1449-1468, doi:10.1021/acsami.8b17754 (2019).

58 Que, Y. et al. Silicate ions as soluble form of bioactive ceramics alleviate aortic aneurysm and dissection. *Bioactive Materials*, doi:https://doi.org/10.1016/j.bioactmat.2022.07.005 (2022).

Reference:

1 Chen, B. et al. Exosomes Are Comparable to Source Adipose Stem Cells in Fat Graft Retention with Up-Regulating Early Inflammation and Angiogenesis. *Plastic and Reconstructive Surgery* 144, 816E-827E, doi:10.1097/prs.0000000000006175 (2019).

18. (a) The microsphere encapsulated EPC-EXO may have prolonged the duration of the EXO in the infarct region in the infarct mouse model, but the duration of the EXO secretion was not evaluated *in vivo*. (b) The microspheres alone may also have induced angiogenesis and a control group with only microspheres without EPC-EXOs was not studied. (c) What is the time course of the benefit of these interventions? Will the benefit last more than 21 days?

a. The microsphere encapsulated EPC-EXO may have prolonged the duration of the EXO in the infarct region in the infarct mouse model, but the duration of the EXO secretion was not evaluated *in vivo*.

Re: According to the reviewer's suggestion, we have supplemented quantitative analysis the secretion from microspheres and internalization of EXOs by cells *in vivo*.

First, we quantitatively counted the secretion of EXOs *in vivo* according to the PKH26 intensity of fluorescence, and found the microspheres could continue to release EXOs after 21 days (Figure S11). Second, we found that EXOs were taken by myocardial cells, endothelial cells and fibroblasts, which also proved that EXOs were released from microspheres (Figure S4).

In the results section, the sentence “Quantitative analysis also confirmed significantly higher fluorescence intensity in microspheres group than that in PBS group (Figure S11).” has been added in the revised manuscript. (Page 15 Line 13-14)

In the methods section, the sentence “The release of EXOs *in vivo* was quantified according to fluorescence intensity change.” has been added in the revised manuscript. (Page 42 Line 9-10)

In the results section, the new results have been added in Figure S4 and the sentences “Furthermore, from the immunofluorescence staining images of CTNT, CD31 and Vimentin, it is clear to see that CS-EPC-EXO were co-localized with these cells, indicating the up-take of the exosomes by cardiomyocytes, endothelial cells and fibroblasts in infarct heart (Figure S4).” have been added in the revised manuscript. (Page 20 Line 16-18)

In the methods section, the sentences “Evaluation of the internalization of EXOs by cells *in vivo*. Male C57BL/6 mice were used to evaluate the internalization of EXOs *in vivo*. First, 20μL of microsphere+CS-EPC-EXO (PKH26) (1mg/mL EXOs) was injected immediately after LAD ligation via 30 G needles at two symmetric sites in the infarct area located at the cardiac base. Mice were killed on day 3, and the hearts paraffin sections were stained with immunofluorescence staining (cardiomyocytes (CTNT, 1:500, Abcam), endothelial cells (CD31, 1:500, Servicebio) and fibroblasts (Vimentin, 1:500, Abcam)).” have been added in the revised manuscript. (Page 43 Line 11-17)

b. The microspheres alone may also have induced angiogenesis and a control group with only microspheres without EPC-EXOs was not studied.

Re: According to reviewer’s suggestion, microspheres alone have been added to evaluate the angiogenic capacity in the area of myocardial infarction (Figure S10). It could be found that the angiogenic capacity of microspheres alone was equivalent to that of PBS. This result indicates that CS-EPC-EXO was the key to promote angiogenesis in the myocardial infarction area.

In the results section, the sentence “In addition, since the microspheres alone did not significantly promote angiogenesis, it was further confirmed that CS-EPC-EXO was the key activator to induce angiogenesis (Figure S10).” has been added in the revised manuscript. (Page 17 Line 8-10)

In the methods section, the sentence “Furthermore, microspheres alone were used as control group to observe the effect of promoting angiogenesis.” has been added in the revised manuscript. (Page 41 Line 27-28)

c. What is the time course of the benefit of these interventions? Will the benefit last more than 21 days?

Re: Our results showed that the release of EXOs from microspheres in infarct area could last 21 days after injection, so we believed that the benefits of EXOs for myocardial infarction treatment would also last for more than 21 days. We thank the reviewer to raise this issue, which is very important for clinical applications, and we will consider to evaluate the long-term effect of our approach.

19. The authors evaluated an innovative (but complex) intervention with microsphere encapsulated CS stimulated EPC derived EXOs. The results are encouraging. However, (a) the authors should

place their results in context of other studies of cells, genes, proteins and extracellular vesicles injected into the infarct region after coronary ligation. (b) In addition, the authors need to include additional controls for their studies.

Re: According to the literature, after injecting MSCs cells or human cord blood cells, the number of new blood vessels was 6/HPF~20/HPF (high-power field: HPF)^{1,42}. Furthermore, many studies have revealed that the therapeutic effect of cell derived exosomes is almost the same as that of cell therapy^{42,43}. A study has also shown that the angiogenic effect of cell therapy is significantly higher than that of gene or protein therapy⁴⁴. In contrast, in our experiment, the number of new vessels in CS-EPC-EXO group was 40/HPF. Therefore, we think that the highly bioactive exosomes prepared by CS stimulation may be more effective in promoting angiogenesis in the myocardial infarction region as compare to cell, gene or protein therapy, although further comparative experiments are required to confirm this statement, and we will consider this in our future study.

Regarding the comparison between the microsphere encapsulated exosomes and without microsphere encapsulation, according to reviewer's comments, we have supplemented the experiment of EPC-EXO injection without microsphere encapsulation, and proved that the therapeutic effect of Microsphere+EPC-EXO was clearly higher than that of EPC-EXO.

Therefore, in the Discussion section, the sentences "Especially in promoting angiogenesis, the effect of CS induced highly active exosomes is not only higher than that of the exosomes without CS stimulation, it seems also more effective than other approaches for the treatment of myocardial infarction such as cell, gene or protein therapies reported in the literature⁴²⁻⁴⁴, but this needs to be proofed by careful comparative studies." has been added in the revised manuscript. (Page 29 Line 22-25)

The newly added references in the revised manuscripts are as follows:

42 Rani, S., Ryan, A. E., Griffin, M. D. & Ritter, T. Mesenchymal Stem Cell-derived Extracellular Vesicles: Toward Cell-free Therapeutic Applications. *Molecular Therapy* 23, 812-823, doi:10.1038/mt.2015.44 (2015).

43 Bian, S. Y. et al. Extracellular vesicles derived from human bone marrow mesenchymal stem cells promote angiogenesis in a rat myocardial infarction model. *Journal of Molecular Medicine-Jmm* 92, 387-397, doi:10.1007/s00109-013-1110-5 (2014).

44 Shyu, K. G., Wang, B. W., Hung, H. F., Chang, C. C. & Shih, D. T. B. Mesenchymal stem cells are superior to angiogenic growth factor genes for improving myocardial performance in the mouse model of acute myocardial infarction. *Journal of Biomedical Science* 13, 47-58, doi:10.1007/s11373-005-9038-6 (2006).

Reference:

1 Ma, N. et al. Human cord blood cells induce angiogenesis following myocardial infarction in NOD/scid-mice. *Cardiovascular Research* 66, 45-54, doi:10.1016/j.cardiores.2004.12.013 (2005).

REVIEWER COMMENTS

Reviewer #1 (Remarks to the Author):

The reviewer's concerns were addressed.

Reviewer #2 (Remarks to the Author):

The authors have scholarly addressed the various concerns and definitely improved the manuscript. I still have two issues:

The first pertains to the difference between the Exo-treated groups. The manuscript states that "Microsphere+CS-EPCEXO more significantly improved the cardiac systolic function of mice as compared with Microsphere+EPC-EXO (Figure 4D and Figure S6)...". However, looking at Figure 4D, the comparison is only between the microsphere+Exo and the microsphere+EXO+silicate ion stimulation groups and there is no significant difference between them; the only observation is that there are higher numerical values for ejection fraction and fractional shortening in the silicate ion stimulation group compared with the PBS and EXO-only groups. It might thus be worth discussing whether calcium silicate ion stimulation of exosomes has an added value over conventional exo stimulation as long as exosomes are embedded in microspheres.

A second concern relates to Figure S3. One would have rather expected a punctuate pattern of exo expression in the different cell types instead of these relatively large red spots. Could some leakage of the dye have happened and confuse the interpretation of the images that do not look very convincing? At best, injection of a control (dye alone) should have been done. Otherwise, I would suggest to temper the assumption that "CS-EPC-EXO were co-localized with these cells, indicating the up-take of the exosomes by cardiomyocytes, endothelial cells and fibroblasts".

Reviewer #3 (Remarks to the Author):

The authors have provided an extensive response to the comments of the reviewers and provided new information which has been added to the manuscript. However, as is frequently the case, the new information also raises new questions.

The authors may wish to provide the rationale for their terminology.

1. They have now provided more information about the cells they termed endothelial progenitor cells. These cells appear to be early outgrowth cells (5-7 days, EOCs) which are characterized by CD133 (hematopoietic marker) and CD14 (monocyte marker). Late outgrowth cells are called endothelial colony-forming cells (ECFCs) which lose these markers and express CD34 and vascular endothelial growth factor receptor 2 (VEGFR2). Early outgrowth cells (EOCs) do not form blood vessels, but they can release various pro-angiogenic molecules. The authors should indicate the subgroup of their cells. Throughout their paper (Introduction, Results and Discussion) the authors assert that EPCs induce angiogenesis. They should revise these sections to indicate that ECFCs participate in neovascularization, however, the cells which they employed were EOCs and not ECFCs and were unlikely to participate in new blood vessel formation.
2. In addition, their figure S1 indicates the variety of cell types found in their mixture of cells. They should indicate the variation associated with their estimate in their figure and text. Which subgroups of cells were found in their mixture? Most must have been early outgrowth cells (EOCs). However, they may have a mixture of EOCs and ECFCs. Please specify the mean and variation of the cell types.
3. The authors have provided more information about their exosomes. However, these particles have a range of sizes and contents. Recent Consensus Statements have recommended that these particles be called extracellular vesicles (Journal of Extracellular Vesicles 2018 (7), 1535750 <https://doi.org/10.1080/20013078.2018.1535750>). In addition, most authors have noted that the diameter of EVs depends on the detection method employed. Therefore, the consistency of these particles may not correspond to previous reports employing different techniques. Extracellular vesicles can include small EVs (40-150 nm, large EVs (100-1000 nm), extracellular autophagic

vesicles (40-1000 nm) and apoptotic vesicles (100-1000 nm). What variety of particles were employed in this study? The authors should consider using the term extracellular vesicles rather than exosomes.

4. The authors provided an indication of the contents of the exosomes. Could they also add the variation of the measurements and how many measurements were made? They should also comment on the significance of this variation.

5. In figure S4, the authors show co-localization of PKH26 labeled EXOs and endothelial cells, cardiomyocytes and fibroblasts. Could they also indicate the number of slides evaluated per group and the mean and variation of co-localized cells per high powered field?

6. The authors agree that many mRNA species were likely upregulated by the exosomes. They should not state that one is the "strongest angiogenesis promoting miRNA" unless they have performed comparative studies.

7. The authors state: "In this study, 20 μ L solution contains 20 μ g of EXOs. According to reviewer's comments, we have calculated the number of EXOs particles in 1 μ g EXOs, which contains 2.44*10⁹ particles. Therefore, 48.8*10⁹ particles have been injected into the myocardial infarction site." However, could they provide the variation associated with their estimate?

8. In their introduction, the authors state that previous studies only evaluated the therapeutic effect of exosomes for 72 hours. They cannot conclude that the effects did not persist beyond 72 hours if the studies did not investigate the effects beyond 72 hours.

9. The authors state: "we have applied for two patents on silicate bioceramic ion solutions for the treatment of myocardial infarction." Although those patents do not involve exosomes, they do represent competing interests and should be disclosed.

10. The authors agree that their method of stimulating exosome release from cells is one of many approaches which may be beneficial. They state: "more careful comparative studies will be required to evaluate the applicability of the CS-EPC-EXO exosomes." That sentence should be added to their discussion rather than the sentence: "CS is a unique biomaterial, which has good anti-inflammatory and vascular regeneration effects." Their paper should provide a balanced view.

The also state: "more careful comparative studies will be required to evaluate the applicability of the CS-EPC-EXO exosomes." This sentence should be added to their discussion.

11. For both the size and purity of the exosomes, the authors should indicate the variability of their measurements. The authors report the purity of commercial kits as >1.5*10⁹. Since they are unable to statistically compare their results to those of the commercial kits, they cannot determine whether their purity is equivalent. All they can do is to report the numbers and they should not include their comments on the comparison since no comparison was made.

12. The authors state: "We agree with the reviewer that the in vitro activity effect of the exosomes will be different than the in vivo one." This statement should be added to their discussion.

13. They also state: "We selected the primary doses of the exosomes for in vivo experiment based on the literature³⁵." However, reference 35 is a careful review of the literature and Kennedy and colleagues concluded: "Two out of the 12 in vivo studies [24,32] failed to report the dose of EVs used; out of the remaining studies, variations in isolation method, dosing unit (particle number, μ g protein or number of parent cells from which EVs were sourced), animal model (mouse or rat) and route of administration (intramyocardial injection; i.m.; or tail vein; t.v.) render all studies incomparable to each other." How did the authors derive their dose of exosomes? Also note that Kennedy and colleagues employ the term extracellular vesicles rather than exosomes.

14. The authors employed echocardiography to calculate the ventricular volumes. This technique assumes that the heart is spherical and then transform the systolic and diastolic diameter measurements to volumes. In the infarct model, the heart is not spherical and therefore the volumes become less reliable. The best way to measure volumes in this animal model is by conductance. The authors may wish to mention this minor concern in their discussion.

15. The statistical results section states: "One-way ANOVA and post hoc Bonferroni tests were used to compare differences between more than two groups." Time related differences between groups (such as figures S6 and S11) must account for multiple testing. Either a two-way time varying ANOVA or an analysis of covariance should be performed to compare groups. Then the authors should report the effects of time and group assignment as well as the interactive effect. If a significant effect is found, they can employ multiple range t tests to specify the differences.

16. The authors state: "We agree with the reviewer, that the in vitro cell migration experiments may not reflect the in vivo events." This statement should be added to their manuscript.

17. From the data presented in figure 4J, Immunofluorescence staining of capillaries and small arteries in the peripheral area of myocardial infarction on day 21 after infarction was greater in microsphere+CS-EPC-EXO than microsphere+EPC-EXO. The number (40 vessels/high powered field in the periphery of the infarct region) is highly dependent on the protocol employed and the conduct of the experiments. The authors state: "Especially in promoting angiogenesis, the effect of CS induced highly active exosomes is not only higher than that of the exosomes without CS stimulation, it seems also more effective than other approaches for the treatment of myocardial infarction such as cell, gene or protein therapies reported in the literature." The second half of that sentence must be removed. The authors cannot compare their results to other experimental preparations. More extensive comparative studies are required to confirm their statement.

In summary, the authors have provided a great deal of additional, important information. Further clarifications would enhance the value of the paper for the readers.

Reviewer #2 (Remarks to the Author):

The authors have scholarly addressed the various concerns and definitely improved the manuscript. I still have two issues:

1. The first pertains to the difference between the Exo-treated groups. The manuscript states that "Microsphere+CS-EPC-EXO more significantly improved the cardiac systolic function of mice as compared with Microsphere+EPC-EXO (Figure 4D and Figure S6)...". However, looking at Figure 4D, the comparison is only between the microsphere+Exo and the microsphere+EXO+silicate ion stimulation groups and there is no significant difference between them; the only observation is that there are higher numerical values for ejection fraction and fractional shortening in the silicate ion stimulation group compared with the PBS and EXO-only groups. It might thus be worth discussing whether calcium silicate ion stimulation of exosomes has an added value over conventional exo stimulation as long as exosomes are embedded in microspheres.

Re: We thank the reviewer for indicating this issue. Indeed, we did not add the statistical comparison between Microsphere+CS-EPC-EXO and Microsphere+EPC-EXO groups. According to reviewer's comment, we have performed statistical analysis of the comparison between other groups (Sham, PBS, and Microsphere+EPC-EXO) and Microsphere+CS-EPC-EXO group in Figure 4D and other Figures such as Figure 5 and Figure S6-S12 and Figure S15-S16 in the manuscript. The results showed that statistically the therapeutic effect of Microsphere+CS-EPC-EXO was significantly higher than that of Microsphere+EPC-EXO. This result indicates that the EXOs stimulated by calcium silicate ions were indeed superior to conventional EXOs.

We have supplemented the statistical differences between Microsphere+CS-EPC-EXO group and other groups (Sham, PBS, and Microsphere+EPC-EXO) in Figure 4, Figure 5, Figure S6-S12 and Figure S15-S16.

Furthermore, we have emphasized the statistical significant difference between Microsphere+CS-EPC-EXO group and Microsphere+EPC-EXO group in the revised manuscript by adding the sentence "it can be clearly seen that Microsphere+CS-EPC-EV more significantly improved the cardiac systolic function of mice as compared with Microsphere+EPC-EV (Figure 4D and Figure S7), which indicates that CS-EPC-EV has higher therapeutic effect than EPC-EV." in Page 16 Line 6-9.

2. A second concern relates to Figure S3. One would have rather expected a punctuate pattern of exo expression in the different cell types instead of these relatively large red spots. Could some leakage of the dye have happened and confuse the interpretation of the images that do not look very convincing? At best, injection of a control (dye alone) should have been done. Otherwise, I would suggest to temper the assumption that "CS-EPC-EXO were co-localized with these cells, indicating the up-take of the exosomes by cardiomyocytes, endothelial cells and fibroblasts".

Re: According to the reviewer's suggestion, we have conducted a new experiment by the injection of dye as a control group, and the new result has been added in Figure S5A. The results showed that injection of PKH26 dye resulted in the distribution of red colored dye in all the area of the sections. Therefore, we speculated that there might be no leakage of dye from cells, and the observation of large red dots may be due to the aggregation of many Microsphere+CS-EPC-EV (PKH26) in cells.

In the Results part, the sentence "Furthermore, from the immunofluorescence staining images

of CTNT, CD31 and Vimentin, it is clear to see that CS-EPC-EV were co-localized with these cells, indicating the up-take of the extracellular vesicles by cardiomyocytes, endothelial cells and fibroblasts in infarct heart (Figure S5).” have been changed as “Furthermore, from the immunofluorescence staining images of CTNT, CD31 and Vimentin, it is clear to see that CS-EPC-EV were co-localized with cardiomyocytes, endothelial cells and fibroblasts, although many large red spots were found in all the groups. In contrast, injection of PKH26 dye directly marked all the area of the sections (Figure S5A), which indicates that PKH26 dye could label all the cells on the section. According to these results, we speculated that these large red dots observed in the image may be the aggregation of many Microsphere+CS-EPC-EV (PKH26).” in the revised manuscript. (Page 20 Line 19-24)

In the Materials and Methods part, the sentence “PKH26 dye injected directly was used as a control group.” has been added in the revised manuscript. (Page 44 Line 3)

Reviewer #3 (Remarks to the Author):

The authors have provided an extensive response to the comments of the reviewers and provided new information which has been added to the manuscript. However, as is frequently the case, the new information also raises new questions.

The authors may wish to provide the rationale for their terminology.

1. They have now provided more information about the cells they termed endothelial progenitor cells. These cells appear to be early outgrowth cells (5-7 days, EOCs) which are characterized by CD133 (hematopoietic marker) and CD14 (monocyte marker). Late outgrowth cells are called endothelial colony-forming cells (ECFCs) which lose these markers and express CD34 and vascular endothelial growth factor receptor 2 (VEGFR2). Early outgrowth cells (EOCs) do not form blood vessels, but they can release various pro-angiogenic molecules. The authors should indicate the subgroup of their cells. Throughout their paper (Introduction, Results and Discussion) the authors assert that EPCs induce angiogenesis. They should revise these sections to indicate that ECFCs participate in neovascularization, however, the cells which they employed were EOCs and not ECFCs and were unlikely to participate in new blood vessel formation.

Re: We thank for the reviewer’s kind reminder. According to the reviewer’s comments, we have carefully read literature and found that EPCs including two main subgroups such as EOCs and ECFCs are characterized by many markers. In most references^{7,8,37-39} EOCs were characterized by CD133, which could also be labeled by CD34 and VEGFR2 (CD133+/CD34+/VEGFR2+), and ECFCs were characterized by CD34 and VEGFR2, which could not be labeled by CD133 (CD133-/CD34+/VEGFR2+). Furthermore, a reference⁴⁰ showed a subgroup of EPCs, which was characterized by CD133 and VEGFR2, and could not be labeled by CD34 (CD133+/CD34-/VEGFR2+). This subgroup cells of EPCs are also considered as ECFCs, and are functionally potent with respect to homing and vascular repair. Therefore, we have performed the flow cytometry analysis with triple fluorescent markers and found that CD133+/CD34+/VEGFR2+ positive cells were 71.61±2.84%, CD133-/CD34+/VEGFR2+ positive cells were 0.00±0.00% and CD133+/CD34-/VEGFR2+ positive cells were 26.08±2.68% (Figure S2). The cells labeled with CD133, CD34 and VEGFR2 suggest that the ratio of EOCs was 71.61±2.84%. The cells labeled with CD133 and VEGFR2 but not CD133 and the cells labeled with CD133 and VEGFR2 but not CD34 suggest that the ratio of ECFCs was 26.08±2.68%.

Based on these results, we have revised the manuscript as the following:

In the Introduction part, the sentences “EPCs have high proliferative and differentiation capacity, and can induce neovascularization in ischemic areas through two independent mechanisms. On the one hand, EPCs can differentiate into mature endothelial cells, and directly promote neovascularization by incorporating themselves into newly formed vessels. On the other hand, EPCs can stimulate the proliferation and migration of endothelial cells and recruit more EPCs from blood circulation system by releasing pro-angiogenic factors and chemokines such as stromal cell-derived factor 1 (SDF-1) and vascular endothelial growth factor (VEGF) in a paracrine manner.....” have been changed as “EPCs have high proliferative and differentiation capacity, and can induce neovascularization in ischemic areas through two independent mechanisms because of two subgroups named as early outgrowth cells (EOCs) and endothelial colony-forming cells (ECFCs)^{7,8}. On the one hand, ECFCs can differentiate into mature endothelial cells, and directly promote neovascularization by incorporating themselves into newly formed vessels⁸. On the other hand, EOCs can stimulate the proliferation and migration of endothelial cells and recruit more EPCs from blood circulation system by releasing pro-angiogenic factors and chemokines such as stromal cell-derived factor 1 (SDF-1) and vascular endothelial growth factor (VEGF) in a paracrine manner⁹⁻¹¹ although itself does not new blood vessels, which indirectly stimulates angiogenesis^{12,13}.” in the revised manuscript. (Page 3 Line 12-20)

In the Results part, the sentence “Furthermore, EOCs within the isolated EPCs were characterized by co-expression of CD133, CD34 and VEGFR2 (CD133+/CD34+/VEGFR2+), and ECFCs were characterized by co-expression of CD34 and VEGFR2 (CD133-/CD34+/VEGFR2+), or CD133 and VEGFR2 (CD133+/CD34-/VEGFR2+)^{7,8,37-40}. Therefore, in the isolated EPCs, about 71.61±2.84% cells were EOCs, and 26.08±2.68% cells were ECFCs (Figure S2).” has been added in the revised manuscript. (Page 6 Line 10-15)

Furthermore, in this study, we observed that DiO-EPCs injected through the tail vein formed a vascular ring in the myocardial infarction area, which indicates that possibly a small amount of ECFCs within the EPCs participated in the process of angiogenesis.

Therefore, In the Results part, the sentences “From Figure S2, we found that 71.61±2.84% of EPCs were EOCs, and 26.08±2.68% were ECFCs. It is known that ECFCs may participate in neovascularization, while EOCs does not⁸. This result indicates that a small amount of ECFCs in the DiO-EPCs participated in the process of angiogenesis.” have been added in the revised manuscript. (Page 20 Line 15-18)

In the Discussion part, the sentence “.....but also recruited and promoted the homing and angiogenic function of EPCs.” have been changed to “.....but also recruited and promoted the homing and angiogenic function of EPCs, and ECFCs within the recruited EPCs population may directly involved in the formation of vascular structures.” in the revised manuscript. (Page 34 Line 7-9)

In the Methods part, the sentences “Furthermore, the subgroups of EPCs (EOCs or ECFCs) were characterized by the flow cytometry analysis with triple fluorescent markers for three times (APC-A: VEGFR2; PE-A: CD133; FITC-A: CD34) (n=3). The percentage of the subgroups of EPCs were calculated as follows. The percentage of EOCs (CD133+/CD34+/VEGFR2+) = The percentage of VEGFR2+ * The percentage of CD133+/CD34+; The percentage of ECFCs (CD133-/CD34+/VEGFR2+) = The percentage of

VEGFR2+ (N) * The percentage of CD133-/CD34+; The percentage of ECFCs (CD133+/CD34-/VEGFR2+) = The percentage of VEGFR2+ * The percentage of CD133+/CD34-.” have been added in the revised manuscript. (Page 36 Line 15-22)

The following new references have been cited in the revised manuscript:

- 7 Abdulkadir, R. R., Alwjwaj, M., Othman, O. A., Rakkar, K. & Bayraktutan, U. Outgrowth endothelial cells form a functional cerebral barrier and restore its integrity after damage. *Neural Regeneration Research* 15, 1071-1078, doi:10.4103/1673-5374.269029 (2020).
- 8 Tasev, D. et al. CD34 expression modulates tube-forming capacity and barrier properties of peripheral blood-derived endothelial colony-forming cells (ECFCs). *Angiogenesis* 19, 325-338, doi:10.1007/s10456-016-9506-9 (2016).
- 9 Grunewald, M. et al. VEGF-induced adult neovascularization: recruitment, retention, and role of accessory cells. *Cell* 124, 175-189, doi:10.1016/j.cell.2005.10.036 (2006).
- 10 Yamaguchi, J. et al. Stromal cell-derived factor-1 effects on ex vivo expanded endothelial progenitor cell recruitment for ischemic neovascularization. *Circulation* 107, 1322-1328, doi:10.1161/01.cir.0000055313.77510.22 (2003).
- 11 Bammert, T. D. et al. Phenotypic differences in early outgrowth angiogenic cells based on in vitro cultivation. *Cytotechnology* 71, 665-670, doi:10.1007/s10616-019-00305-6 (2019).
- 37 Su, S. H. et al. Dysregulation of Vascular Endothelial Growth Factor Receptor-2 by Multiple miRNAs in Endothelial Colony-Forming Cells of Coronary Artery Disease. *Journal of Vascular Research* 54, 22-32, doi:10.1159/000449202 (2017).
- 38 Dome, B. et al. Identification and clinical significance of circulating endothelial progenitor cells in human non-small cell lung cancer. *Cancer Research* 66, 7341-7347, doi:10.1158/0008-5472.can-05-4654 (2006).
- 39 Mauro, E. et al. Mobilization of endothelial progenitor cells in patients with hematological malignancies after treatment with filgrastim and chemotherapy for autologous transplantation. *European Journal of Haematology* 78, 374-380, doi:10.1111/j.1600-0609.2007.00831.x (2007).
- 40 Friedrich, E. B., Walenta, K., Scharlau, J., Nickenig, G. & Werner, N. CD34(-)/CD133(+)/VEGFR-2(+) endothelial progenitor cell subpopulation with potent vasoregenerative capacities. *Circulation Research* 98, E20-E25, doi:10.1161/01.res.0000205765.28940.93 (2006).

2. a. In addition, their figure S1 indicates the variety of cell types found in their mixture of cells. They should indicate the variation associated with their estimate in their figure and text. b. Which subgroups of cells were found in their mixture? Most must have been early outgrowth cells (EOCs). However, they may have a mixture of EOCs and ECFCs. Please specify the mean and variation of the cell types.

a. In addition, their figure S1 indicates the variety of cell types found in their mixture of cells. They should indicate the variation associated with their estimate in their figure and text.

Re: According to the reviewer’s suggestion, we have performed flow cytometry analysis with single fluorescent marker analysis for 3 times. The results showed that CD133 Positive Cells/Total Cells was 85.3±0.2%, CD34 Positive Cells/Total Cells was 80.9±0.4% and VEGFR2 Positive Cells/Total Cells was 91.9±0.4%. This results with the variation have been added in the revised Figure S1.

In the Results part, the sentence “The results showed that 98.6% of the cells expressed CD133, 74.5% expressed CD34, and 87.4% expressed VEGFR2.” has been changed as “The results showed that 85.3±0.2% of the cells expressed CD133, 80.9±0.4% expressed CD34, and 91.9±0.4% expressed VEGFR2.” in the revised manuscript. (Page 6 Line 7-8)

b. Which subgroups of cells were found in their mixture? Most must have been early outgrowth cells (EOCs). However, they may have a mixture of EOCs and ECFCs. Please specify the mean and variation of the cell types.

Re: Therefore, we have performed the flow cytometry analysis with triple fluorescent markers and found that CD133+/CD34+/VEGFR2+ positive cells were 71.61±2.84%, CD133-/CD34+/VEGFR2+ positive cells were 0.00±0.00% and CD133+/CD34-/VEGFR2+ positive cells were 26.08±2.68%. The cells labeled with CD133, CD34 and VEGFR2 suggest that the ratio of EOCs was 71.61±2.84%. The cells labeled with CD133 and VEGFR2 but not CD34 and the cells labeled with CD133 and VEGFR2 but not CD34 suggest that the ratio of ECFCs was 26.08±2.68%. The new results have been shown in Figure S2.

In the Results part, the sentence “Furthermore, EOCs within the isolated EPCs were characterized by co-expression of CD133, CD34 and VEGFR2 (CD133+/CD34+/VEGFR2+), and ECFCs were characterized by co-expression of CD34 and VEGFR2 (CD133-/CD34+/VEGFR2+), or CD133 and VEGFR2 (CD133+/CD34-/VEGFR2+)^{7,8,37-40}. Therefore, in the isolated EPCs, about 71.61±2.84% cells were EOCs, and 26.08±2.68% cells were ECFCs (Figure S2).” has been added in the revised manuscript. (Page 6 Line 10-15)

In the Methods part, the sentences “Furthermore, the subgroups of EPCs (EOCs or ECFCs) were characterized by the flow cytometry analysis with triple fluorescent markers for three times (APC-A: VEGFR2; PE-A: CD133; FITC-A: CD34) (n=3). The percentage of the subgroups of EPCs were calculated as follows. The percentage of EOCs (CD133+/CD34+/VEGFR2+) = The percentage of VEGFR2+ * The percentage of CD133+/CD34+; The percentage of ECFCs (CD133-/CD34+/VEGFR2+) = The percentage of VEGFR2+ (N) * The percentage of CD133-/CD34+; The percentage of ECFCs (CD133+/CD34-/VEGFR2+) = The percentage of VEGFR2+ * The percentage of CD133+/CD34-.” have been added in the revised manuscript. (Page 36 Line 15-22)

The following new references have been cited in the revised manuscript:

7 Abdulkadir, R. R., Alwjwaj, M., Othman, O. A., Rakkar, K. & Bayraktutan, U. Outgrowth endothelial cells form a functional cerebral barrier and restore its integrity after damage. *Neural Regeneration Research* 15, 1071-1078, doi:10.4103/1673-5374.269029 (2020).

8 Tasev, D. et al. CD34 expression modulates tube-forming capacity and barrier properties of peripheral blood-derived endothelial colony-forming cells (ECFCs). *Angiogenesis* 19, 325-338, doi:10.1007/s10456-016-9506-9 (2016).

37 Su, S. H. et al. Dysregulation of Vascular Endothelial Growth Factor Receptor-2 by Multiple miRNAs in Endothelial Colony-Forming Cells of Coronary Artery Disease. *Journal of Vascular Research* 54, 22-32, doi:10.1159/000449202 (2017).

38 Dome, B. et al. Identification and clinical significance of circulating endothelial progenitor cells in human non-small cell lung cancer. *Cancer Research* 66, 7341-7347, doi:10.1158/0008-5472.can-05-4654 (2006).

39 Mauro, E. et al. Mobilization of endothelial progenitor cells in patients with hematological malignancies after treatment with filgrastim and chemotherapy for autologous

transplantation. *European Journal of Haematology* 78, 374-380, doi:10.1111/j.1600-0609.2007.00831.x (2007).

40 Friedrich, E. B., Walenta, K., Scharlau, J., Nickenig, G. & Werner, N. CD34(-)/CD133(+)/VEGFR-2(+) endothelial progenitor cell subpopulation with potent vasoregenerative capacities. *Circulation Research* 98, E20-E25, doi:10.1161/01.res.0000205765.28940.93 (2006).

3. The authors have provided more information about their exosomes. However, these particles have a range of sizes and contents. Recent Consensus Statements have recommended that these particles be called extracellular vesicles (*Journal of Extracellular Vesicles* 2018 (7), 1535750 <https://doi.org/10.1080/20013078.2018.1535750>). In addition, most authors have noted that the diameter of EVs depends on the detection method employed. Therefore, the consistency of these particles may not correspond to previous reports employing different techniques. Extracellular vesicles can include small EVs (40-150 nm), large EVs (100-1000 nm), extracellular autophagic vesicles (40-1000 nm) and apoptotic vesicles (100-1000 nm). What variety of particles were employed in this study? The authors should consider using the term extracellular vesicles rather than exosomes.

Re: According to the TEM and NTA results, the particle size obtained in this study was about 100 nm, which is in the range of small extracellular vesicles (Figure 1). We agree with the reviewer that the particle size results may be dependent on different detection techniques. Therefore, according to reviewer's suggestion, we have revised the manuscript by changing the term "exosomes" to "extracellular vesicles". In the results part, the sentence "Since the particle size was between 40-150 nm, the obtained EVs belongs to small EVs⁴¹." has been added in the revised manuscript. (Page 7 Line 11-12)

The following new reference has been cited in the revised manuscript:

41 Thery, C. et al. Minimal information for studies of extracellular vesicles 2018 (MISEV2018): a position statement of the International Society for Extracellular Vesicles and update of the MISEV2014 guidelines. *Journal of Extracellular Vesicles* 7, doi:10.1080/20013078.2018.1535750 (2018).

4. The authors provided an indication of the contents of the exosomes. Could they also add the variation of the measurements and how many measurements were made? They should also comment on the significance of this variation.

Re: In this study, we measured the contents of the extracellular vesicles for 3 times. We have added the mean values in the revised Table S3-S7.

In the results part, the sentences "The angiogenic factor content was as follows: EPC-EV: VEGFA: $(1.05 \pm 0.17) \times 10^{-15} \mu\text{g}$; SDF-1: $(3.78 \pm 0.44) \times 10^{-14} \mu\text{g}$; IGF-1: $(1.63 \pm 0.05) \times 10^{-16} \mu\text{g}$; eNOS: $(1.09 \pm 0.05) \times 10^{-13} \mu\text{g}$; HGF: $(0.73 \pm 0.11) \times 10^{-11} \mu\text{g}$. CS-EPC-EV: VEGFA: $(2.52 \pm 0.53) \times 10^{-15} \mu\text{g}$; SDF-1: $(0.70 \pm 0.06) \times 10^{-13} \mu\text{g}$; IGF-1: $(2.29 \pm 0.18) \times 10^{-16} \mu\text{g}$; eNOS: $(1.70 \pm 0.22) \times 10^{-13} \mu\text{g}$; HGF: $(1.24 \pm 0.01) \times 10^{-11} \mu\text{g}$. The maximum measurements variation was in the VEGFA groups, and the VEGFA variation value was 21% in CS-EPC-EV and 16% in EPC-EV. This variation was comparable with the variation of angiogenic factor content in EVs in other studies^{42,43}." have been added in the revised manuscript. (Page 7 Line 23-29)

The following new references have been cited in the revised manuscript:

42 Torreggiani, E. et al. EXOSOMES: NOVEL EFFECTORS OF HUMAN PLATELET LYSATE ACTIVITY. *European Cells & Materials* 28, 137-151, doi:10.22203/eCM.v028a11 (2014).

43 Yurtsever, A. et al. Structural and mechanical characteristics of exosomes from osteosarcoma cells explored by 3D-atomic force microscopy. *Nanoscale* 13, 6661-6677, doi:10.1039/d0nr09178b (2021).

5. In figure S4, the authors show co-localization of PKH26 labeled EXOs and endothelial cells, cardiomyocytes and fibroblasts. Could they also indicate the number of slides evaluated per group and the mean and variation of co-localized cells per high powered field?

Re: According to the reviewer's suggestion, the number of PKH26-EVs co-localized cells in high powered field have been counted and 5 slides per group were analyzed (n=5). The results have been added in revised Figure S5B.

In the Results part, the sentence "Moreover, we counted the number of PKH26-EVs co-localized cells in high magnification field (Cardiomyocytes (CTAT and PKH26 co-localized): 19.6±3.8; Endothelial cells (CD31 and PKH26 co-localized): 6.6±1.1; Fibroblasts (Vimentin and PKH26 co-localized): 8.8±1.5.), and the results showed that the cell uptake of the PKH26-EVs in cardiomyocytes was the highest as compared to other cells (Figure S5B)." has been added in the revised manuscript. (Page 20 Line 25-29)

In the Methods part, the sentence "Five random pictures (high magnification field, 60X) were taken by confocal laser scanning microscope (Leica TCS SP8, Germany). The number of PKH26-EVs co-localized cells was counted and 5 slides were counted in each group." has been added in the revised manuscript. (Page 44 Line 4-6)

6. The authors agree that many mRNA species were likely upregulated by the exosomes. They should not state that one is the "strongest angiogenesis promoting miRNA" unless they have performed comparative studies.

Re: According to reviewer's suggestion, the sentence "In particular, the amount of miR-126a-3p is the highest among these miRNAs, and which is known as one of the strongest angiogenesis promoting miRNA among these miRNAs⁷⁰." has been changed as "In particular, the amount of miR-126a-3p is the highest among these miRNAs, which has been known as a strong angiogenesis promoting miRNA⁷²." in the revised manuscript. (Page 32 Line 4-5)

7. The authors state: "In this study, 20µL solution contains 20µg of EXOs. According to reviewer's comments, we have calculated the number of EXOs particles in 1µg EXOs, which contains 2.44*10⁹ particles. Therefore, 48.8*10⁹ particles have been injected into the myocardial infarction site." However, could they provide the variation associated with their estimate?

Re: We thank for the reviewer's suggestion. The variation associated with the number of the particles have been supplemented in the Table S4. Based on this result, we have calculated the variation of the particles injected into the myocardial infarction site[(46.0±3.6)*10⁹ particles of EPC-EV and (47.3±1.8)*10⁹ particles of CS-EPC-EV]. The results have been added in revised Table S4.

In the Methods part, the sentence "The total number of EPC-EV and CS-EPC-EV particles in each injection was (46.0±3.6)*10⁹ and (47.3±1.8)*10⁹, respectively (Table S4)." has been

added in the revised manuscript. (Page 42 Line 16-17)

8. In their introduction, the authors state that previous studies only evaluated the therapeutic effect of exosomes for 72 hours. They cannot conclude that the effects did not persist beyond 72 hours if the studies did not investigate the effects beyond 72 hours.

Re: According to the reviewer's suggestion, we have revised the description in the introduction. The sentence "Furthermore, considering the treatment of myocardial infarction using EXOs, most studies using EXOs injection to treat myocardial infarction only observed the therapeutic effect within 72 hours²⁷⁻²⁹ possibly due to the low retention rate of EXOs in vivo." has been changed as "Furthermore, most studies using EVs injection to treat myocardial infarction within 72 hours³⁰⁻³², and some studies have found that the treatment effect of EVs alone on myocardial infarction within 28 days was not as good as that of the EVs encapsulated by hydrogels^{33,34}. Based on these researches, we speculated that the low retention rate of EVs in vivo may be one of the main reasons for the limited therapeutic effect." in the revised manuscript. (Page 4 Line 24-28)

The following new references have been cited in the revised manuscript:

33 Hu, X. Y. et al. Islet-1 Mesenchymal Stem Cells-Derived Exosome-Incorporated Angiogenin-1 Hydrogel for Enhanced Acute Myocardial Infarction Therapy. *Acs Applied Materials & Interfaces* 14, 36289-36303, doi:10.1021/acsami.2c04686 (2022).

34 Han, C. S. et al. Human umbilical cord mesenchymal stem cell derived exosomes encapsulated in functional peptide hydrogels promote cardiac repair. *Biomaterials Science* 7, 2920-2933, doi:10.1039/c9bm00101h (2019).

9. The authors state: "we have applied for two patents on silicate bioceramic ion solutions for the treatment of myocardial infarction." Although those patents do not involve exosomes, they do represent competing interests and should be disclosed.

Re: According to the reviewer's suggestion, in the Competing Interests part, the sentences "Shanghai Institute of Ceramics Chinese Academy of Sciences has applied for two patents, related to silicate bioceramic ion solutions for the treatment of myocardial infarction, in which one has been granted and another is pending. J.C. and C.O. are the inventors of these two patents." have been added in the revised manuscript. (Page 45 Line 10-12)

10. The authors agree that their method of stimulating exosome release from cells is one of many approaches which may be beneficial. They state: "more careful comparative studies will be required to evaluate the applicability of the CS-EPC-EXO exosomes." That sentence should be added to their discussion rather than the sentence: "CS is a unique biomaterial, which has good anti-inflammatory and vascular regeneration effects." Their paper should provide a balanced view.

The also state: "more careful comparative studies will be required to evaluate the applicability of the CS-EPC-EXO exosomes." This sentence should be added to their discussion.

Re: We thank for the reviewer's suggestion. In the discussion part, the sentence "However, more careful comparative studies will be required to further evaluate the applicability of the CS-EPC-EV extracellular vesicles." has been added in the revised manuscript. (Page 31 Line 28-29)

11. For both the size and purity of the exosomes, the authors should indicate the variability of their measurements. The authors report the purity of commercial kits as $>1.5 \times 10^9$. Since they are unable to statistically compare their results to those of the commercial kits, they cannot determine whether their purity is equivalent. All they can do is to report the numbers and they should not include their comments on the comparison since no comparison was made.

Re: According to the reviewer's suggestion, the variability of measurements have been indicated in revised Table S7. Furthermore, the comments on the comparison with commercial kits have been deleted in the revised manuscript.

Therefore, in the results part, the sentence "The purity of the CS-EPC-EXO was 2.32×10^9 particles/ μg , and the purity of the EPC-EXO was 2.43×10^9 particles/ μg (Table S7), which was in the same level as the purity of exosomes separated by most commercial kits ($>1.5 \times 10^9$)." has been changed as "The purity of CS-EPC-EV and EPC-EV was $(2.32 \pm 0.14) \times 10^9$ and $(2.43 \pm 0.19) \times 10^9$ particles/ μg , respectively (Table S7)." in the revised manuscript. (Page 7 Line 16-17)

12. The authors state: "We agree with the reviewer that the in vitro activity effect of the exosomes will be different than the in vivo one." This statement should be added to their discussion.

Re: According to the reviewer's suggestion, the sentence "However, it must be mentioned that the activity of EVs in vitro is different from that in vivo, and more studies are needed to further explore the biological mechanism of CS-EPC-EV in vivo." have been added in the revised manuscript. (Page 35 Line 5-7)

13. They also state: "We selected the primary doses of the exosomes for in vivo experiment based on the literature³⁵." However, reference 35 is a careful review of the literature and Kennedy and colleagues concluded: "Two out of the 12 in vivo studies [24,32] failed to report the dose of EVs used; out of the remaining studies, variations in isolation method, dosing unit (particle number, μg protein or number of parent cells from which EVs were sourced), animal model (mouse or rat) and route of administration (intramyocardial injection; i.m.; or tail vein; t.v.) render all studies incomparable to each other." How did the authors derive their dose of exosomes? Also note that Kennedy and colleagues employ the term extracellular vesicles rather than exosomes.

Re: We thank for the reviewer's comments. In fact, we determined the injection dose according to the references^{42,43} listed in this review (Trends in Cardiovascular Medicine 31, 405-415, doi:10.1016/j.tcm.2020.08.003 (2021).), in which the minimum dose of extracellular vesicles' protein injection to C57BL/6 mice was 10 μg , and the maximum dose was 50 μg . Therefore, we thought that the injection dose of extracellular vesicles' protein within 10-50 μg was in bio-safe and effective range. Indeed, as indicated by the reviewer that the optimal doses for EV treatment have not been determined based on the review literature, and it is possible that different type of EVs, from different cell types, isolated using different preparation methods may have different optimal application doses.

In the results part, the sentence "The Microsphere+CS-EPC-EXO microspheres were injected into the infarct area of mice to evaluate the therapeutic function at a dosage of 20 μg EXOs based on a standard injection protocol reported in the literature." has been changed as "The Microsphere+CS-EPC-EV microspheres were injected into the infarct area of mice to evaluate the therapeutic function at a dose of 20 μg EVs based on the injection protocol reported in the

literature (10-50µg)^{45,46}. However, it is important to indicate that, considering the difference between different EVs, 20µg may not be the optimal dose for our EVs, and more studies are needed to determine the optimal doses in future.” in the revised manuscript. (Page 15 Line 3-7)

Furthermore, “extracellular vesicles” have been used to replace the “exosomes” in the whole manuscript.

The following new references have been cited in the revised manuscript:

45 Liu, H. B. et al. Exosomes derived from dendritic cells improve cardiac function via activation of CD4(+) T lymphocytes after myocardial infarction. *Journal of Molecular and Cellular Cardiology* 91, 123-133, doi:10.1016/j.yjmcc.2015.12.028 (2016).

46 Khan, M. et al. Embryonic Stem Cell-Derived Exosomes Promote Endogenous Repair Mechanisms and Enhance Cardiac Function Following Myocardial Infarction. *Circulation Research* 117, 52-64, doi:10.1161/circresaha.117.305990 (2015).

14. The authors employed echocardiography to calculate the ventricular volumes. This technique assumes that the heart is spherical and then transform the systolic and diastolic diameter measurements to volumes. In the infarct model, the heart is not spherical and therefore the volumes become less reliable. The best way to measure volumes in this animal model is by conductance. The authors may wish to mention this minor concern in their discussion.

Re: According to the reviewer’s suggestion, we have supplemented this concern in discussion part. The sentence “It must be mentioned that echocardiography was used to calculate ventricular volume in this study, and this technique assumes that the heart is spherical and then transform the systolic and diastolic diameter measurements to volumes. In the infarct model, the heart is not spherical and therefore the volumes become less reliable^{53,54}. Therefore, pressure volume catheter may be used to accurately measure ventricular volume in subsequent studies⁵⁵.” has been added in the revised manuscript. (Page 29 Line 23-28)

The following new references have been cited in the revised manuscript:

53 Kirkpatrick, J. N., Vannan, M. A., Narula, J. & Lang, R. M. Echocardiography in heart failure - Applications, utility, and new horizons. *Journal of the American College of Cardiology* 50, 381-396, doi:10.1016/j.jacc.2007.03.048 (2007).

54 Michel, L. et al. Real-time Pressure-volume Analysis of Acute Myocardial Infarction in Mice. *Jove-Journal of Visualized Experiments*, doi:10.3791/57621 (2018).

55 Chen, C. W. et al. Sustained release of endothelial progenitor cell-derived extracellular vesicles from shear-thinning hydrogels improves angiogenesis and promotes function after myocardial infarction. *Cardiovascular research* 114, 1029-1040, doi:10.1093/cvr/cvy067 (2018).

15. The statistical results section states: “One-way ANOVA and post hoc Bonferroni tests were used to compare differences between more than two groups.” Time related differences between groups (such as figures S6 and S11) must account for multiple testing. Either a two-way time varying ANOVA or an analysis of covariance should be performed to compare groups. Then the authors should report the effects of time and group assignment as well as the interactive effect. If a significant effect is found, they can employ multiple range t tests to specify the differences.

Re: According to the reviewer’s suggestion, two-way time varying ANOVA analysis have been

used firstly to compare the time related differences between groups, and for the groups with significant differences, multiple range t tests have been performed to further specify the differences. According to the results, the significant differences have been revised in Figures S6 and S11.

In the Results part, the sentence “The result showed that the EVs and EVs-containing microsphere groups significantly improved the cardiac function after myocardial infarction, while the cardiac function of PBS group declined gradually with time (Figure 4C and Figure S5 and Figure S6).” has been changed as “When we compare the same group in different time points, the results showed that the cardiac function in the EVs and EVs-containing microsphere groups was significantly improved on day 14 as compared with that on day 7 after myocardial infarction. Furthermore, with prolonged time period up to 21 days, we found that the cardiac function on day 21 was only significantly improved in EVs-containing microsphere groups as compared to that on day 14, while no significant difference was found between pure EVs without microsphere treatment group. In contrast, the cardiac function in PBS group declined with time up to 21 days (Figure 4C and Figure S6 and Figure S7).” in the revised manuscript. (Page 15 Line 26-29 and Page 16 Line 1-4).

The sentence “Quantitative analysis also confirmed significantly higher fluorescence intensity in microspheres group than that in PBS group (Figure S12).” has been changed as “Quantitative analysis also confirmed that the fluorescence intensity decreased obviously from day 0 to day 21 no matter in Microsphere-EV group or EV group. Moreover, at each time point, the fluorescence intensity was significantly higher in Microsphere-EV group than that in EV group (Figure S12).” in the revised manuscript. (Page 15 Line 13-16).

In the Methods part, the sentence “Two-way time varying ANOVA and t-test were used to compare the time related differences between groups.” has been added in the revised manuscript. (Page 44 Line 21-22)

16. The authors state: “We agree with the reviewer, that the in vitro cell migration experiments may not reflect the in vivo events.” This statement should be added to their manuscript.

Re: According to the reviewer’s suggestion, the sentence “However, it must be mentioned that the activity of EVs in vitro is different from that in vivo, and more studies are needed to further explore the biological mechanism of CS-EPC-EV in vivo.” have been added in the revised manuscript. (Page 35 Line 5-7)

17. From the data presented in figure 4J, Immunofluorescence staining of capillaries and small arteries in the peripheral area of myocardial infarction on day 21 after infarction was greater in microsphere+CS-EPC-EXO than microsphere+EPC-EXO. The number (40 vessels/high powered field in the periphery of the infarct region) is highly dependent on the protocol employed and the conduct of the experiments. The authors state: “Especially in promoting angiogenesis, the effect of CS induced highly active exosomes is not only higher than that of the exosomes without CS stimulation, it seems also more effective than other approaches for the treatment of myocardial infarction such as cell, gene or protein therapies reported in the literature.” The second half of that sentence must be removed. The authors cannot compare their results to other experimental preparations. More extensive comparative studies are required to confirm their statement.

Re: According to reviewer’s suggestion, we have revised the second half of that sentence. The

sentence “Especially in promoting angiogenesis, the effect of CS induced highly active exosomes is not only higher than that of the exosomes without CS stimulation, it seems also more effective than other approaches for the treatment of myocardial infarction such as cell, gene or protein therapies reported in the literature⁴³⁻⁴⁵, but this needs to be proofed by careful comparative studies.” has been changed as “Especially in promoting angiogenesis, the effect of CS induced highly active extracellular vesicles is significantly higher than that of the extracellular vesicles without CS stimulation.” in the revised manuscript. (Page 29 Line 21-23)

REVIEWERS' COMMENTS

Reviewer #3 (Remarks to the Author):

The authors have adequately responded to the concerns of this reviewer - congratulations. However, the manuscript needs to be carefully reviewed for English wording. The final number of references may be excessive and could be reduced.

Reviewer #3 (Remarks to the Author):

1. The authors have adequately responded to the concerns of this reviewer - congratulations. However, the manuscript needs to be carefully reviewed for English wording. The final number of references may be excessive and could be reduced.

Re: We greatly appreciate the reviewer's congratulations. Following the reviewer's recommendation, we submitted our manuscript to "Springer Nature Author Services" for language editing and have included the editing certificate as shown below.

Furthermore, we have trimmed down some of the excessive references in the manuscript. The total number of references has been reduced from 97 to 80.

The deleted references are listed below, and the reference list and reference citing in the manuscript have been revised:

- 26 Li, H. & Chang, J. Bioactive silicate materials stimulate angiogenesis in fibroblast and endothelial cell co-culture system through paracrine effect. *Acta biomaterialia* 9, 6981-6991, doi:10.1016/j.actbio.2013.02.014 (2013).
- 52 Wang, X. Y. et al. Silicon-Enhanced Adipogenesis and Angiogenesis for Vascularized Adipose Tissue Engineering. *Adv. Sci.* 5, 15, doi:10.1002/advs.201800776 (2018).
- 53 Kirkpatrick, J. N., Vannan, M. A., Narula, J. & Lang, R. M. Echocardiography in heart failure - Applications, utility, and new horizons. *Journal of the American College of Cardiology* 50, 381-396, doi:10.1016/j.jacc.2007.03.048 (2007).
- 54 Michel, L. et al. Real-time Pressure-volume Analysis of Acute Myocardial Infarction in Mice. *Jove-Journal of Visualized Experiments*, doi:10.3791/57621 (2018).
- 55 Chen, C. W. et al. Sustained release of endothelial progenitor cell-derived extracellular vesicles from shear-thinning hydrogels improves angiogenesis and promotes function after myocardial infarction. *Cardiovascular research* 114, 1029-1040, doi:10.1093/cvr/cvy067 (2018).
- 68 Wang, X. T. et al. Chitosan/Calcium Silicate Cardiac Patch Stimulates Cardiomyocyte Activity and Myocardial Performance after Infarction by Synergistic Effect of Bioactive Ions and Aligned Nanostructure. *Acs Applied Materials & Interfaces* 11, 1449-1468, doi:10.1021/acsami.8b17754 (2019).
- 83 Yu, B. et al. Bimodal Imaging-Visible Nanomedicine Integrating CXCR4 and VEGFa

Genes Directs Synergistic Reendothelialization of Endothelial Progenitor Cells. *Advanced science* (Weinheim, Baden-Wurttemberg, Germany) 7, 2001657, doi:10.1002/advs.202001657 (2020).

84 Ziegler, M. et al. The bispecific SDF1-GPVI fusion protein preserves myocardial function after transient ischemia in mice. *Circulation* 125, 685-696, doi:10.1161/circulationaha.111.070508 (2012).

85 Westenbrink, B. D. et al. Erythropoietin improves cardiac function through endothelial progenitor cell and vascular endothelial growth factor mediated neovascularization. *European heart journal* 28, 2018-2027, doi:10.1093/eurheartj/ehm177 (2007).

86 Claes, F., Vandeveld, W., Moons, L. & Tjwa, M. Another angiogenesis-independent role for VEGF: SDF1-dependent cardiac repair via cardiac stem cells. *Cardiovascular research* 91, 369-370, doi:10.1093/cvr/cvr184 (2011).

89 Carneiro, G. D., Godoy, J. A. P., Werneck, C. C. & Vicente, C. P. Differentiation of C57/BL6 mice bone marrow mononuclear cells into early endothelial progenitors cells in different culture conditions. *Cell Biology International* 39, 1138-1150, doi:10.1002/cbin.10487 (2015).

90 Zhang, Z. et al. Design of a biofluid-absorbing bioactive sandwich-structured Zn-Si bioceramic composite wound dressing for hair follicle regeneration and skin burn wound healing. *Bioactive Materials* 6, 1910-1920, doi:https://doi.org/10.1016/j.bioactmat.2020.12.006 (2021).

91 Zhou, F. et al. LncRNA H19 abrogates the protective effects of curcumin on rat carotid balloon injury via activating Wnt/ β -catenin signaling pathway. *European journal of pharmacology* 910, 174485, doi:10.1016/j.ejphar.2021.174485 (2021).

92 Li, H. et al. Injectable AuNP-HA matrix with localized stiffness enhances the formation of gap junction in engrafted human induced pluripotent stem cell-derived cardiomyocytes and promotes cardiac repair. *Biomaterials* 279, 121231, doi:10.1016/j.biomaterials.2021.121231 (2021).

93 Li, H. K. et al. Injectable AuNP-HA matrix with localized stiffness enhances the formation of gap junction in engrafted human induced pluripotent stem cell-derived cardiomyocytes and promotes cardiac repair. *Biomaterials* 279, doi:10.1016/j.biomaterials.2021.121231 (2021).

94 Wu, Y. et al. Release of VEGF and BMP9 from injectable alginate based composite hydrogel for treatment of myocardial infarction. *Bioactive Materials* 6, 520-528, doi:10.1016/j.bioactmat.2020.08.031 (2021).

95 Wang, X. H. et al. Intra-Myocardial Injection of Both Growth Factors and Heart Derived Sca-1(+)/CD31(-) Cells Attenuates Post-MI LV Remodeling More Than Does Cell Transplantation Alone: Neither Intervention Enhances Functionally Significant Cardiomyocyte Regeneration. *Plos One* 9, doi:10.1371/journal.pone.0095247 (2014).